# Self-Supervised Visual Representation Learning for Medical Image Analysis: A Comprehensive Survey

**Siladittya Manna**                                                        *siladittya_r@isical.ac.in*
*Computer Vision and Pattern Recognition Unit*
*Indian Statistical Institute, Kolkata*

**Saumik Bhattacharya**                                                     *saumik@ece.iitkgp.ac.in*
*Department of Electronics and Electrical Communication Engineering*
*Indian Institute of Technology Kharagpur*

**Umapada Pal**                                                             *umapada@isical.ac.in*
*Computer Vision and Pattern Recognition Unit*
*Indian Statistical Institute, Kolkata*

**Reviewed on OpenReview:** *https://openreview.net/forum?id=3Wg1oErMcJ*

## Abstract

Deep learning has developed as a great tool for many computer vision or natural language processing tasks. However, supervised deep learning algorithms require a large amount of labelled data to achieve satisfactory performance. Self-supervised learning, a subcategory of unsupervised learning, circumvents the issue of the requirement of a large amount of data by learning representations from the data without labelled examples. Over the past few years, Self-supervised learning has been applied to various tasks to achieve performance at par with or surpassing the supervised counterparts in several tasks. However, the progress has been so rapid, that a comprehensive account of these developments is lacking. In this study, we attempt to present a review of those methods and show how the self-supervised learning paradigm evolved over the years. Additionally, we also present an exhaustive review of the self-supervised methods applied to medical image analysis. Furthermore, we also present an extensive compilation of the details of the datasets used in the different works and provide performance metrics of some notable works on image and video datasets.

## 1 Introduction

The advent of Machine Learning has boosted the development of different fields of study such as Artificial Intelligence, Computer Vision, and Natural Language Processing. Although there are classical methods, machine learning algorithms have outperformed almost all classical algorithms in various applications. With the invention of artificial neural networks and the increasing availability of computational capabilities to researchers, the upscaling of artificial neural networks has become possible. This led to the advent of the deep learning paradigm.

Deep learning has played a crucial role in facilitating researchers to make substantial progress in numerous fields such as signal processing, computer vision (CV), natural language processing (NLP), time series analysis, and others. Over the years different architectures with parameters ranging from millions to billions have been proposed and used to achieve almost human-level performance in various tasks like object detection, segmentation, and classification, or machine translation, chatbots, or even multi-modal applications like captioning, visual question answering, etc. Deep learning has also found its way to applications in medical image

analysis. Applications in brain MRI, knee MRI, colonoscopy videos, chest X-ray images, mammograms, etc. are plentiful.

However, supervised deep learning has its share of pros and cons. One drawback of supervised deep learning methods is the requirement for large amounts of labeled data. Without that, supervised deep learning models tend to overfit and fail to generalise. Even if data scarcity is not an issue, supervised deep learning models require long training periods to achieve satisfactory performance. To prevent overfitting problems, researchers often use transfer learning techniques to train supervised deep learning models on small-scale datasets, based on knowledge learnt from training on large-scale datasets. Deep learning models trained on large-scale datasets like ImageNet or MS-COCO are often used as pre-trained models in many applications, even if there is a domain mismatch between the pre-training dataset and the target dataset. In real-life applications such as medical image analysis, labeled data is limited or hard to obtain. Medical scans from mammography or magnetic resonance imaging require expert domain knowledge for efficient and reliable annotation, which proves to be labor-intensive and time-consuming.

To deal with the issues in supervised deep learning, many machine learning paradigms like semi-supervised learning, self-supervised learning, etc. have emerged. In this study, we are going to focus primarily on self-supervised learning algorithms. In later sections, we will also discuss the applications of self-supervised learning strategies in different medical image modalities for representation learning.

Although several previous surveys have also reviewed the work on SSL, none has provided a detailed and minutely tailored discussion of each work like ours. There have been reviews on self-supervised learning such as Gui et al. (2023) which provides a comprehensive overview of the domain and also cited works on its applications in different sub-domains of computer vision, like point clouds, recommender systems, depth estimation, etc. but lacks a detailed overview of those works. Some older surveys like Jing & Tian (2021) and Mao (2020) do not discuss research work conducted after 2020. Other surveys such as Wu et al. (2023b) and Liu et al. (2023d) focus only on techniques related to graph neural networks, while Yu et al. (2024a) and Qi & Shah (2022) discuss contrastive and adversarial techniques, respectively. From the literature, we find three previous surveys on medical image analysis such as Zhang et al. (2023b), Wang et al. (2023f) and Huang et al. (2023). Zhang et al. (2023b) provides a benchmark of popular SSL frameworks and also provides a detailed analysis of the effect of data imbalance in different frameworks. Wang et al. (2023f) comes close to our work but differs in the categorization strategy. We believe that the categorization we followed in our work is fine-grained, providing a better resolution of the general overview for a survey, while Wang et al. (2023f) follows a coarser stratification. Huang et al. (2023) provides an implementation-orientated survey rather than going deep into the concepts of individual frameworks. Another work Balestriero et al. (2023), presents a brief review of the origins of SSL and contemporary contrastive and non-contrastive frameworks. Balestriero et al. (2023) is primarily a cookbook with most part of the work dedicated to discussing the recipes for successful SSL frameworks. However, Balestriero et al. (2023) only traces the origin of SSL only to a few years prior to 2020.

In this survey, our aim is to systematically discuss and categorize the work done in the domain of self-supervised learning. We investigate the roots of self-supervised learning, identifying the first work Bridle et al. (1991) to propose the use of pseudo-labels and the concepts of 'fairness' and 'firmness', currently widely known as uniformity and alignment, respectively, and reinvented in Wang & Isola (2020). We also discuss the work DeSa (1993) which can be considered to be the first work to coin the term "self-supervised learning", a principle published in Bridle et al. (1991) three decades ago to modern day. Most importantly, we have attempted to exhaustively cover the important and substantially contributing works in the domain of SSL, and discuss the contribution in most of them individually. Additionally, we have also exhaustively covered works on medical image analysis using SSL, categorising them two-fold: firstly based on modality and secondly on the SSL strategy. Furthermore, our survey is more updated than the previous ones.

We also systematically compile the details of different datasets used in the works discussed in our survey, and tabulate them according to their respective modality. We also present a compilation of the performance metrics of the notable works in each category of the SSL frameworks on the natural image and video datasets, which gives a broad overview of the evolution of the SSL domain over the years.

### 1.1 What is Self-Supervised Learning?

In the (to the best of our knowledge) first paper on Self-Supervised Learning (SSL) (Bridle et al., 1991), the authors address the problem of classifying data without prior domain knowledge or labeled examples. This leads us to characterize SSL as being primarily concerned with the learning of representation from unlabeled data. Hence, we can say that SSL is a subcategory of Unsupervised Learning. The representations or features learned are then transferred to perform various tasks. Hence, Self-Supervised Learning consists of two phases: (a) Pretext or Pre-training or Surrogate tasks, and (b) Downstream or Target tasks.

Pretext or pre-training tasks are used in self-supervised learning on unlabeled data to learn representations without utilising any human annotations. A common approach is to generate pseudo-labels from the data itself to facilitate learning representations. Since the pretext tasks' sole purpose is to learn representations, a host of algorithms or methods have been used for this purpose. These tasks can be categorized into various types, such as generative, context-based, paired-embedding-based, clustering, or grouping-based methods. We review and discuss these methods individually in the later sections of this study. The pretext tasks aid in obtaining pre-trained weights, similar to ImageNet pre-trained weights, which are then used in the Downstream or Target Tasks. The downstream task can be object detection, classification, segmentation, machine translation, etc. depending on the data, on which pre-training was conducted. In the downstream task, labeled data is used for further fine-tuning and evaluation. Carefully observing the definition of pretext tasks, we can say that (unconditional) GANs and AEs also fall under the purview of SSL.

However, it is not always possible to learn effective and useful representations from the target data itself using self-supervised learning algorithms only. Lately, there has been a development in algorithms, where models or architectures trained on datasets different from those of the target dataset have been used as prior information in self-supervised learning, such as DINO (Caron et al., 2021) in UP-DETR (Dai et al., 2021a), DETReg (Bar et al., 2022). In MoCo-CXR (Sowrirajan et al., 2021), the authors pre-train MoCo on Chest X-ray images using ImageNet pre-trained weights as initialization. As the domain of the data on which the pre-trained architectures are not the same as the target dataset, we can still consider the entire learning process to be self-supervised learning. Using a pre-trained network for extracting information from the data in the pre-training phase does not violate the unwritten rule of self-supervised learning; that is, no information about the target label information is used during the pre-training phase.

### 1.2 Self-supervised Learning and Human Psychology

In Orhan et al. (2020), the authors aim to answer the question behind the origin of infants' ability to learn shapes or animals and to discriminate between them, using ego-centric videos and modern self-supervised learning architectures. The authors intend to determine how much of this knowledge learned by infants can be acquired by generic learning architectures receiving sensory data through the eyes of a developing child, and how much of it requires more substantive inductive biases. The use of self-supervised learning algorithms in this study links this paradigm to the field of psychology.

In fact, from a study by Raymond B. Cattell on the Theory of Fluid and Crystallised Intelligence (Cattell, 1963) and later extended by John Leonard Horn in his doctoral dissertation Horn (1971) and also in Horn & Cattell (1966), we can speculate that the learning paradigm of self-supervised learning is similar to the development of fluid intelligence in humans. As defined in Cattell (1963), fluid general ability refers to the ability to adapt to new situations, whereas crystallised general ability refers to those cognitive abilities in which skilled judgment has become crystallized. In Horn (1971), we learn that fluid intelligence and crystallized intelligence are, in fact, co-dependent. In light of this theory, we can formulate self-supervised learning as the fluid ability to learn novel knowledge bases or representations based on self-designed cues, or in other words, without cues from external agents. When the self-supervised pre-trained weights are transferred to other downstream tasks, it resembles the utilization of fluid intelligence to help build up crystallized intelligence, as described in Horn (1971).

Indeed, the problem statement in Bridle et al. (1991) of learning to classify samples without prior knowledge or labelled examples, fits the analogy of infants having the ability to discriminate between animal classes, which in turn points to the similarity of this learning paradigm with fluid intelligence or ability in the field of psychology.

### 1.3 Motivation of the Survey

Although previous surveys have touched on SSL, none has delved into each work as comprehensively as ours. Although there are reviews on self-supervised learning, they lack a detailed examination of individual works. Our work differs in the categorisation approach, too. The categorisation we employ is more detailed, offering a finer resolution for surveying. Our survey aims to systematically discuss and classify work in the realm of self-supervised learning. We trace the origins of self-supervised learning back to the first work by Bridle et al. (1991), and cover the most recent works too. Importantly, we strive to comprehensively cover the significant and impactful works in the field of SSL, discussing their contributions individually. Additionally, we extensively cover works on medical image analysis using SSL and categorize them based on modality and SSL strategy.

### 1.4 Methodology of the Survey

In this review, we aimed to provide a comprehensive and detailed examination of the SSL landscape. Additionally, we intended to focus on works that apply SSL principles to medical image analysis. Our goal was to highlight significant studies that contribute to both foundational and recent advancements in the field, rather than including every single paper on the topic. The review is structured into two major sections: one covering the foundations of SSL, and the other exploring its applications in medical image analysis. We categorized SSL algorithms based on different algorithmic principles and selected papers chronologically through a thorough literature search.

To ensure a thorough and systematic review, we conducted an extensive literature search using the following databases: PubMed, IEEE Xplore, and Google Scholar. Our search was not limited to a specific timeframe, allowing us to include both pioneering works and the latest research developments.

We used specific search queries to capture a wide range of relevant studies, such as,

- "Self-supervised learning"
- "Contrastive learning"
- "Self-supervised decorrelation"
- "Self-supervised learning medical image analysis"
- "Self-supervised learning magnetic resonance imaging"
- "Self-supervised learning ultrasound"
- "Self-supervised learning echocardiography"

Additionally, we also obtained several studies outside of these search queries from literature mapping databases like Connected Papers, which allowed us to investigate the SSL landscape densely.

**Inclusion and Exclusion Criteria:** The above search keywords yielded numerous publications. However, we intended to choose studies with significant contributions. Hence, we adopted the following criteria.

**Inclusion criteria:**

- Studies proposing novel SSL algorithms.
- Studies presenting analyses on SSL frameworks.
- Studies which applied SSL principles for medical imaging modalities.

**Exclusion criteria:**

- Studies not primarily focused on SSL.
- Studies with applications in domains other than medical image analysis.

- Papers with insufficient experimental validation or unclear methodologies.

We meticulously extracted and categorized relevant information from the selected papers. The broad spectrum of SSL algorithms was divided into sub-categories based on different algorithmic principles, such as contrastive learning, generative approaches, and clustering-based methods. We chronologically organized the papers to highlight the evolution and progress of each subcategory of SSL frameworks and the application of the same SSL techniques to medical image analysis.

By providing a structured and comprehensive review, we aim to offer valuable insights into the current state and future directions of SSL in medical image analysis, aiding researchers and practitioners in navigating this rapidly evolving field.

It is to be noted that, no part of this review was written or edited using large language models like ChatGPT.

## 1.5 Organization of the Survey

In the following subsections, we will discuss the different approaches used for representation learning in the self-supervised learning paradigm. We will start with the classical approaches, such as context-based pretext tasks (Sec. 2.1), wherein we will discuss spatial and temporal context-based pretext tasks for representation learning on both images and videos. Following that, we will discuss the clustering-based frameworks in Sec. 2.2. After that, we dive into the discussion of paired embedding-based methods (Sec. 2.3), which includes both contrastive methods (Sec. 2.3.1) and non-contrastive (Sec. 2.3.2) methods as well. Additionally, we discuss different works in the medical image analysis domain with applications of SSL. We divide the discussion in terms of imaging modality, such as, MRI & CT (Sec. 3.1), Ultrasound (Sec. 3.2), Endoscopy (Sec. 3.3), Radiographs (Sec. 3.4), Retinal images (Sec. 3.5), histopathology (Sec. 3.6), echocardiogram (Sec. 3.7) and skin images (Sec. 3.8). We also discuss two benchmarking datasets in the domain of SSL. Additionally, we summarise the different datasets used in all the works documented in this survey in Sec. 4. In addition to that, we also present a compilation of the performance metrics of some notable works on natural image and video datasets in Sec. 5. In Sec. 6, we discuss the challenges and limitations associated with the current SSL methods in the medical image analysis domain. Finally, we end this survey with a conclusion and discussion on the future directions of SSL in the medical image analysis domain in Sec. 7.

## 2 Self-Supervised Algorithms and Frameworks

Before delving into the discussion of different categories of SSL frameworks, we can get a taxonomical overview of the same from Fig. 1 and 2. For the convenience of the readers, the taxonomy tree has been divided into two parts in the two figures. In these figures (Fig. 1 and 2), we can observe the different levels of hierarchy that the SSL frameworks can be categorized into based on the approach adopted in those works. Fig. 1 and 2 also provide a visual summary of the first part of the survey where we discuss the notable SSL frameworks on natural image and video data. In the following subsections, we delve into a detailed discussion of the different categories of SSL frameworks and understand the basic differences between them.

### 2.1 Context Based Pretext Tasks

In this section, we primarily discuss the different context-based pretext tasks used in self-supervised pretraining. We can categorize them primarily into seven types, namely (Spatial) Context Encoding, geometrical transformation prediction, Jigsaw Puzzle Solving, Colorization, Counting, Spatio-Temporal Context Prediction (used primarily for videos), and Masked Image Modeling. Works that cannot be categorized specifically into any of the above categories have been included in the Miscellaneous category. We will start our discussion with geometrical transformation prediction-based pretext tasks and then continue with the others. The abstract illustration of a few foundational context based frameworks are depicted in Fig. 3.

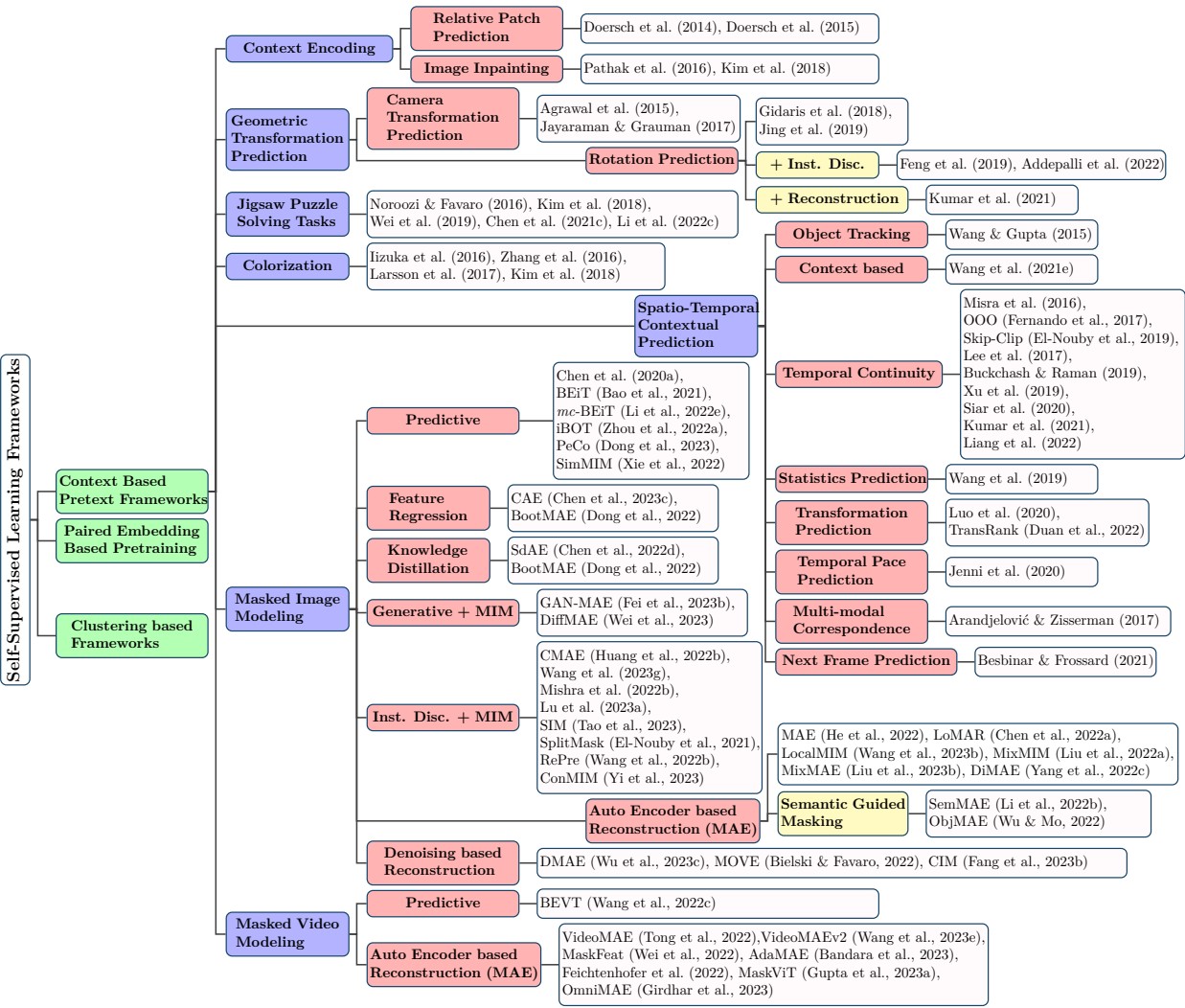

Figure 1: Taxonomy and Summary of SSL Frameworks (Part 1). "Inst. Disc.", "MAE" and "MIM" stands for Instance-Instance Discrimination, Masked Auto Encoder and Masked Image Modelling, respectively.

### 2.1.1 Context Encoding

**Relative Patch Prediction:** Context encoding for unsupervised feature learning was first introduced in Doersch et al. (2014). In Doersch et al. (2014), the authors use the prediction of the context of a single patch as a supervisory task to learn object clusters for unsupervised object discovery. In a later work, Doersch et al. (2015), the authors employed AlexNet (Krizhevsky et al., 2012) to classify the position of a patch with one reference patch sampled a priori as context.

**Image Inpainting:** Generative Context encoding via Image Inpainting was first introduced in Pathak et al. (2016), where the authors adopted a DCGAN (Radford et al., 2016) based generative pipeline, with a joint loss, consisting of $l_2$ and adversarial loss. To prevent a trivial solution, the authors condition the generator on the masked region only. The proposed method showed considerable improvement over the nearest neighbor-based image inpainting method.

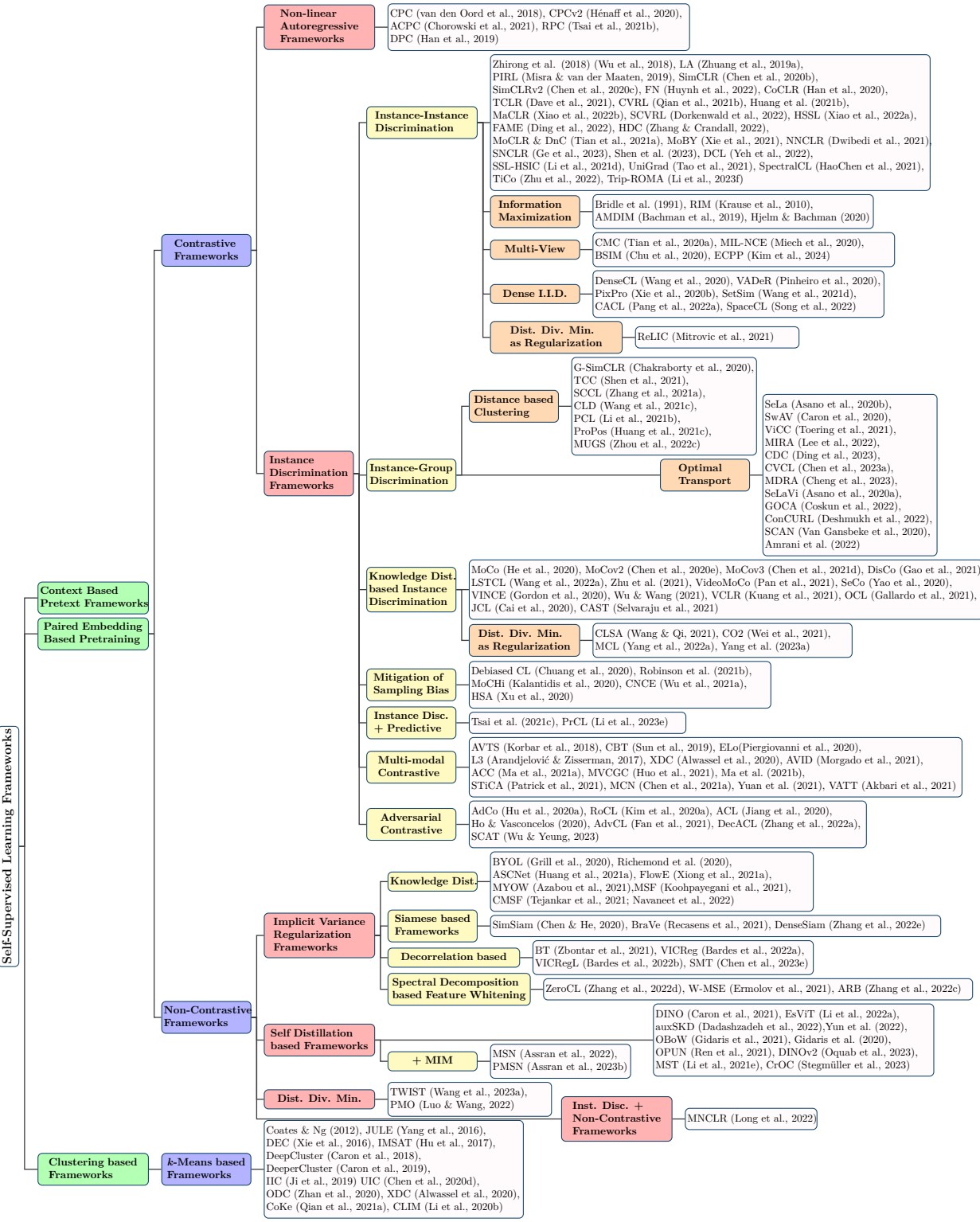

Figure 2: Taxonomy and Summary of SSL Frameworks (Part 2). "I.I.D.", "Dist. Div. Min.", "Inst. Disc." and "MIM" stands for Instance-Instance Discrimination, Distribution Divergence Minimization, Instance Discrimination and Masked Image Modelling, respectively.

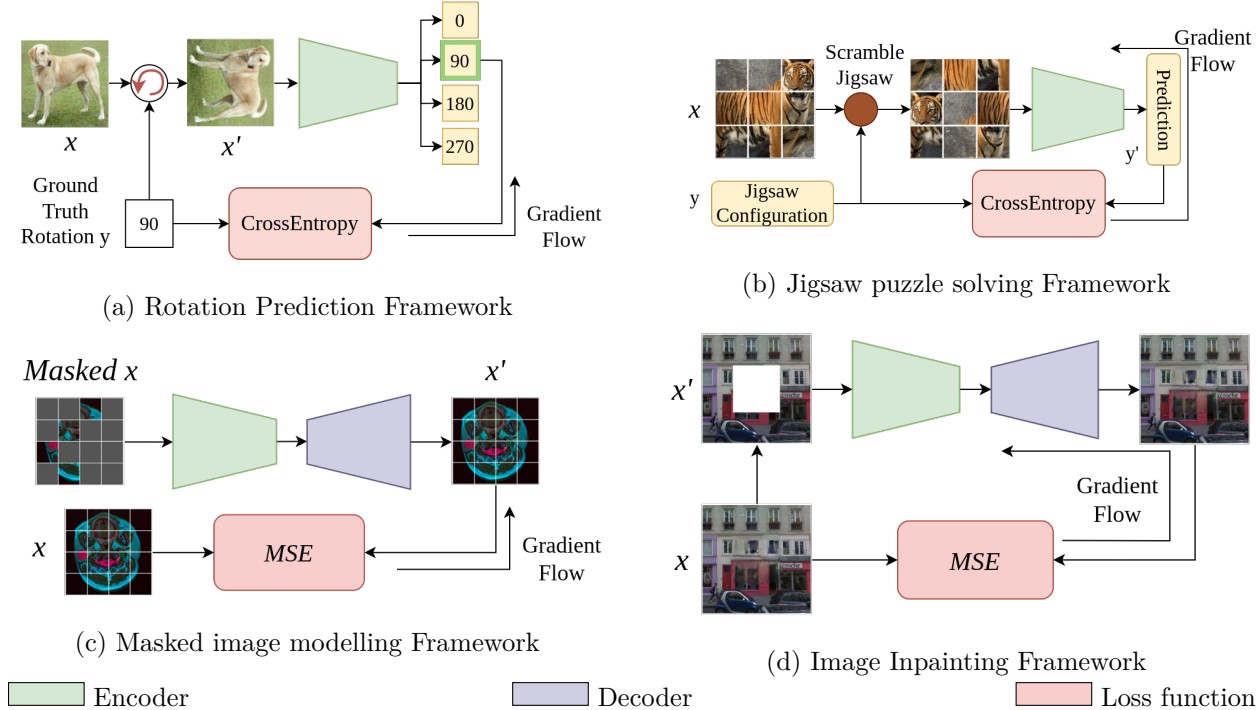

Figure 3: Illustration of Context-based Frameworks. "Gradient flow" indicates the direction along which the parameter gradient propagation occurs, that is, starting from the loss and through the network. "MSE" stands for Mean Squared Error.

### 2.1.2 Geometrical Transformation Prediction

**Camera Transformation Prediction:** The task of using geometric transformation as a supervisory signal was first introduced in Agrawal et al. (2015). The authors used prediction of the camera transformation from pair images as a pretext task. The supervisory signal was obtained from the odometry data in the KITTI (Geiger et al., 2012) and SF (Chen et al., 2011) datasets. Another work along similar lines was presented in Jayaraman & Grauman (2017), where the objective is to learn the ego-motion equivariance from image pairs selected from ego-motion videos, and the sensory signals were used to create pseudo-labels in the pretext task. Jayaraman & Grauman (2017) also combined contrastive loss to enforce equivariance between image pairs with supervised classification loss.

**Rotation Prediction:** In the work RotNet (Gidaris et al., 2018), the authors claimed that training the network to predict the rotation of the images forces the network to learn to locate salient objects in the image and the semantic features in the objects to effectively classify the orientation of the dominant features in the objects.

**Combining with Instance Discrimination:** To improve representation learning, Feng et al. (2019) combined rotation prediction with instance discrimination task to learn both equivariant and rotation invariant representations.

**3D Rotation Prediction:** The work done in RotNet was transferred to videos in Jing et al. (2019), where the authors adopted a 3DCNN architecture (3DRotNet) to account for both spatial and temporal information in videos.

**Combining with Future Frame Prediction:** Deriving from the above work and formulating a multi-task learning scheme by using a 3D Convolutional Auto-Encoder to predict future frames, in addition to classifying rotation applied on the frames, Kumar et al. (2021) outperforms the vanilla 3DRotNet by a considerable margin on video retrieval benchmark tasks.

### 2.1.3 Jigsaw Puzzle Solving

The primary idea behind using jigsaw puzzle solving is to learn spatially invariant contextual information by learning the relative arrangement of patches with each other. The common approach involves dividing the image into several patches and numbering them in order. Then the position of the patches is rearranged, and a deep learning model is trained to predict the arrangement of the patches or generate the original input from the rearranged input.

To the best of our knowledge, the first such work that used jigsaw puzzle solving as a pretext task was presented in Noroozi & Favaro (2016), where the authors presented a context-free network by keeping the computation of features from individual patches remains independent until the first fully-connected layer. This measure adopted by the authors prevents the learning of low-level artifacts by the backbone network. The authors also use a Hamming distance-based selection of a set of arrangements of the patches. Another issue the authors discuss in their work is the learning of *shortcuts* in self-supervised learning. This phenomenon results in the learning of features suitable for solving the pretext task but not the downstream task. In jigsaw puzzle solving, it happens when the network learns to associate each patch to an absolute position, but not based on their textural or structural semantics.

In Kim et al. (2018), where the authors combined jigsaw puzzle solving, image colourization, and image inpainting in a multi-task learning problem for self-supervised learning of representations from images. In Wei et al. (2019), the authors take an iterative reorganization approach by probabilistically assigning patches to a particular position and optimizing the relativistic position assignment of any two patches. The iterative process is repeated until convergence when the patches are all assigned to their optimal positions. In Mundhenk et al. (2017), the authors propose a bag of different methods that are applied for self-supervised learning of representations and also study the effect of each method on performance on the ImageNet benchmark. In another innovative approach Chen et al. (2021c), the authors clustered the patches and predicted the cluster for each patch of an image as a pretext task. JigsawGAN (Li et al., 2022c) combined the flow information from the prediction of the position of the jigsaw patches with the GAN-based generative task for representation learning.

Following a similar strategy to Noroozi & Favaro (2016), jigsaw puzzle solving was also applied to video data in Ahsan et al. (2019). Each frame is divided into a $2 \times 2$ grid and all the patches over all the time steps are rearranged. A network is trained to put the jumbled patches in place by learning both spatial context and temporal order of the events in the frames. The authors also adopt a curriculum learning-based approach by first training the network on an easier jigsaw puzzle-solving task, followed by a harder task. In Kim et al. (2019), the basic idea revolves around classifying the order in which the 16 pieces obtained from selected 4 frames are shuffled using a 3D CNN with separated encoding of each 3D space-time piece of the puzzle to prevent learning of low-level cues or trivial solutions.

### 2.1.4 Colorization

The principle of image colourization as a pretext task is based upon the fact that to colour a grayscale image, the model needs to know the semantic information of the image or scene and the location too. This fundamental principle makes end-to-end image colourization algorithms suitable for learning representations from images without using human-annotated labels.

**Generative:** In the first known work which used image colourization as a pretext task Iizuka et al. (2016), the authors used a combination of self-supervised pre-training and supervised classification for representation learning.

**Predictive:** The next work that utilized colourization as a pretext task was Zhang et al. (2016), where the authors treat the colourization problem as a classification problem. The *ab* colour space is quantized into bins with a grid size of 10, resulting in 313 classes. The authors also took care of class imbalance by using a weighted multinomial classification task as the pretext task for class rebalancing. Next, the work Larsson et al. (2017) is heavily inspired by Larsson et al. (2016). The framework proposed in Larsson et al. (2016)

predicts a colour histogram at each location of the pixels. Furthermore, the framework uses the hypercolumn strategy Hariharan et al. (2015) as per-pixel descriptors.

### 2.1.5 Video Spatiotemporal Contextual Prediction

The objective of video representation learning is to learn both spatial and temporal features. We will discuss the works done in this sub-domain and categorize them according to the pretext strategy used.

**Context-based strategy:** SSCAP (Wang et al., 2021e) uses context-based SSL representation learning frameworks for feature extraction from video frames for subsequent co-occurrence action parsing for action segmentation.

**Transformation Prediction:** In Luo et al. (2020), the strategy adopted consists of generating a number of separate clips from a video in order and removing one clip randomly. Several spatial and temporal operations are applied to the removed clip, and the network is trained to predict the option that has been used to alter the removed clip. Recently, another pretext task was proposed in Duan et al. (2022), where a ranking-based framework was used to learn semantic and temporal information from unlabeled videos by scoring the transformations relative to one another.

**Object Tracking:** In one of the first papers on self-supervised video representation learning Wang & Gupta (2015), two different instances of the same object are obtained by tracking a patch containing an object over the frames of the video, using improved density trajectory on SURF feature points. The primary objective of the learning process was to map the features of the object in two different patches from two different time stamps close to each other. However, to prevent the collapse of representations, hard negative mining for sampling negative samples was used to optimize a ranking loss. Building upon the work done in Doersch et al. (2015) and Wang & Gupta (2015), Wang et al. (2017b) utilized transitive relation invariance, constituted of both inter-instance relations between different object instances of similar appearance and intra-instance relations between identical objects at different timesteps for visual representation learning.

**Using Temporal Continuity:** Using the sequential order of frames in a video can also provide useful information to learn unsupervised representation from videos, as presented in Misra et al. (2016). In this work, the representations are learned by classifying if a tuple consisting of 3 frames sampled from a high-motion window in the video, is in the correct order. Similar to Misra et al. (2016), sequential variation in visual features has been used to learn representations in OOO (Fernando et al., 2017), Lee et al. (2017), Skip-Clip (El-Nouby et al., 2019) as well. In OOO (Fernando et al., 2017), several clips from a video are used as input. All but one clip is in the correct order. The network is tasked with predicting the index of the sample with the frames in the incorrect order. Whereas in Lee et al. (2017), the network is tasked to predict the order in which 4 frames in the input are arranged.

The concept of temporal coherence is perfectly utilized in Skip-Clip (El-Nouby et al., 2019), where the objective of ranking clips based on a given context clip as plausible future clips of the given context is used as self-supervision. A detailed analysis of the method presented in Misra et al. (2016) was conducted in Buckchash & Raman (2019) and improved by using a different sampling technique. A different take on learning representations from videos was proposed in Xu et al. (2019), where a clip-based order ranking strategy with 3D CNN as the backbone was used. A similar approach is also used in Siar et al. (2020) by using 3D CNN to predict the order of the frames selected randomly from non-overlapping clips from a video.

The effect of combining multiple pretext tasks in video representation learning is shown in Liang et al. (2022), where the authors jointly optimize three types of losses to learn representations from videos. Firstly, a binary classification loss to classify clips from the same video in the same batch as continuous or discontinuous. Secondly, another classification loss to predict the location of discontinuity in the clips as well. Finally, a contrastive loss is used to learn the feature representation of the missing section in the discontinuous clips.

**Multi-modal correspondence:** In Arandjelović & Zisserman (2017), by utilizing audio-video correspondence learning as a pretext task, the authors were able to achieve performance at par with the contemporary state-of-the-art SSL methods on image classification benchmark on the ImageNet dataset.

**Temporal Pace Prediction:** Jenni et al. (2020) uses speed prediction and temporal context prediction for learning video representations.

**Statistic Prediction:** In Wang et al. (2019), both appearance and motion statistics were used for representation learning. Motion boundary, spatial-aware motion statistics, spatiotemporal colour diversity statistics, and dominant colour labels are some of the motion and appearance statistics used in the work.

### 2.1.6 Masked Image Modeling

Masked Image Modeling (MIM) is a relatively new direction of research in self-supervised learning. Although the terminology seems different, the principle is the same as contextual information learning, as in Doersch et al. (2015) and Pathak et al. (2016), where the primary task is predicting the masked portion of the image from the unmasked regions. The concept is adopted from the BERT masked language modelling framework Devlin et al. (2019) in the domain of Natural Language Processing (NLP).

**Predictive Masked Image Modelling:** Some initial works like Chen et al. (2020a), BEiT (Bao et al., 2021) introduced the concept of MIM in SSL. In BEiT (Bao et al., 2021), a ViT-based encoder is used to predict the visual tokens of the masked image patches. The tokenization uses a discrete variational auto-encoder (dVAE) (Rolfe, 2017). $mc$-BEiT (Li et al., 2022e) uses an off-the-shelf tokenizer to generate soft probabilities for tokens to incorporate the possibility that semantically similar patches may be allocated with discrepant token IDs and semantically dissimilar patches may be allocated with the same token ID due to their low-level similarities. Unlike BEiT, which uses a pre-trained tokenizer separately, iBOT (Zhou et al., 2022a) uses a momentum-updated encoder as an online tokenizer and uses it to train a target encoder using a self-distillation-based masked image modelling framework. PeCo (Dong et al., 2023) attempts to improve BEiT by replacing the dVAE-based tokenizer with a VQ-VAE (van den Oord et al., 2017) based tokenizer as a learnable perceptual visual codebook to be used in BERT-like pre-training for visual representation learning.

SimMIM (Xie et al., 2022) introduces a simple framework for Masked Image Modeling, by using raw pixel value regression as the objective function when reconstructing the original image from the masked input image. Following BERT (Devlin et al., 2019), SimMIM (Xie et al., 2022) uses a learnable mask token vector to replace each masked patch.

**Auto Encoder based MIM:** Masked Auto Encoders (MAE) (He et al., 2022) are another prime example of instance-based SSL frameworks which utilise masked image modelling. These frameworks also come under the purview of Masked Image Modeling and primarily reconstructive approaches to representation learning. Unlike BERT, MAE uses an encoder-decoder architecture with a high masking ratio of 70-80% for optimal performance. The encoder only processes a small portion of the patches, whereas the decoder processes both the input latent representations and the mask tokens to learn generalized representations.

To make MAE more efficient, LoMAR (Chen et al., 2022a) uses local masked reconstruction by sampling windows of patches from images and performs predictive reconstruction from embeddings of both masked and unmasked patches. LocalMIM (Wang et al., 2023b) uses signals from different layers in the ViT encoder to reconstruct input at multiple scales, to learn both fine- and coarse-level representations.

In MixMAE (Liu et al., 2023b) the visible tokens from two unlabeled images are mixed by using non-overlapping masking on each of the images. The encoder processes the mixed input to reconstruct both the input images, thereby saving computing resources on processing less informative mask tokens.

**Semantic Guided MAE:** SemMAE (Li et al., 2022b) uses semantic guided masking as a strategy for learning representation. However, to obtain the semantic information, it uses a separate iBOT (Zhou et al., 2022a) pre-trained encoder and a Style-GAN (Karras et al., 2021) based decoder to obtain the attention maps for different semantic regions. Similarly to SemMAE, ObjMAE (Wu & Mo, 2022) uses segmentation or CAM (Zhou et al., 2016) to select object-wise patches to learn object-wise decoupled representations.

**Feature Regression based MIM:** CAE (Chen et al., 2023c) involves combining masked representation regression and masked patch reconstruction. Another such work BootMAE (Dong et al., 2022) uses the output of a momentum-updated target encoder as a target for feature prediction in addition to pixel regression from multiple scales/levels of the online encoder.

**Knowledge distillation based MIM:** SdAE (Chen et al., 2022d) uses a similar self-distillation framework. It also uses a multifold mask strategy to reduce computational overhead in the teacher network pipeline. Another such work BootMAE (Dong et al., 2022) also uses knowledge distillation as an auxiliary task.

**Denoising MIM:** DMAE (Wu et al., 2023c) aims to learn robust representations by adding noise to the input image itself to produce corrupted images. In CIM (Fang et al., 2023b), to generate the corrupted image, a dVAE (Rolfe, 2017) and a small pre-trained BEiT (Bao et al., 2021) are used as the frozen tokenizer and generator, respectively. Following BEiT, DALL-E (Ramesh et al., 2021) tokenizer provides token targets for the small BEiT generator. The image generated by BEiT serves as input to the enhancer network.

**Combining Generative algorithms in MIM:** In GAN-MAE (Fei et al., 2023b), MAE serves as the corrupt image generator and a GAN-like discriminator is used to classify the output image from MAE as fake or real. To reduce parameters, the discriminator and MAE encoder use shared parameters, which improve efficiency. MOVE (Bielski & Favaro, 2022) also combines differentiable image inpainting using MAE and adversarial training to learn representations for object segmentation and detection.

In DiffMAE (Wei et al., 2023), denoising diffusion probabilistic model (DDPM) (Ho et al., 2020) is combined with MAE (He et al., 2022) for visual representation learning. In the forward diffusion process, the masked image is used as input, and at the end of the diffusion process, the masked patches approximate the standard Gaussian distribution. Denoised masked regions are generated in the reverse process by the decoder from the encoded visible tokens, timestep, and noisy masked regions.

**Domain Generalizing MIM:** DiMAE (Yang et al., 2022c) attempts to tackle multi-domain data by learning domain-invariant representations using an FFT-based style-mix algorithm to convert data from multiple domains into a single input and reconstructing image from each domain using a domain-specific decoder.

**Combining Instance-based learning with MIM:** Several works have attempted to combine instance-based learning with masked image modelling. Contrastive MAE (CMAE) (Huang et al., 2022b), Simple CMAE (Mishra et al., 2022b), Wang et al. (2023g), and Lu et al. (2023a) uses both masked patch/frame reconstruction to learn locally sensitive semantic representations and contrastive loss optimization to maximize similarity between different representations and also to learn the discriminative relation between different images. Similarly to CMAE, RePre (Wang et al., 2022b) also uses a contrastive objective to maximize the representational similarity between different augmented videos. However, it uses a specialized reconstruction decoder to reconstruct the masked patches from multiple hierarchy features obtained from the ViT encoder.

SplitMask (El-Nouby et al., 2021) uses a unique variation of MAE by splitting an image into non-overlapping sets of patches and reconstructing the whole image from both sets separately. SplitMask also employs contrastive loss to maximize the representational similarity between the descriptors of the two sets of patches from the same image.

Siamese image modelling (SIM) (Tao et al., 2023) minimizes pairwise similarity with all the negative tokens obtained from the target encoder as the instance discrimination task. Furthermore, SIM shows that colour augmentation, generally used in instance discrimination methods, and believed to not be beneficial for MIM as reported by MAE (He et al., 2022), can be made beneficial if used under proper training settings. SIM uses different views of the same image to effectively use colour augmentations for representation learning, as the colour variation information is leaked when used with the same view.

However, ConMIM (Yi et al., 2023) uses only the contrastive learning strategy, where a positive pair of images is obtained by using a masked and unmasked version of the same image. The unmasked image is passed through the momentum-updated encoder to form a patch-level feature look-up dictionary.

**Architectural Modifications of MAE:** ConvMAE (Gao et al., 2022) introduces a hybrid convolution-transformer encoder architecture following Co-AtNet (Dai et al., 2021b), UniFormer (Li et al., 2023d), Early Conv (Xiao et al., 2021). The convolution layers are primarily used for the high-resolution embeddings, while the transformer layers are used for the low-resolution embeddings.

ConvNeXtv2 (Woo et al., 2023) uses a sparse convolutional encoder and a convolutional block decoder as a fully convolutional MAE, inspired by the ConvNeXt (Liu et al., 2022c) architecture. Li et al. (2022d) use a two-stage masking strategy on image inputs to a pyramid-based ViT encoder for learning image representations. HiViT-MIM (Zhang et al., 2022f) uses a hierarchical ViT (Zhang et al., 2023f) architecture as an encoder for MIM, instead of vanilla ViT.

### 2.1.7 Masked Video Modeling

**Predictive strategy:** While the previous work discusses masked image modelling, applying the same on videos is tricky as it involves an additional temporal dimension. BEVT (Wang et al., 2022c) uses a VideoSwin (Liu et al., 2022b) transformer as a shared image and video encoder, but a separate decoder for image and video. Both the image and video network are jointly trained for spatial representation and temporal dynamics learning. MaskFeat (Wei et al., 2022) is another similar framework that uses masked feature prediction to learn visual features for video understanding but uses a dVAE codebook such as BEiT for tokenization.

**Autoencoder based MVM:** VideoMAE (Tong et al., 2022) is one of the foundational works that employs a method called tube masking to handle the two factors, temporal redundancy and correlation, and to learn representations from videos. VideoMAEv2 (Wang et al., 2023e) further scales VideoMAE by using a dual masking strategy to make video understanding more efficient. In addition to an encoder mask, VideoMAEv2 also uses a decoder mask following MAR (Qing et al., 2023).

Feichtenhofer et al. (2022) presents a simple extension of MAE (He et al., 2022) to video understanding, by dividing a video into a regular grid of patches in spacetime. The authors also observed that a high masking ratio of space-time agnostic masking reduces the computational complexity in the encoder.

AdaMAE (Bandara et al., 2023) uses an adaptive token sampler to select tokens from high-activity regions with higher probability compared to tokens from background or low-activity regions in videos. MaskViT (Gupta et al., 2023a) uses VQ-GAN (Esser et al., 2021) to encode video frames into discrete latent codes. Given a frame, the model learns to predict masked tokens in future frames, using spatial and spatiotemporal attention blocks in each layer, which also reduces the memory requirements. OmniMAE (Girdhar et al., 2023) aims to learn a single model for both image and video using an omnivorous network (Girdhar et al., 2022), and treating images as a single frame video.

## 2.2 Clustering-based Pretext Tasks

*k-Means*: Coates & Ng (2012) presented one of the first pre-training approaches using *k*-means algorithm (MacQueen, 1967) to learn patchwise feature dictionaries. DeepCluster (Caron et al., 2018) utilizes *k*-means clustering algorithm to generate pseudo-labels from features extracted by the convolutional neural networks. These pseudolabels are then used for the cross-entropy loss-based classification task for representation learning. DeeperCluster (Caron et al., 2019) combines context-based pretext task with *k*-means-based clustering step in DeepCluster for better pre-training. UIC (Chen et al., 2020d) improves DeepCluster by using the 1-iteration embedding clustering step, making it scalable for larger datasets. An online version of DeepCluster was presented in ODC (Zhan et al., 2020) where pseudo-labels evolve along with the parameters, preventing rapid change to the pseudo-labels, with two separate memory banks for samples and centroids and without the requirement for an extra feature extraction step. The principle of DeepCluster (Caron et al., 2018) was also applied to multimodal data in XDC (Alwassel et al., 2020).

*Agglomerative Clustering*: JULE (Yang et al., 2016) proposed a recurrent network-based unsupervised framework that combines agglomerative clustering for pseudolabel generation and subsequent classification.

*Using Online k-means*: CLIM (Li et al., 2020b) combines $k$-means clustering with kNN for positive sample selection. CoKe (Qian et al., 2021a) utilises an online constrained $k$-means algorithm to compute pseudo-labels and cluster centres to capture the global distribution of the data.

*Autoencoder based clustering*: DEC (Xie et al., 2016) instead uses an autoencoder for parameter initialization and a KL-divergence based clustering and parameter optimization step. IDEC (Guo et al., 2017) incorporates the auto-encoder in the DEC framework itself to preserve the local embedding structure. SDMVC (Xu et al., 2023a) further improves IDEC to multiple views by using an autoencoder for each view and clustering all views for global discriminative feature learning.

*Mutual Information maximization*: RIM (Krause et al., 2010) improves Bridle et al. (1991) by using a regularizer to prevent cluster fragmentation and complex cluster boundaries. IMSAT (Hu et al., 2017) uses RIM (Krause et al., 2010) for the clustering step and then uses an information maximization step based on cross-entropy.

## 2.3 Paired Embedding Based Pretext Tasks

While initial self-supervised learning frameworks based on context encoding, transformation prediction, and jigsaw puzzle solving laid the foundation of SSL in the early years of its development, researchers soon realized the limits of such methods. It was the need of the time which led to the emergence of a new class of frameworks. These frameworks use information from paired embeddings to optimize a specific objective or loss function and learn optimal parameters in the process. We can categorize these frameworks primarily into two categories: (1) Contrastive and (2) Non-contrastive. We will discuss several foundational and current state-of-the-art frameworks in both categories below.

### 2.3.1 Contrastive Learning Frameworks

Contrastive learning can be considered as learning by comparing different samples. The primary objective of contrastive learning frameworks is to discriminate between dissimilar samples in pairs termed as negative pairs, and closely map similar samples in pairs termed as positive pairs. Triplet loss-based contrastive learning frameworks have been around for a long time, and used in works like face detection (Chopra et al., 2005; Schroff et al., 2015), metric learning (Weinberger & Saul, 2009), etc. However, the use of contrastive loss in an unsupervised setting was observed in early works like Sermanet et al. (2017); Hyvärinen & Morioka (2016). In TCN (Sermanet et al., 2017), the authors used a triplet loss-based contrastive loss for self-supervised representation learning from multiview videos. Different views of an action or event at the same time step constituted the positive pair, and the embeddings of the samples in the positive pair were trained to be located close to the embedding space. On the other hand, clips or images from different time steps formed the negative pairs and were trained to be distant from each other. The representations were transferred for imitation learning in self-supervised robotics. In Fig. 4, we illustrate the abstract representation of some foundational contrastive frameworks which serve as the baseline for the current advances in self-supervised learning.

**Non-linear Autoregressive Frameworks**
The paradigm of contrastive self-supervised learning (SSCL) frameworks received a huge boost in performance with the advent of CPC (van den Oord et al., 2018), primarily based on the principle of noise contrastive estimation (NCE) (Gutmann & Hyvärinen, 2012). For vision tasks, CPC uses an autoregressive style predictive framework. Images are first divided into patches and encoded. To predict each patch, the patches preceding it were used and the encoder was optimized using InfoNCE loss. The primary function of optimizing the InfoNCE loss was shown to maximize the mutual information between the input and the encoded representations. CPC described the InfoNCE loss as the categorical cross-entropy of classifying the positive sample correctly.

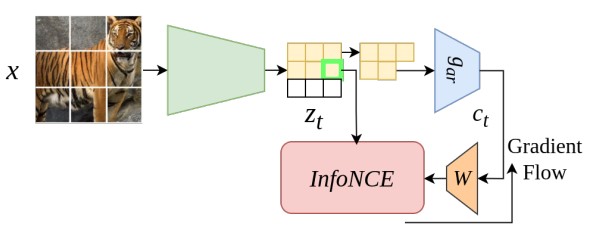

(a) CPC (Non-linear autoregressive) Framework

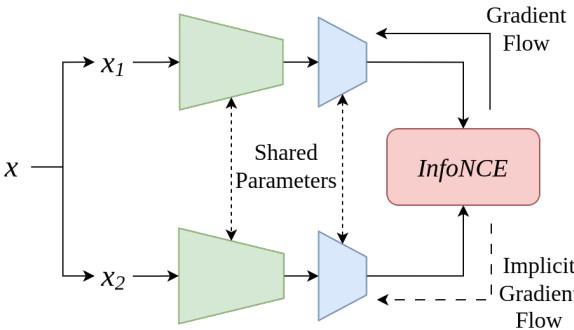

(b) SimCLR (Instance discrimination based) Framework

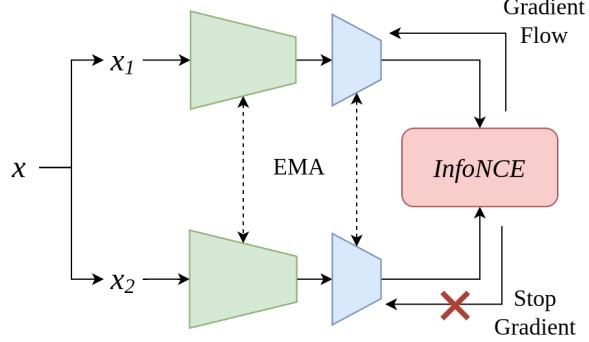

(c) MoCo (Knowledge distillation based instance discrimination) Framework

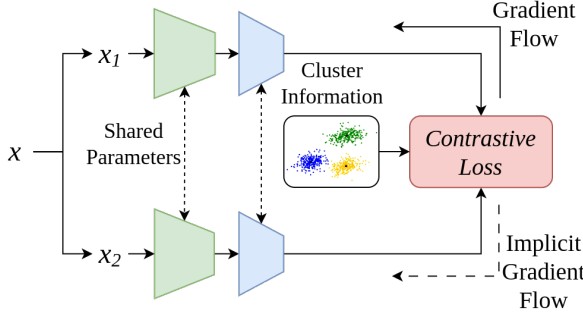

(d) SwAV (Instance-Group discrimination) Framework

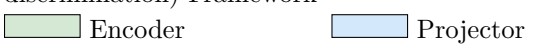

Figure 4: Illustration of Contrastive Frameworks. "Implicit gradient flow" means the gradient flow is not restricted for the second views ($x_2$) of the sample $x$ in frameworks like SimCLR and SwAV. In MoCo, the second view $x_2$ is passed through the momentum updated encoder, hence, no gradient flows through the same, which is represented by "Stop gradient". "EMA" denotes Exponential moving average.

The principle of CPC (van den Oord et al., 2018) was later used again in several works such as CPCv2 (Hénaff et al., 2020) and ACPC (Chorowski et al., 2021). CPCv2 applies several modifications to CPC, which include increasing model capacity, applying layer normalization, using more context information for predicting patch embeddings, and patch-based augmentations. Harley et al. (2020) also adopt a non-probabilistic version of CPC for both top-down and bottom-up representations in ego-motion stabilized videos. CPC was further improved in RPC (Tsai et al., 2021b), where the authors aimed to solve three primary issues in contrastive learning, training stability, sensitivity to minibatch, and downstream performance by eliminating the logarithm of contrastive loss and using an additional $l_2$-regularization. RPC can be associated with Chi-squared divergence. DPC (Han et al., 2019) used the principle of CPC for learning video representation.

**Instance Discrimination Frameworks**

The principle of instance discrimination frameworks is primarily based on the idea of contrasting samples. Representations of similar samples are drawn closer, while representations from dissimilar samples are pushed away. This principle prevents the collapse of representations. Instance discrimination-based frameworks can be further divided into several sub-categories (as described below), depending on the approach used for contrasting the sample instances. In the following subsections, we explore the evolution of instance discrimination-based SSL frameworks, along with the characteristics of each type of such framework.

**Instance-Instance Discrimination:** A different perspective on instance-instance discrimination was presented much before CPC in the work Dosovitskiy et al. (2014), where sampled patches from a randomly sampled number of images and applied random transformations on the images. A class label is assigned to the set of the transformed version of each image. The pretext task is simply a multiclass classification task, which involves learning to classify the images of the set of images.

*Dawn of Instance-Instance Discrimination*: Concurrently with CPC (van den Oord et al., 2018), Wu et al. (2018), proposed a novel instance discrimination framework based on the same principle of NCE. Wu et al. (2018) treated each instance as a separate class of its own and maintained a memory bank to construct a non-parametric softmax classifier for self-supervised representation learning. This work laid the foundation for several SSCL frameworks for vision tasks.

Similarly to Wu et al. (2018), the basic framework of LA (Zhuang et al., 2019a) is based primarily on a clustering step to identify close and background neighbours from a momentum-updated memory bank and then apply a local aggregation metric based on contrast loss. PIRL (Misra & van der Maaten, 2019) uses a different formulation of contrastive loss following Hadsell et al. (2006). The final objective function in PIRL is a convex combination of two contrastive losses, one that maximizes similarity between an image and its augmented or transformed self, and the other that maximizes the dissimilarity between an image and the other images in the dataset. To implement the second part, PIRL follows a similar strategy of using a memory bank as Wu et al. (2018) and LA (Zhuang et al., 2019a). In fact, the formulation of Wu et al. (2018) is a special case of PIRL.

*Large Batch Instance Discrimination*: The first major SSL framework to not use a memory bank for the formation of negative pairs was SimCLR (Chen et al., 2020b). SimCLR mitigated the requirement for a memory bank by using a large batch size and showed that increasing the number of negative pairs improves performance. It also emphasized the role of augmentation and non-linear projectors in the quality of representations in SSL.

SimCLRv2 (Chen et al., 2020c) further found that increasing the number of parameters results in better representation learning for semi-supervised and fine-tuning performance, if the labels are fewer. The popularity of SimCLR is evident from its application of SimCLR as the foundation model for several works like sound classification (Fonseca et al., 2021), online distillation (Bhat et al., 2021), learning with noisy labels (Zheltonozhskii et al., 2022), etc. FN (Huynh et al., 2022) attempts to improve the performance of the SimCLR baseline by eliminating the influence of false negative pairs. The basic framework of SimCLR has also been adopted in contrastive video representation learning frameworks such as CoCLR (Han et al., 2020), TCLR (Dave et al., 2021), CVRL (Qian et al., 2021b), Huang et al. (2021b), MaCLR (Xiao et al., 2022b), SCVRL (Dorkenwald et al., 2022), HSSL (Xiao et al., 2022a), FAME (Ding et al., 2022) and HDC (Zhang & Crandall, 2022).

Another work presents MoCLR and DnC (Tian et al., 2021a), which presents an improved version of SimCLR. It uses a mixture of experts trained on each superset and a base model trained on the whole data, for knowledge distillation to the final model. It yields considerable performance improvement but at the cost of large training datasets, pre-training encoder, and a large number of pretraining epochs. The strategy used in MoCLR is also used in MoBY (Xie et al., 2021).

InfoMin (Tian et al., 2020b) empirically inferred that the augmentations used in contrastive learning are downstream task-dependent and found a U-shaped relationship between an estimate of mutual information and the quality of the representation in a variety of settings.

*Positive Samples Augmented Instance Discrimination*: NNCLR (Dwibedi et al., 2021) further incorporates multiple positive instances in SimCLR (Chen et al., 2020b) by sampling various positive samples from the neighbourhood of each instance in the sample manifold represented by a support set queue as in MoCov1 (He et al., 2020) or MoCov2 (Chen et al., 2020e). Similar to NNCLR, SNCLR (Ge et al., 2023) also samples positive samples from a neighbour support set using the nearest neighbour algorithm. However, SNCLR employs an additional weight calculation step to compute the correlation of the sampled neighbours with the instances which is then used in the N-pair contrastive loss.

*Positive-Negative Decoupling*: In DCL (Yeh et al., 2022), the authors identified a negative-positive-coupling effect in InfoNCE loss which proved harmful to learning efficiency in contrastive frameworks. DCL proposed to remove this effect by eliminating the positive term from the denominator in InfoNCE loss, resulting in a significant improvement in performance without the requirement of large batch size, such as in SimCLR (Chen et al., 2020b) or momentum encoding in MoCo (He et al., 2020).

*Contrastive Learning without InfoNCE*: SSL-HSIC (Li et al., 2021d) examines contrastive learning from a statistical dependence point of view. This work proves that InfoNCE approximates SSL-HSIC with a variance-based regularization and proposes a loss inspired by the HSIC (Hilbert-Schmidt Independence Criterion) bottleneck. For computing HSIC, SSL-HSIC uses an estimator provided by Gretton et al. (2005).

UniGrad (Tao et al., 2021) presents an unifying framework using gradient analysis showing that different SSL frameworks optimize in a similar mechanism. UniGrad proposes a gradient-based framework, which has the primary aim of maximizing the cosine similarity between positive samples and the similarity between negative samples close to zero and obtains considerably comparative performance with the contemporary state-of-the-art SSL frameworks. TiCo (Zhu et al., 2022) introduced a novel contrastive framework by using a squared contrastive loss instead of the widely used InfoNCE loss. The primary difference between the loss used in TiCo and InfoNCE is the uniformity (Wang & Isola, 2020) or denominator term. Optimizing the TiCo loss also makes contrastive learning easier, as it encourages the representations of negative samples to be orthogonal.

A concurrent work by HaoChen et al. (2021) explores spectral contrastive learning, where the authors use the population augmentation graph to effectively partition the same into sub-graphs, which are representative of fine-grained sub-classes of the actual classes in the downstream task. In reality, this work uses a learnable spectral decomposition component of the embeddings to learn the most important eigenvectors to maximise linear probe performance.

*Triplet loss based Instance discrimination*: Wang et al. (2021b) uses a truncated triplet loss to deal with the under-clustering and over-clustering issues in contrastive learning frameworks.

Trip-ROMA (Li et al., 2023f) presents a novel approach by combining triplet loss with binary cross-entropy for contrastive SSL. It also presents a randomness-based similarity measure which can be used with foundation models like SimCLR to prevent overfitting or shortcuts as studied in Robinson et al. (2021a).

*Metrics in Contrastive SSL:* Although there have been many improvements in downstream performance with the advent of contrastive learning algorithms, there was no metric in the literature except linear probing or kNN accuracy, which could be used to measure the quality of representations. Wang & Isola (2020) analyzed self-supervised representation learning using two metrics, alignment and uniformity and studied the relationship of the two metrics with the downstream performance. Furthermore, it also presented several illustrations to understand representation learning on the hyper-sphere. An intriguingly similar work was also presented in Ye et al. (2019). Moon et al. (2022) empirically discovered that the alignment and uniformity are directly correlated with the downstream performance of both instance-level and dense-level downstream tasks.

**Dense Contrastive Learning:** DenseCL (Wang et al., 2020) introduces a pixel-wise dense contrastive learning framework for dense representation learning, specifically for tasks like semantic or instance segmentation, depth estimation, etc. It uses cosine similarity-based correspondence matching for the formation of pixel-level pairs. The final loss is a convex combination of the loss at the image level and the loss at the pixel level of InfoNCE. VADeR (Pinheiro et al., 2020) also presents a dense representation learning framework using an encoder-decoder architecture for computing pixel-level representations. Around the same time, PixPro (Xie et al., 2020b) proposed a pixel-based contrastive learning framework. PixPro used a threshold to identify similar feature map pixels and then used an N-pair contrastive loss to maximize the similarity between their representations. SetSim (Wang et al., 2021d) tries to improve DenseCL by finding the correspondence set of features for the query feature vectors from the queue, as well as, pixel-level features from the feature attention maps. Similar to DenseCL, CACL (Pang et al., 2022a) uses a cycle loss to maximize the probability of getting a sample back as the positive sample of its neighbour, in addition to a global and pixel contrastive loss.

Generally, vision-based contrastive learning algorithms are trained on images with single instances. To scale the same for images with multiple instances as in the PASCAL VOC (Everingham et al., 2010) or COCO (Lin et al., 2014) datasets, SpaceCL (Song et al., 2022) proposes a space correlation module to effectively model local semantic similarity and space-correlated similarity in overlapping crops from the same image.

Recent works like Shen et al. (2023) use asymmetric masking to generate positive samples. The authors use stop-gradient like SimSiam (Chen & He, 2020) to prevent the collapse of representations with InfoNCE as the loss function.

**Multi-view frameworks:** CMC (Tian et al., 2020a) extends this to more than two views using a self-supervised version of N-pair loss objective (Sohn, 2016). In addition to the modified objective, CMC also uses a memory bank similar to the above frameworks. Another major contribution of CMC is the introduction of the ImageNet100 dataset in the SSL domain, which is a subset of the 100 class of the original ImageNet1K dataset (Deng et al., 2009). Similar to CMC, MIL-NCE (Miech et al., 2020) uses multiple positive pairs for multi-modal representation learning. However, the objective function differs from CMC. The generation of multiple spurious positive samples was explored by using CutMix augmentation (Yun et al., 2019) in BSIM (Chu et al., 2020). ECPP (Kim et al., 2024) improves CMC by increasing the number of augmentations using crop-only transforms of smaller dimensions like SwAV (Caron et al., 2020), and also discarding the augmented versions from the list of negatives in the denominator of the contrastive loss term.

In a recent work Hu et al. (2024), the authors use a combination of different modules for multi-view self-supervised learning. The proposed framework uses a pseudo-label guided positive pair sampling step to inhibit the effect of false negative pairs in contrastive learning, both feature and cluster correlation maximization to improve feature alignment, and finally, a divergence minimization based clustering loss to further enforce compactness and separability.

**Information Maximization:** One of the oldest foundational works on SSL is Bridle et al. (1991). In this work, the output is used as the probability distribution over the class label, a discrete random variable. The objective is to maximize the difference between the entropy of the average of the outputs (referred to as fairness) and the average of the entropy of the outputs (referred to as firmness). Maximizing the entropy of the average of the outputs prevents the dimensional collapse, while minimizing the average of the entropy of the outputs prevents collapse of the representation to a single point in the latent space. This principle is the backbone of all self-supervised learning frameworks.

AMDIM (Bachman et al., 2019) is one significant work contemporary to Wu et al. (2018), LA (Zhuang et al., 2019a), PIRL (Misra & van der Maaten, 2019) and MoCo (He et al., 2020) but did not use a memory bank. In AMDIM (Bachman et al., 2019), the authors presented a self-supervised version of Deep InfoMax (Hjelm et al., 2019) and expanded the architecture to incorporate multiscale features. This innovation allowed them to maximize the mutual information between samples in positive pairs efficiently by increasing the set of samples as a whole. The framework of AMDIM was also used for video representation learning in Hjelm & Bachman (2020).

Tschannen et al. (2020) explored the role of mutual information (MI) estimators in contemporary self-supervised learning frameworks such as CPC, CMC, and AMDIM. Firstly, it showed that MI and downstream performance are loosely connected, and maximizing MI is not necessary to learn good representations.

**Distribution Divergence Minimization as Regularization:** ReLIC (Mitrovic et al., 2021) explores SSL from a causal perspective with content and style as latent variables. This work argues that the conditional distribution of the class representations given the content should remain invariant under style changes. For this purpose, the authors used the KL divergence as a regularizer along with contrastive loss. von Kügelgen et al. (2021) tries to understand the success of data augmentations in enhancing pre-training performance from a theoretical perspective by treating content and style as latent variables.

**Adversarial Contrastive Learning:** AdCo (Hu et al., 2020a) uses learnable adversarial negatives to optimize the parameters using contrastive loss. AdCo formulates the problem as a minimax game between optimizing the parameters and negative adversaries. RoCL (Kim et al., 2020a), ACL (Jiang et al., 2020), Ho & Vasconcelos (2020), AdvCL (Fan et al., 2021), DecACL (Zhang et al., 2022a), SCAT (Wu & Yeung,

2023) are other works that aim to improve representation consistency by using adversarial training with the SimCLR framework as the baseline.

**Knowlege Distillation based Instance Discrimination Frameworks:** Another significant work using a memory bank for negative sample mining was presented in MoCo (He et al., 2020). However, the queueing procedure in MoCo differs from the contrastive frameworks mentioned above. Instead of storing a representation for each sample in the dataset, MoCo uses a momentum-updated encoder to extract representations to store in the fixed-size memory bank. All the representations in the memory bank act as negative samples and contribute to forming a negative pair with a sample, whereas a positive pair is obtained by pairing two differently augmented versions of the sample. This extends the idea of knowledge distillation but does not use the predicted labels by a teacher model to supervise a student model. Instead, it used the distribution of the likelihoods of an augmented version of the query to supervise the retrieval of another version of the query from a pool of representations. LSTCL (Wang et al., 2022a) also uses MoCo as a baseline framework for video representation learning in addition to BYOL and SimSiam. DisCo (Gao et al., 2021) distils knowledge from a large self-supervised pre-trained model to a student encoder. The student encoder is used as the target encoder in the MoCov2 framework to train a mean student encoder, used for the downstream tasks. MiCE (Tsai et al., 2021a) uses MoCo as the baseline framework. Using a mixture of experts comprised of a teacher and student network as in MoCo, and a gating function to channel the input to the appropriate experts, MiCE improves downstream performance by a considerable margin.

MoCov2 (Chen et al., 2020e) attempted to further improve MoCo (He et al., 2020) by incorporating two design properties of SimCLR, that is, non-linear projection head, and stronger data augmentation. Another attempt to further scale up MoCo (He et al., 2020) was presented in MoCov3 (Chen et al., 2021d). MoCov3 ditched the memory bank as it showed minimal gain when used with a large batch size. In MoCov3, an additional prediction head was also used in the online encoder, following BYOL (Grill et al., 2020). In addition to the aforementioned contributions, MoCov3 presented an exhaustive analysis to train vision transformers (Dosovitskiy et al., 2021) self-supervised. Zhu et al. (2021) proposes a simple yet effective feature transformation, which creates both hard positives and diversified negatives to improve training with MoCov2 as the baseline framework. VideoMoCo (Pan et al., 2021) has presented an extension of the MoCo framework to videos, where the authors used temporal adversarial learning to augment videos. Similar other works which use MoCo as the baseline framework for video representation learning are SeCo (Yao et al., 2020), VINCE (Gordon et al., 2020), Wu & Wang (2021), VCLR (Kuang et al., 2021). VCLR uses SeCo as the baseline framework and scales it to perform contrastive learning at the video level. Using MoCov2 as the baseline framework, OCL (Gallardo et al., 2021) found that self-supervised pre-training improves online continual training due to the generalization ability of SSL methods.

In one concurrent work JCL (Cai et al., 2020), instead of penalizing the individual positive sample when paired with the query, multiple positives are simultaneously paired and penalized by using a Gaussian approximation of the neighbourhood of the query. In another work, Zhao et al. (2022) used only MoCo with Swin transformers for representation learning.

CAST (Selvaraju et al., 2021) uses Deep-USPS (Nguyen et al., 2019) to identify salient regions and a constrained cropping augmentation method to avoid the inclusion of noisy background regions. Using MoCo (He et al., 2020) as the baseline, CAST uses an additional attention loss to base predictions on correct regions, as contrastive methods often use wrong regions to match query and key images.

**Distribution Divergence Minimization as Regularization:** CLSA (Wang & Qi, 2021) uses distribution divergence minimization using representations from a memory bank, in addition to contrastive loss for better representation learning with MoCov2 as baseline. CO2 (Wei et al., 2021) also uses the MoCov2 framework as a baseline and uses a KL divergence-based consistency regularization loss in addition to contrastive loss and also improves MoCov2 by a considerable margin. A similar approach is also explored in MCL (Yang et al., 2022a) and Yang et al. (2023a).

**Mitigating Sampling Bias:** One noteworthy characteristic of contrastive SSL is that samples with the same label were paired into negative pairs. Debiased CL (Chuang et al., 2020) attempted to remove the negative sampling bias from the available positive sample pairs only. A similar approach to reducing sampling bias's influence in contrastive learning for sentence representations was also presented in Zhou et al. (2022b).

In a similar theoretical approach to Chuang et al. (2020), Robinson et al. (2021b) proposed to utilise hard negative samples to improve the generalization of self-supervised learning frameworks and improve downstream performance. MoCHi (Kalantidis et al., 2020) synthesizes hard negatives by a convex combination of negative samples during the start of training, and harder negatives by mixing the query with negative samples as the samples begin to separate from each other. A similar approach for use on medical images was presented in Zhao & Zhou (2022). Another work which uses negative sampling for contrastive learning of visual representations is CNCE (Wu et al., 2021a). HSA (Xu et al., 2020) uses positive sample mining using the k-nearest neighbour algorithm from hierarchical features extracted from the encoder.

**Instance-Group Discrimination:** Clustering-based pretext tasks aim to adapt clustering algorithms for the generation of pseudo-labels for end-to-end training of visual features.

**Distance-based clustering:** In G-SimCLR (Chakraborty et al., 2020), the authors used an autoencoder to generate representations for a $k$-means clustering step to obtain pseudo-labels from each batch before applying contrastive loss. TCC (Shen et al., 2021) uses both cluster and instance-level contrastive learning like most of its concurrent counterparts. SCCL (Zhang et al., 2021a) is another work which uses both contrastive and KL-divergence based cluster assignment loss for representation learning.

IIC (Ji et al., 2019) simply utilises mutual information to learn representations from data in an unsupervised manner, discarding instance-specific details and also avoiding degenerate solutions due to the individual cluster assignment entropy maximization which prevents assignment of all samples to a single cluster.

In CLD (Wang et al., 2021c), the authors add a spherical $k$-means based feature grouping step to apply instance-group discrimination, to reduce the effect of instance-instance discrimination between similar samples. Concurrently to CLD, PCL (Li et al., 2021b) formulated the proposed framework as an Expectation-Maximization algorithm. After clustering the features from the momentum encoder, negative sample prototypes are sampled, and finally, the NCE loss is optimized. ProPos (Huang et al., 2021c) optimizes the MSE loss between a sample and its Gaussian distributed positive neighbours, in addition to the objective proposed in PCL.

In another recent work, MUGS (Zhou et al., 2022c) uses three levels of granularity for representation learning, instance level, local group level, and global group level. For local-group-level discrimination, it uses averaged out historical tokens stored in a memory queue as neighbors. For the global-level, it builds a set of learnable group prototypes for instance-group discrimination.

**Optimal Transport based pre-training:** Simply stated, clustering requires assigning samples to each of the clusters. When a uniformity condition is applied to the assignment problem, it can be treated as an optimal transport problem.

In SeLa (Asano et al., 2020b), the first step is the same as the previous clustering-based pretext task, that is, cluster-based pseudo-label assignment and cross-entropy-based optimization. The second step involves *Sinkhorn-Knopp* algorithm-based transport polytope computation (Cuturi, 2013). SwAV (Caron et al., 2020) uses clustering to compute prototypes which are used to predict codes of one view from another using *Sinkhorn-Knopp* algorithm (Cuturi, 2013). SwAV is the first SSL framework to surpass ImageNet supervised features on COCO (Lin et al., 2014), Places205 (Zhou et al., 2014), and VOC07 datasets (Caron et al., 2020). A similar approach was also adopted in ViCC (Toering et al., 2021) for self-supervised video representation learning. SMoG (Pang et al., 2022b) further improves SwAV by adding a group-level discrimination branch to it. MIRA (Lee et al., 2022) also improves SwAV by not using the equipartition constraint, rather it constraints the marginal entropy by mutual information regularization. Another recent work CDC (Ding et al., 2023) basically combines MoCo with SwAV, but the cosine similarities are computed between feature vectors expressed probabilities over the clusters. CVCL (Chen et al., 2023a) presents a SwAV-like framework (discussed below) for learning representations by using multi-view samples.

SEER (Goyal et al., 2022) explores the challenges of scaling the pre-training architectures using SwAV as the baseline framework and also addresses some of the engineering challenges and complexity of training at this scale.

Instead of using $k$-means for generating pseudo-labels like in DeepCluster, (Caron et al., 2018), ODC (Zhan et al., 2020) or CoKe (Qian et al., 2021a), SCAN (Van Gansbeke et al., 2020) first uses a pretext task to learn representations and then obtains the clusters using neighbour sampling with the prior knowledge of the number of classes in the dataset. The parameters are then fine-tuned again using cross-entropy loss in a similar fashion to SeLa (Asano et al., 2020b). Whereas Amrani et al. (2022) uses the number of classes similar to SCAN to directly train an end-to-end classifier without labels by optimizing for same-class prediction of two augmented views of the same sample, under the condition of uniform distribution over the classes similar to SeLa or SwAV.

Similar to SeLa and SwAV, MDRA (Cheng et al., 2023) also uses optimal transport for relationship alignment, which is another term used to assign samples to prototypes. However, each feature vector is decomposed into subgroups along the feature dimension, and the procedure of relationship alignment is applied to each subgroup.

One fundamental shortcoming of SeLa, SwAV, and SMoG is the assumption of a uniform distribution over the prototypes. However, using a uniform distribution ensures the possibility of converging to degenerate cases, but ignores that real-life datasets are skewed. SeLaVi (Asano et al., 2020a) presents a solution to this using a permutation matrix in the energy equation of the Sinkhorn-Knopp algorithm, which sorts the prototype entropies.

Later, in GOCA (Coskun et al., 2022), the optimal assignment of one mode is used as the optimal assignment prior for the other mode, and vice versa to combine the two information sources, for multimodal video representation learning.

ConCURL (Deshmukh et al., 2022) combines SwAV (Caron et al., 2020) with instance discrimination (Wu et al., 2018) for consensus based clustering.

**Contrastive + Predictive**: Tsai et al. (2021c) provides an information-theoretic perspective to understand the properties of SSL and also provides a combined framework of contrastive and predictive learning objectives. In addition to that, it also uses an inverse predictive learning step to discard task-irrelevant information. In PrCL (Li et al., 2023e) also, the authors combined image inpainting, a context-based predictive task, with contrastive learning to improve the quality of representations in pre-training.

**Multi-modal contrastive learning**:

AVTS (Korbar et al., 2018) uses triplet contrastive loss instead of InfoNCE loss, with distance-based self-supervised synchronization between video and audio modalities. CBT (Sun et al., 2019) uses the principle of masked language modelling on videos and paired textual information separately, as well as cross-modal contrastive learning to maximize mutual information between visual and textual modes of information. Although ELo (Piergiovanni et al., 2020) does not use a single loss, we can categorize it under this subsection, as it uses a multimodal contrastive loss for video representation learning.

Multi-modal contrastive works discussed already in the previous sections are L3 (Arandjelović & Zisserman, 2017), XDC (Alwassel et al., 2020). AVID (Morgado et al., 2021) uses cross-modal contrastive learning and within-modal positive discrimination using a sampled positive and negative set. ACC (Ma et al., 2021a) uses cross-modal contrastive learning with MoCo as the baseline, but the output from the momentum updated encoder of the other modality is used as the target representation. MVCGC (Huo et al., 2021) uses positives sampled from the pool of samples of the other modality as hard positives and also incorporates cross-modal information by optimizing N-pair contrastive loss. Ma et al. (2021b) also uses the same loss. However, the authors factorize the feature space into a spatially local/temporally global (S-local/T-global) subspace and a spatially global/temporally local (S-global/T-local) subspace and define two N-pair cross-modal contrastive objectives in each of the subspaces. Unlike AVID, STiCA (Patrick et al., 2021) uses cross-modal and intra-modal contrastive loss for multi-modal representation learning.

MCN (Chen et al., 2021a) combines contrastive learning with reconstruction and clustering loss for multi-modal (video, audio, text) representation learning. Similarly to MCN, Yuan et al. (2021) explore SSL for multimodal representation learning with MoCov2 as the baseline framework. VATT (Akbari et al., 2021)

uses MIL-NCE (Miech et al., 2020) to learn representation from video and text, and an additional NCE loss for video and audio.

FNACL (Sun et al., 2023) uses false negative suppression and true negative enhancement in the contrastive learning framework for sound source localization tasks. One of the most recent works Xuan et al. (2024) also utilises multi-modal contrastive learning for sound source localization in videos.

### 2.3.2 Non-Contrastive Frameworks

Non-contrastive frameworks can be defined simply as those frameworks which do not explicitly use contrastive loss for self-supervised pre-training. Primarily, these frameworks discard the negative pairs and only use the positive pairs in the self-supervised pretraining phase while using a symmetric or asymmetric network architecture. One of the first non-contrastive frameworks can be traced back to DeSa (1993) and DeSa (1994), where the primary objective is to minimize the disagreement between the information from two different modalities. An innovative yet simple approach of using uniformly sampled noise from $l_2$ unit sphere as fixed target representations to avoid collapse in self-supervised learning was presented in NAT (Bojanowski & Joulin, 2017).

In the following discussion, we provide a comprehensive account of the non-contrastive self-supervised learning frameworks, divided into subcategories, such as, implicit variance regularization or self-distillation frameworks. These frameworks primarily employ architectural modifications like predictor or gradient update modifications like stop-gradient to learn representations and prevent collapse without the need for explicit negative pairs. In Fig. 5, we present the abstract illustration of some foundational non-contrastive frameworks for better understanding.

**Implicit Variance Regularization Framework**
   The requirement of a large number of negative samples for instance discrimination-based contrastive learning led researchers to invent negative-free contrastive learning methods. However, to keep the samples from collapsing, that is, keeping the negative samples far from each other, it was necessary to have access to the statistics of the same. Therefore, instead of peeking into the batch dimension, researchers used information available along the embedding dimensions. This allowed these new frameworks to use the information from negative samples without explicitly contrasting with the same. Regulating the variance along the embedding dimensions, allows the information to be distributed over the embedding dimensions, preventing dimensional collapse. The previous statement is supported by findings in Tian et al. (2021b), where the predictor used in works like BYOL (Grill et al., 2020) or SimSiam (Chen & He, 2020) is said to behave as a whitening transform preventing dimensional collapse. Furthermore, Halvagal et al. (2023) also emphasises the role of predictor and stop-gradient in preventing collapse and through eigenspace analysis of the predictor, shows that both BYOL and SimSiam perform implicit variance regularization for asymmetric euclidean and cosine losses.

**Knowledge Distillation-based Frameworks:**

In recent years, one of the groundbreaking and foundational works in the negative-free contrastive or dimension contrastive learning literature has been presented in BYOL (Grill et al., 2020). BYOL used a student (online) - teacher (target) network architecture as in knowledge distillation, but the teacher (target) network learns from the past iterations of the student (online) network. The objective was to maximize the similarity between the representations predicted by the online network and the representations of the target network. The predictor MLP is essential to prevent the collapse of representations in BYOL. Initially, it was hypothesized that the collapse of representations was prevented because of the batch normalization (BN) layers used in BYOL and that the BN induced an implicit contrastive effect on the embedding representations. However, these hypotheses were rejected in Richemond et al. (2020). Although BYOL does not satisfy the criteria of dimension contrastive frameworks exactly, however by enabling the information to be distributed uniformly over the embedding dimensions using the BN layers in BYOL projector MLP, the prevention of dimensional collapse is achieved, which is the primary principle of dimension contrastive frameworks. Alternatively, Tian et al. (2021b) states that the representational dynamics are decoupled for Euclidean losses as in BYOL, and converge to finite eigenvalues in the predictor's eigenspace. As the eigenvalues correspond

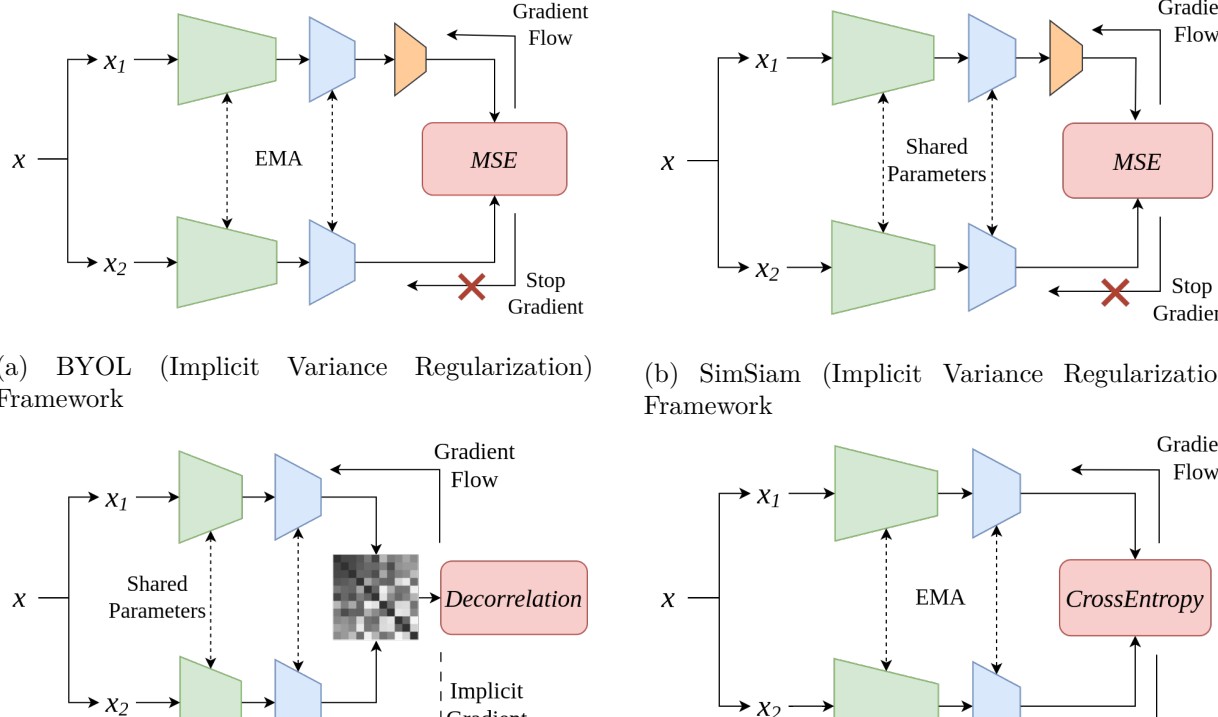

(a) BYOL (Implicit Variance Regularization) Framework

(b) SimSiam (Implicit Variance Regularization) Framework

(c) Barlow Twins (Decorrelation based) Framework

(d) DINO (Self-distillation) Framework

Figure 5: Illustration of Non-contrastive Frameworks. "Implicit gradient flow" means the gradient flow is not restricted for the second views ($x_2$) of the sample $x$ in frameworks like SimCLR and SwAV. In MoCo, the second view $x_2$ is passed through the momentum updated encoder, hence, no gradient flows through the same, which is represented by "Stop gradient". "EMA" denotes Exponential moving average.

to the variance of the representations, the underlying mechanism constitutes an implicit form of variance regularization.

ASCNet (Huang et al., 2021a) uses BYOL as the baseline framework for video representation learning using both appearance and speed consistency as the objective. FlowE (Xiong et al., 2021a) also uses BYOL to predict the representations of another frame from one frame after applying the flow transformation.

In another recent work, MYOW (Azabou et al., 2021) combines a distance loss between a sample $x$ and mined samples from the latent neighbourhood of the sample $x$, with the objective used in BYOL (Grill et al., 2020). MSF (Koohpayegani et al., 2021) added a positive sampling step from a large memory bank to BYOL to improve consistency regularization in BYOL. CMSF (Tejankar et al., 2021; Navaneet et al., 2022) improves MSF by utilising different sources of knowledge like multi-modal embeddings to constrain the nearest neighbour search space.

*Siamese Frameworks:* SimSiam (Chen & He, 2020) uses an architecture similar to BYOL with an encoder, projector and predictor, but it does not use a momentum-updated encoder. Instead, SimSiam uses a stop gradient to prevent collapse. This modifies the objective into an alternating optimization problem. Halvagal et al. (2023) show that SimSiam performs an implicit variance regularization when using the stop-gradient, without which the eigenvalues in the predictor's eigenspace diverge or collapse of representations occurs. However, the authors show that without a stop gradient, the collapse of representations occurs, which is also observed in BYOL without a momentum encoder. As mentioned in SimSiam, BYOL arXiv

v3 included a version of BYOL without a momentum encoder, but the learning rate of the predictor was increased 10×. BraVe (Recasens et al., 2021) also uses SimSiam as the baseline framework but for learning multimodal representation from videos. DenseSiam (Zhang et al., 2022e) adds a dense pixel-wise similarity loss and a region-based contrastive loss to SimSiam for dense representation learning.

*Joint Embedding Predictive Architectures (JEPAs)*:

I-JEPA (Assran et al., 2023a) bridges two sub-domains, MAEs and negative-free contrastive learning or dimension contrastive learning. The primary principle of I-JEPA is based on the principle of predicting the embedding of the masked regions, where the target representations are obtained from a momentum-updated target network. In principle, I-JEPA is similar to BYOL in all but one aspect. I-JEPA uses a multi-block masking strategy instead of morphological transformations. M-JEPA (Bardes et al., 2023) proposes a JEPA for learning a self-supervised optical flow estimator. Combining M-JEPA with VICReg (Bardes et al., 2022a), results in MC-JEPA. A-JEPA (Fei et al., 2023a), and V-JEPA (Bardes et al., 2024) are extensions of I-JEPA to audio and videos, respectively.

**Decorrelation based Frameworks:**

The works discussed in this section primarily use Barlow Twins (BT) (Zbontar et al., 2021) presents an innovative approach without using any similarity-based loss. The framework proposed in BT uses an objective which can be understood as an instantiation of information bottleneck, maximising the variability of the representations over the dimensions, thereby preventing dimensional collapse, and also discarding redundant information arising from applied distortions or augmentations. The advantage of BT over InfoNCE-based frameworks is that it does not require a large batch size and benefits from large-dimensional embeddings. Barlow Twins was adopted as the baseline framework in several applications like MohammadAmini et al. (2022), Graph BT (Bielak et al., 2022), Gomez-Villa et al. (2021), etc. He & Ozay (2022) argue that Barlow Twins output whitened features, and explore the relation between collapse of representations and whitening of features, and the exponent of eigenspectrum which follows the power law decides the gap. The authors also propose a post-hoc method to scale the eigenspectrum of the pre-trained encoder, which eliminates the need to train a linear classifier on top of the encoder for downstream tasks. Hua et al. (2021) explores the concept of collapse in BT (Zbontar et al., 2021) and WMSE (Ermolov et al., 2021) as baseline frameworks and claims to discover dimensional collapse in SSL as well. This work explores the role of feature decorrelation and proposes Shuffled Decorrelated BN for improved representation learning and prevention of dimensional collapse in SSL.

VICReg (Bardes et al., 2022a) also uses the decorrelation principle like BT, in addition to an invariance term to minimize the distance between positive samples, and a variance term to maintain the variance of each term above a predefined threshold, thus enforcing the embeddings to be different and preventing collapse. VICRegL (Bardes et al., 2022b) improves VICReg by adding location-based and feature-based matching of embeddings across both views of positive samples. Shwartz-Ziv et al. (2023) present an information-theoretic perspective of SSL with VICReg as the baseline framework. SMT (Chen et al., 2023e) is the simplest form of VICReg but uses a more restrictive linearity criterion for similarities, where local, temporal, or spatial neighbour linear interpolation is used to define similarity on the manifold.

**Spectral Decomposition based Feature Whitening:**

W-MSE (Ermolov et al., 2021) does not use a separate predictor like BYOL and also avoids collapse while using the same loss as BYOL. W-MSE achieves this by using a Cholesky decomposition step to whiten the features along the batch dimension. Whitening prevents the pre-representations from collapsing by having a scattering effect on the samples and forces the vectors thereby obtained to be uniformly distributed on the unit sphere.

ZeroCL (Zhang et al., 2022d) proposes a novel approach to self-supervised representation learning using independent invariance minimization along both batch and feature dimensions after instance-wise and feature-wise ZCA whitening, respectively. ZeroCL eliminates the redundancy reduction term in BT (Zbontar et al., 2021) by feature-wise whitening of the representations before calculating the loss. However, like W-MSE, ZeroCL involves a whitening step using Cholesky decomposition that is computationally expensive and is of the order of $\mathcal{O}(n^3)$.

ARB (Zhang et al., 2022c) proposes another new approach using orthonormal bases of one view of a sample as a target for the feature representations of the other view. However, ARB divides the representation in subgroups similar to MDRA (Cheng et al., 2023) before computing the orthonormal bases to reduce computational cost. To account for non-full rank cases, ARB computes the pseudo-bases by using a spectral decomposition of the correlation matrix of the output representations, which involves a computationally expensive step.

**Self-Distilation based Frameworks**

Self-distillation is similar to knowledge distillation (Hinton et al., 2015) in supervised learning, but without *a priori* teacher network. In DINO (Caron et al., 2021) the parameters of the teacher network are generally obtained from the momentum encoding of the parameters of the student network over training iterations. The student network is trained to learn local-to-global correspondences by matching the probability distribution of both networks. EsViT (Li et al., 2022a) improves MoCov3 (Chen et al., 2021d) / DINO (Caron et al., 2021) by studying the properties of ViT (Dosovitskiy et al., 2021) in SSL. EsViT observes that ViTs can automatically discover semantic correspondence between local regions, but the use of multi-stage ViT causes a loss of property. EsViT proposes a novel non-contrastive region matching pretext task to capture the local region dependency in the features. ReSSL (Zheng et al., 2021b) presents a novel framework based on relational consistency between instances instead of explicitly repelling instances in negative pairs and pulling augmented views of the same instance. It uses a teacher-student framework to obtain representations of two views of an instance and computes the similarity distribution of each instance with the representations in the memory bank. Finally, the KL divergence is minimized to enforce the relation consistency between the two augmented views of an instance. A similar approach to ReSSL is also applied in ISD (Tejankar et al., 2020), however, instead of a memory queue, it uses a collection of random samples to approximate the neighbourhood of the sample. In another work, SCE (Denize et al., 2022) combines MoCov2, ReSSL, and N-pair contrastive loss for video representation learning. Whereas, auxSKD (Dadashzadeh et al., 2022) applies a framework similar to DINO for spatio-temporal representation learning as an auxiliary task on top of a predictive primary pretext task. Yun et al. (2022) improves DINO by mining positive patches from the neighbouring patches and using the aggregated representation as the target.

The bag-of-words approach has been used in classical computer vision in the past. Gidaris et al. (2020) uses a self-distillation type approach, where a visual word vocabulary is used to quantize/encode the representations and generate a probabilistic softmax output. OBoW (Gidaris et al., 2021) differs from SwAV only in the clustering part. Similarly to Gidaris et al. (2020), OBoW uses a bag-of-words or a queue of features as a visual-words vocabulary, which is synonymous with the cluster prototypes in SwAV.

OPUN (Ren et al., 2021) uses a self-distillation approach similar to DINO, but uses an online clustering algorithm for prototypes, where the authors use an additional loss for new cluster formation. In more recent work, DINOv2 (Oquab et al., 2023) scales self-supervised pretraining in terms of data and model size. It combines DINO (Caron et al., 2021) and iBOT (Zhou et al., 2022a) with the centering of SwAV (Caron et al., 2020) and KoLeo regularization (Sablayrolles et al., 2019).

MST (Li et al., 2021e) also adopts a similar approach, using a self-distillation architectural framework such as DINO (Caron et al., 2021), and also optimizes a reconstruction loss from the student network. Visual tokens in the student network are masked using an attention-guided mask strategy, conditioned on the teacher network encoder output, to mask out low response patches.

CompRess (Koohpayegani et al., 2020) learns a small student network from a large self-supervised teacher network each with a separate memory bank using knowledge distillation (Hinton et al., 2015). Similar to CompRess, SEED (Fang et al., 2021) emphasized that small models with fewer parameters cannot learn instance discrimination effectively. However, the similarity distribution is computed by randomly sampling instances from a dynamically maintained queue, and it fails to effectively model similarity of those highly related samples. To solve this issue, BINGO (Xu et al., 2021a) proposes a new self-supervised distillation method consisting of two components: inter-sample distillation for pushing two augmentations of the same instance together and intra-sample distillation for pushing all instances in one bag to be more similar with the anchor one.

*Combining Masked Image Modelling:* MSN (Assran et al., 2022) combines masked image modeling with the self-distillation framework. However, unlike DINO (Caron et al., 2021), MSN uses a set of prototypes to calculate the softmax probabilities. PMSN (Assran et al., 2023b) relaxes the condition of uniform clustering in MSN by minimizing KL divergence with a power law distribution instead of a negative entropy term in MSN.

*Combining Clustering*: CrOC (Stegmüller et al., 2023) uses DINO as the baseline framework and adds a representation centroid-based self-distillation pipeline with it.

**Distribution Divergence Minimization:** TWIST (Wang et al., 2023a) presents an interesting approach by minimizing the divergence between the probability distributions of two augmented samples, along with the entropy of each sample. TWIST also uses a diversity term to ensure that representations of different samples are different to prevent collapse. Although similar in principle to DINO, the loss used in TWIST uses KL divergence explicitly.

PMO (Luo & Wang, 2022) uses a matching operator to match the representations of the input to a prior distribution, without the need to contrast the positive and negative samples, thus preventing collapse.

**Non-Contrastive + Instance Disc. Frameworks:** MNCLR (Long et al., 2022) combines SimSiam with MoCo (He et al., 2020) to build a multi-network framework for SSL.

## 2.4 Miscellaneous

In this subsection, we include the works that cannot be explicitly categorized into the above-mentioned categories. In general, these works combine multiple frameworks into one single one. In addition to that, we have also included works which discuss metrics for evaluating SSL frameworks and conduct analysis on different aspects of SSL, in this subsection as well.

The design of the object counting problem as a pretext task in SSL can be seen in Noroozi et al. (2017), where the authors use a contrastive loss, instead of a regression loss, to prevent trivial solutions, a common problem in SSL.

Kolesnikov et al. (2019) studied the effect of CNN architectures with different SSL frameworks and found that: a) architecture choices which negligibly affect performance in the fully labeled setting may significantly affect performance in the self-supervised setting, b) the quality of learned representations in CNN architectures with skip connections does not degrade toward the end of the model, c) increasing the number of filters in a CNN model and consequently, the size of the representation significantly and consistently increases the quality of the learned visual representations, and d) linear probing performance is sensitive to the learning rate.

VFS (Xu & Wang, 2021) uses both MoCo (He et al., 2020) and SimSiam (Chen & He, 2020) for two different versions of their proposed approach for correspondence learning from videos. Besbinar & Frossard (2021) uses next frame prediction as the pretext task for reconstruction-based self-supervised representation learning.

Addepalli et al. (2022) combines handcrafted pretext task rotation prediction with foundation models such as SimCLR (Chen et al., 2020b), MoCov2 (Chen et al., 2020e), BYOL (Grill et al., 2020), SwAV (Caron et al., 2020) in a multitask learning environment.

Although SSL has advanced rapidly, there has been dearth of metrics for benchmarking those frameworks. Gwilliam & Shrivastava (2022) proposed several metrics for benchmarking and analyzing self-supervised frameworks in their work.

Islam et al. (2021) show that combining supervised loss with self-supervised contrastive loss improves transfer learning performance. Zhang et al. (2021c) also used contrastive loss as a regularizer along with cross-entropy loss in the downstream finetuning stage. Around the same time, Cole et al. (2021) asks some important questions about the impact of data quality and quantity, task granularity, and pretraining domain on the quality of representations learned in contrastive learning, and also answers them through empirical analysis.

As stated in Ryali et al. (2021), in self-supervised learning, commonly used augmentation pipelines treat images holistically, ignoring the semantic relevance of parts of an image leading to the learning of spurious correlations. This work addresses this problem by investigating a class of simple, yet highly effective background augmentations, which encourage models to focus on semantically-relevant content by discouraging them from focusing on image backgrounds. Basaj et al. (2021) proposes several visual probing tasks previously used in NLP to evaluate SSL frameworks.

UnMix (Shen et al., 2020) employs image mixing methods like CutMix (Yun et al., 2019) and MixUp (Zhang et al., 2018) to implement an unsupervised counterpart of label smoothing in supervised learning to improve representation learning. MixCo (Kim et al., 2020b) and i-Mix (Lee et al., 2021) are other concurrent works exploring the same image mixing strategy for contrastive learning algorithms.

ScoreCL (Kim et al., 2023) adds a score-based weighting mechanism to each term in both contrastive and non-contrastive frameworks to improve representation learning. Li & Liu (2023) uses a two-stage pretraining for video representation learning, the first stage being a contrastive learning stage, and the second stage a combination of distillation and reconstruction tasks.

Allen-Zhu & Li (2023) study the effect of the ensemble in the testing phase and whether the ensemble can be distilled into a single model. Ruan et al. (2023) also explores the ensembling of teacher-student networks using different weighting schemes for cross-entropy objectives.

**Summary:** In the above section, we have discussed some of the foundational research works in the field of self-supervised learning which have laid the path for adoption and subsequent application of the same in different domains. Along with the foundational works, we have also discussed several modifications and derivations of those works which have improved the baselines. Through these discussions, we observe the wide spectrum of pretext tasks that have been discovered and effectively implemented for representation learning from unlabeled data. We also categorized the works based on the principle of their respective baseline framework, which allows a deeper understanding of these works on both conceptual and fundamental levels.

In the following section, we look into the works which have employed the foundational frameworks or their derivations in medical image analysis applications. We explore works on multiple modalities, namely, magnetic resonance imaging, computed tomography, ultrasound, echocardiography, etc. In this section also, we categorize the works based on the type of framework used. We have aimed to discuss works with diverse anatomical application regions within the same modality, allowing us to widen our understanding of the application spectrum.

## 3   Application of SSL in Medical Image Analysis

Self-supervised learning has demonstrated significant prowess in the domain of representation learning from medical imaging modalities. Unlike natural image datasets, medical image datasets are not abundant and are also expensive to annotate. Thus, self-supervised learning aids in adapting the parameters to the data distribution of the medical imaging modalities better than using ImageNet-pretrained weights. We have discussed studies supporting the above statement later in this section. Self-supervised learning frameworks have been applied to different medical imaging modalities for a host of tasks like classification, segmentation, anomaly detection, image reconstruction, etc.

Classification tasks include tasks like tumor classification from magnetic resonance or computed tomography images, injury classification from knee magnetic resonance images, identifying normal or abnormal tissues in histopathological images, etc. Similarly, SSL pre-training has also found application for segmentation tasks like tumor segmentation, tissue substructure segmentation, skin lesion segmentation, organ segmentation from whole-body magnetic resonance or computed tomography images, etc.

SSL is also used for reconstructing medical images from corrupted images or for denoising or removing artefacts in the same. Self-supervised learning is also applied for super-resolution of medical images. Discovering patterns in medical data for anomaly detection by learning normal patterns without the need for extensive labeled data is also possible with self-supervised representation learning. Generation of synthetic data to deal with data scarcity is also made possible with the help of self-supervised pre-training.

Furthermore, self-supervised learning can also be used for visualization of medical image datasets, multi-modal analysis of medical data, volumetric analysis of medical data such as fetal pose estimation, reference plane detection in ultrasound data, etc.

In the following subsections, we discuss several studies which utilise self-supervised frameworks for learning representation for the above mentioned downstream tasks.

## 3.1 MRI & CT

### 3.1.1 Application of Context based frameworks

Models Genesis (Zhou et al., 2021b) uses an image restoration-based task (from corrupted images) to learn image representations. Semantic Genesis (Haghighi et al., 2020) builds on Models Genesis by adding another stage of image reconstruction pre-training before the image restoration pipeline. Jana et al. (2021b) also uses image restoration as a pretext task to learn representations from CT images for liver fibrosis diagnosis. CaiD (Taher et al., 2022) reconstructs the original image from the corrupted version of the same image to learn context-aware representations in addition to an instance discrimination task.

Chen et al. (2019) uses restoration of corrupted images as a pretext task using a reconstruction-based framework. Sli2Vol (Yeung et al., 2021) uses a slice reconstruction-based strategy to learn representations to segment regions in 3D CT or MRI volume. Lu et al. (2020) and TractSeg (Lu et al., 2021) use pseudo-labels obtained from tractography for reconstruction to learn representations from fMRI data for segmentation.

Demirel et al. (2021) uses a reconstruction-based framework proposed by Yaman et al. (2020) for simultaneous multi-slice image reconstruction task itself. Jana et al. (2021a) and Zhang et al. (2021b) use a combination of GAN-based and AE-based reconstruction for unsupervised representation learning.

SSL-LNE (Ouyang et al., 2021) also uses a reconstruction-based framework to learn the disease progression trajectory of individuals. Akçakaya et al. (2022) gives an overview of the different unsupervised methods used for biomedical image reconstruction. In another work, SwinMAE (Xu et al., 2023b) uses a Swin transformer based encoder-decoder architecture to learn representations from MR data for parotid tumor segmentation.

Sun et al. (2021b) utilizes simulated artefacts obtained from the downsampling of MR scans to incorporate cortical thickness as anatomical guidance. In the testing phase, an iterative training stage is used to learn a site-specific segmentation network.

Dong et al. (2021) reconstruct a fixed number of slices preceding and following the input slice of CT scans to learn representations. This work also uses an auxiliary task similar to BYOL (Grill et al., 2020) in addition to the reconstruction-based task. Similarly to Dong et al. (2021), Alice (Jiang et al., 2023) uses a combination of masked image modelling and maximization of similarity between semantically aligned crops obtained using SAM (Yan et al., 2020) for representation learning.

Chen et al. (2022e) show how masked image modelling outperforms traditional contrastive learning by speeding up convergence and greatly improving downstream task performance and can be utilized to advance 3D medical image modelling in a variety of situations.

Matzkin et al. (2020) uses 3D reconstruction from postoperative CT scans to estimate missing bone flaps. Another work Zhang et al. (2023c) also uses a 3D reconstruction-based framework based on UNETR or Swin-UNETR for representation learning. SSPT-bpMRI (Yuan et al., 2023) uses a 3D UNet for reconstruction of 3D volume from an augmented sub-volume. The representations are then used for the detection and diagnosis of csPCa (prostate cancer).

OneSeg (Wu et al., 2022b) uses a reconstruction framework to learn the semantic correspondence between two different 2D slices from 3D CT scans. In the inference phase, the annotated data is propagated using this pre-trained reconstruction framework from a randomly selected representative slice.

Huang et al. (2022a) uses symmetric positional encoding for Brain MR slices and 3D VHOG as targets for reconstruction from masked 3D voxels. VectorPose (Zhang et al., 2023g) uses boundary and voxel

reconstruction, as well as spatial vector prediction, to learn spatial and anatomy-sensitive representations of 3D volumes.

Mazher et al. (2024) uses a style transfer-based approach to learn representations from private datasets without compromising the data privacy of clients in a federated learning setting. The transferred model weights are used for subsequent downstream tasks such as segmentation. Several other works like Zhao et al. (2023), $M^3AE$ (Liu et al., 2023a), Tajbakhsh et al. (2019) also use a reconstruction-based framework.

Several works like Jog et al. (2016), Zhao et al. (2018), Xu et al. (2021b) and Zhao et al. (2021) use super-resolution as a pretext task. This allows the networks to learn contextual representations specific to the data, and also deal with the scarcity of high-resolution medical data.

PrimeGeoSeg (Tadokoro et al., 2023b) and Tadokoro et al. (2023a) synthesize 3D volumetric data using geometric shapes to emulate 3D MR scans and train a segmentation network using the same pretext task. In another work, Zhang et al. (2023e) uses a synthetic tumour data generation pipeline for learning to segment brain tumours. To et al. (2021) uses a reconstruction-based generative framework to generate augmented samples to deal with data scarcity issues in self-supervised learning.

DualHierNet (Xue et al., 2020) uses low-level features-based adversarial training between the two domains, along with semantic level and edge-level adversarial networks. Tomar et al. (2021) uses a generative style-transfer framework to learning learn volumetric representations for one-shot segmentation of brain MR scans. Other works using generative adversarial training but discussed in other subsections are Jana et al. (2021a), Zhang et al. (2021b), Tao et al. (2020), Yang et al. (2020).

TransMorph (Chen et al., 2022b) also uses a reconstruction-based approach for unsupervised image registration by predicting the deformation between fixed and moving images.

Spitzer et al. (2018) uses a Siamese architecture for representation learning from differently cropped versions of the input by maximizing the similarity between the two encoded inputs and also predicting the transformations applied on both inputs. Yang et al. (2020) uses the rotation and elastic prediction task as the source of self-supervisory signals in their framework, where the downstream segmentation module is also jointly trained with the self-supervision module. Furthermore, for disentanglement of appearance and content codes, the proposed framework also uses a modality-transfer generative module to learn cross-modal content-aware representations in an adversarial training way, and a self-reconstruction module.

Zhuang et al. (2019b) propose an interesting approach by using Rubik's cube solving as a pretext task for representation learning from both MR and CT volumes. Zhu et al. (2020) further improves it by adding more augmentations for better representation learning. Taleb et al. (2020) also explores several context-based tasks for representation learning from MRI and CT images. Tao et al. (2020) uses an adversarial strategy like GAN to learn representations by training the discriminator to predict the correct arrangement as real and the rest as fake.

The application of jigsaw puzzle solving to medical image data was first done in Manna et al. (2022), where the authors used a semi-parallel architecture. However, the authors took a slice-based approach to learn representations from MRI scans. Each slice of the MR scan was divided into 9 patches, similar to Noroozi & Favaro (2016). For each patch, the authors used separate convolutional branches. The outputs were later merged and passed through custom convolutional blocks, to finally predict the arrangement of the patches. Similarly to Noroozi & Favaro (2016) and Ahsan et al. (2019), a hamming distance-based selection strategy was used for the set of patch arrangements. The authors showed the robustness of the jigsaw puzzle-solving strategy to a class imbalance in the data. This work was further improved in SKID (Manna et al., 2023), where the authors mainly used two different convolutional blocks to further improve representation learning. To deal with the 3D nature of MR scans, the authors used a ConvLSTM-based (Shi et al., 2015) classifier in the downstream task and kept the encoder parameters frozen. However, in this work, the authors did not use a Hamming distance-based selection strategy. Instead, the authors used a randomly selected set of patch arrangements. The authors achieved an AUC score almost on par with the supervised baseline MRNet (Bien et al., 2018) on the MRNet dataset.

Taleb et al. (2021) uses a multimodal jigsaw puzzle-solving task, where each patch is from a different modality and the objective is to minimise the reconstruction loss between the input and the output, which is obtained by rearranging the input patches according to the predicted permutation matrix using a differentiable sinkhorn operator.

PCRL (Zhou et al., 2021a) and PCRLv2 (Zhou et al., 2023a) use a combination of three pretext tasks, rotation prediction (Gidaris et al., 2018), context prediction (Pathak et al., 2016), and instance discrimination (Chen et al., 2020b) for learning representation from MR scans. PCRLv2 extends PCRL to multi-scale resolutions for better performance along with other architectural changes. $SSL2$ (Wang et al., 2023d), CSwin (Li et al., 2023g) also uses a similar framework for sclerosis segmentation and prostate cancer detection and segmentation, respectively.

In a recent work by Monsefi et al. (2024), the slices are clustered to encode different features and a classification task is used as the pretext task, where the network predicts the suitable cluster for a collection of slices from MR scans.

Li et al. (2020d) proposes a 2D Slice Order Prediction Based Framework from 3D MR or CT volumes for self-supervised representation learning. Rivail et al. (2019) learns spatio-temporal representation from optical coherence tomography images by predicting the time gap between two input B-scans.

In another work, Bai et al. (2019) predicts anatomical positions from cardiac MR data for representation learning. Blendowski et al. (2019) uses the prediction of the relative patch offset as the pretext task. The pretext task is a regression task for predicting the offset as a pair of parameters along all the 3 axes.

Tajbakhsh et al. (2019) does not propose any novel framework, it attempts to find an answer to the question of self-supervised pre-training on limited data provides more effective weight initialization than random initialization or initial weights transferred from an unrelated domain, by analyzing the performance of rotation-based pretext tasks on lung CT scans.

### 3.1.2 Application of Instance Discrimination-based Frameworks

The work Jamaludin et al. (2017) can be considered as one of the first applications of SSL for medical image analysis. This work uses a patient discrimination task using contrastive loss with vertebrate level prediction as an auxiliary task for representation learning. CADx (Chen et al., 2022c) uses InfoNCE loss in texture information extracted from cervical optical CT images to learn representations to detect high-risk diseases, including high-grade squamous intraepithelial lesions and cervical cancer. Santilli et al. (2021) employ contrastive learning on basal cell carcinoma data for pretraining and transfers the weights to differentiate between cancer and normal breast tissue.

You et al. (2021) uses a momentum-encoded volumetric instance discrimination loss (Chen et al., 2020e), dimension contrastive loss (contrasting representations of different dimensions, treating each dimension as a sample) and a consistency loss between the teacher and student network, along with the supervised loss to learn representations from CT scans for volumetric segmentation.

Wu et al. (2021b) and Wu et al. (2022a) construct local positives (same partition of different scans from a single patient), and negatives (different partition of different scans from a single patient) from partitioned scan volumes, and also take scan partitions from different remote patients as negatives. With these samples in a federated environment, momentum-based contrastive learning (Chen et al., 2020e) is applied to representations for volumetric segmentation in the downstream task.

Inglese et al. (2022) also uses volumetric contrastive learning for classification of neuropsychiatric systemic lupus erythematosus patients. Tang et al. (2022b) used a combination of SimCLR, image inpainting, and rotation prediction to learn representation from 3D medical scans.

Fischer et al. (2023) uses a framework similar to that of Jabri et al. (2020) that uses contrastive random walks for self-supervised semantic representation learning.

DrasCLR (Yu et al., 2024b) uses N-pair contrastive loss by sampling positive and negative samples from the neighbourhood in addition to InfoNCE loss to learn representations of 3D lung CT images.

Chaitanya et al. (2020) used a dense contrastive learning task using global and local pixel-level discrimination for representation learning to segment MR images. Yan et al. (2020) also uses a similar framework. OS2 (Yang et al., 2023d) uses a contrastive learning framework with a novel interactive embedding module for support query (SQIE), equipped with channel-wise co-attention, spatial-wise co-attention, and spatial bias transformation blocks to extract interactive information between slices. Vox2Vec (Goncharov et al., 2023) also uses a contrastive learning framework on multi-scale representations to capture both global semantics and local semantics.

Zheng et al. (2021a) uses representations from multiple hierarchical levels of the encoder. Hierarchical features are aggregated and then used in an instance discrimination task to learn representations from MRI and CT scans. In addition to instance discrimination, the proposed framework also uses other context-based pretext tasks and an auxiliary reconstruction-based task as well.

Nguyen et al. (2023) uses SwAV (Caron et al., 2020) as the baseline framework for clustering semantic representations. To learn the dependence between 2D slices in 3D volumes, the aggregated embedding from all the slices is also trained to map close to embeddings of individual slices. Masked embedding predictions are also used as an auxiliary task.

Windsor et al. (2021) uses contrastive learning-based dense correspondence matching between DXA and magnetic resonance imaging, along with unsupervised image registration, to transfer segmentation annotations between the two modalities. A similar multi-modal contrastive learning is also used in Fedorov et al. (2021a), Fedorov et al. (2021b) and Fedorov et al. (2024) for mutual information maximization using different combinations of local and global representations.

Other works like Dong et al. (2021), CaiD (Taher et al., 2022), CISFA (Hu et al., 2022), CSwin (Li et al., 2023g), Liu et al. (2023c) also use instance discrimination as a pretext task in their proposed framework.

### 3.1.3    Application of Non-Contrastive Frameworks

MsVRL (Zheng et al., 2022) uses BYOL as the baseline framework and extends it to multiscale representations of MR scans.

BT-UNet (Punn & Agarwal, 2022) uses Barlow Twins (Zbontar et al., 2021) to train the encoder, which is later fine-tuned for segmentation tasks on MR scans. This work also presents performance data on histopathological and skin lesions.

Ouyang et al. (2020) and Ouyang et al. (2022) uses superpixel-based semantic segments as pseudolabels for few-shot segmentation. In the downstream phase, the pre-trained model can segment organs from MRI or CT data without fine-tuning.

Li et al. (2021a) used a pre-trained network for feature extraction and subsequent $k$-means clustering for sample re-weighting or imbalance-aware selection. The authors use SimSiam (Chen & He, 2020) as a baseline framework for representation learning.

### 3.1.4    Other Applications

Jiang & Miao (2022) uses a variety of SSL methods to pre-train 3D CNN on MR scans for Alzheimer's disease classification. Tang et al. (2022a) learns a max-tree representation from image features to learn structural information from the image to aid in segmenting the medical image in the subsequent task.

## 3.2    Ultrasound

### 3.2.1    Application of Context-based pretext task:

Jiao et al. (2020b) uses transformation prediction and video frame order prediction as a joint prediction task to learn the representation from ultrasound videos. Hu et al. (2020b) combined context encoding pretext task like Pathak et al. (2016) with adversarial training and DICOM metadata prediction to form the pre-training framework.

Qi et al. (2020) uses a jigsaw based task for learning representation from ultrasound images for utero-placental interface detection in the downstream task. Zhang et al. (2024) uses a combination of two context-based pretext tasks, namely, rotation prediction and image pixel ordering prediction to learn representations, which are then used in the downstream fusion architecture for carotid plaque ultrasound image classification. Fang et al. (2023a) also uses rotation prediction along with self-distillation based objective as the pretext task for endometrial disease classification. The rotation prediction pretext task has also been adopted in Roop et al. (2023) for estimating the angular offset of freehand ultrasound probe movement relative to an ideal viewing angle. Another contemporary work, Xie et al. (2024) uses jigsaw puzzle solving as a pretext task to learn representations from thyroid ultrasound images.

Mishra et al. (2022a) generates four types of pseudo segmentation masks using Otsu's thresholding, Canny edge detection, Chan-Vese segmentation, and a combination of Gaussian blur, Sobel filtering and Otsu's thresholding for the pretext segmentation task. The U-Net model trained using this pretext task is more adapted to the downstream segmentation task, as the objective in both tasks are aligned, and can learn the representations related to the tumour segmentation better in the pretext task.

Xiang et al. (2022) predicts the component modalities in the mixed input image to learn representations from the tumour regions. Two images from different ultrasound modalities, namely, US, SWE, or CDUS are mixed only in the region around the nodule keeping the background information uniform, which enforces improvement of the representation learning ability of the network.

Lin et al. (2022) uses a masked video modelling framework based on TimeSFormer auto-encoder architecture (Bertasius et al., 2021) for pre-training, followed by a correlation-aware contrastive framework to enhance feature resemblance for the downstream classification task. In recent work, Fan et al. (2024) uses masked autoencoder to learn representations from breast ultrasound images for tumour classification. A similar framework is also used in Xu et al. (2024) for representation learning from tongue and breast ultrasound images. Sang et al. (2023) also uses a masked image modelling-based framework to learn representations from thyroid and breast ultrasound data for segmentation downstream tasks. SimICL (Zhou et al., 2024) uses visual in-context learning by predicting the query mask from paired query image and support image and mask as reference using a masked image modelling framework. Zhou et al. (2023b) uses SimMIM as the baseline framework for segmentation of wrist ultrasound images.

FetusMap (Yang et al., 2019a) uses a reconstruction-based framework with a landmark detector for fetal pose estimation, aided by generating the pseudo-labels from an atlas of poses. FetusMapV2 (Chen et al., 2024) further enhances the fetal pose estimation by proposing a better memory management framework along with a pair loss to mitigate confusion caused by symmetrical and similar anatomical structures. Alasmawi et al. (2024) proposes a self-supervised representation learning step followed by a self-labelling step (Van Gansbeke et al., 2020) to cluster fetal ultrasound images. Lamoureux et al. (2023) also uses a reconstruction-based framework for cardiac ultrasound images. In Kang et al. (2023b), the authors use a deblurring-based reconstruction framework for thyroid ultrasound classification. Image registration is used as the pretext task in Ding et al. (2024) to learn representations from carotid ultrasound images for segmentation of plaque in the downstream task.

### 3.2.2 Application of Instance Discrimination framework:

Jiao et al. (2020a) attempt to correlate audio with visual features in ultrasound video, along with ensuring that features of audio and video lie close to each other by minimizing a cross-modal contrastive loss. In Perek et al. (2021), a similar multi-modal contrastive framework is adopted for learning representations from mammography and ultrasound data. A multi-modal contrastive learning framework is also used in Jiao et al. (2023), where the authors used video-audio correspondence prediction and cross-modal contrastive learning framework to learn ultrasound video and speech-audio representations. The representations learnt in the pretext task are used to localise anatomical regions of interest during ultrasound imaging, with only speech audio as a reference in the downstream task.

Zhao & Yang (2021) pairs hand-crafted radiomics features obtained from Thyroid nodule ultrasound images with embeddings of the same from an encoder for learning representations with a contrastive learning framework. Chen et al. (2021f) combines multi-class classification supervised task with SimCLR (Chen et al.,

2020b) for boosting represnetation learning. Chen et al. (2023d) presents a framework similar to Chen et al. (2021f) but uses a weighted contrastive learning framework, where the weights are calculated based on the similarity of the samples in a pair. However, the strategy of positive pair generation by mixing frames with coefficients drawn from a Beta distribution to mitigate the positive-pair dissimilarity defect and similarity redundancy problems is the same in both Chen et al. (2021f) and Chen et al. (2023d) Liang et al. (2023a) also uses a self-supervised contrastive framework for learning representations from ultrasound data which are then used to pseudo-label unlabeled data in the subsequent semi-supervised learning stage for anatomy tracking.

Basu et al. (2022) uses both cross-video and intra-video negative frames from ultrasound frames to learn representations. While the cross-video negatives comprise the easy negatives, the intra-video negatives are gradually introduced. The temporal difference between the anchor and intra-video negatives is also gradually decreased to increase the hardness of the task.

HiCo (Zhang et al., 2023a) presents an interesting approach by using features from multiple hierarchical levels of the encoder to implement fine, medium and coarse-grained contrastive learning in addition to global-local contrast framework to improve classification performance on lung and breast ultrasound data.

Wang et al. (2024) uses transverse and longitudinal views of thyroid ultrasound data to perform single-view and multi-view contrastive learning based on MoCov2 (Chen et al., 2020e) framework but with independent encoders and a shared memory bank. However, the encoders use ImageNet pre-trained weights for initialization in the pretext task, thus effectively making it a two-stage pre-training for thyroid nodule classification and segmentation in the downstream task. DSMT-Net (Li et al., 2024a) also uses MoCov2 in addition to another masked image modelling branch for learning representations from endoscopic ultrasound images for detection of pancreatic and breast tumours. DSMT-net also uses a multi-operator transformation module to extract and transform the ROIs ultrasound image into rectangular input.

### 3.2.3 Application of Non-Contrastive framework:

Inspired by BYOL, SelfCSL (Nguyen & Le, 2021) uses the same to pre-train a randomly initialized backbone for semi-supervised learning on small-scale MedMNIST dataset (Yang et al., 2021; 2023b).

Abdi et al. (2024) uses the Barlow Twins framework as the baseline for learning representations to improve keypoint detection performance in Transmitral Doppler imaging, which is a type of ultrasound imaging. VanBerlo et al. (2024) uses a weight for invariance term in VICReg or Barlow Twins depending on the temporal or spatial distance between the samples in a positive pair in lung ultrasound videos. Similarly, To et al. (2024) also uses VICReg as the pre-training framework for a prototype-based out-of-distribution detection in the downstream task. Li et al. (2023a) uses an innovative triplet dimension contrastive learning, where any two out of three views are paired to form the cross-correlation matrix as in Barlow Twins (Zbontar et al., 2021), resulting in three such matrices. The three matrices are finally combined to form the final matrix and the same loss used in Zbontar et al. (2021) is optimized to learn representations.

Lu et al. (2023b) uses SimSiam (Chen & He, 2020) as the baseline framework to learn representations using a position and channel-based dual attention architecture from prostate ultrasound images for cancer screening.

### 3.2.4 Miscellaneous:

Anand et al. (2022) studies different self-supervised learning frameworks, namely, SimCLR, BYOL, and DINO and also analyse their performance on a private Cardiac ultrasound dataset consisting of almost a million images. VanBerlo et al. (2023) also takes a similar approach of studying the effects of different SSL frameworks for detecting absent lung sliding which indicates presence of a host of conditions like pneumothorax, acute respiratory distress syndrome (ARDS), pulmonary fibrosis, large consolidations, pleural adhesions, atelectasis, right mainstem intubation, and phrenic nerve paralysis (Lichtenstein, 2010; Lichtenstein et al., 2005; Bhoil et al., 2021; Husain et al., 2012). Benjamin et al. (2023) also explores different SSL frameworks for fetal cardiac ultrasound images.

### 3.3 Endoscopic Data

#### 3.3.1 Application of Context-based pre-training

Ross et al. (2017) uses an adversarial training strategy, in which the generator produces recolorized images using a U-Net architecture (Ronneberger et al., 2015), which is used for segmentation in the downstream task. Vats et al. (2021) uses rotation prediction and jigsaw puzzle-solving tasks for self-supervised pre-training from wireless capsule endoscopic images. This work also discusses the primary reasons behind the gaps that occur in the learning of semantic representation due to inadequate self-supervised training. In Hong et al. (2021), a reconstruction-based framework is used on colorectal images to learn representations for polyp segmentation.

#### 3.3.2 Application of Instance Discrimination based framework

Jian et al. (2021) uses instance discrimination (Chen et al., 2020b) in endoscopic images to learn representations for the detection of Helicobacter Pylori infection.

In Intrator et al. (2023), the authors explore primarily two methods, single frame instance discrimination and multiview tracklet discrimination. Following Qian et al. (2021b), the authors choose the pre-trained network for the multiview tracklet discrimination task to apply reidentification approaches in colorectal videos to track polyps over frames, which effectively improves polyp classification and detection performance.

In Colo-SCRL (Chen et al., 2023b), the authors combined VideoMAE (Tong et al., 2022) with VideoMoCo (Pan et al., 2021) for representation learning from paired colonoscopy videos. The downstream task is the retrieval of polyp areas from colonoscopy videos of 2nd screening from a given query video of 1st screening.

#### 3.3.3 Application of Non-contrastive frameworks

FPSiam (Gan et al., 2023) uses SimSiam as the baseline framework to learn representation from frames extracted from colorectal videos. In addition to the baseline frameworks, FPSiam utilizes features from intermediate layers to implement local feature similarity to reduce the aliasing effect of upsampling. Finally, the features are transferred for polyp detection in colorectal videos.

### 3.4 X-Ray / Radiographs

#### 3.4.1 Application of Context-based Frameworks

IDEAL (Mahapatra et al., 2021) takes a saliency map-based interpretability-driven sample selection approach. The only self-supervised part of this work is the use of autoencoder for clustering the X-ray images using the latent feature vectors. DiRA (Haghighi et al., 2022) and Haghighi et al. (2024) also fall into this category of frameworks.

#### 3.4.2 Application of Instance discrimination based framework:

Works like Sowrirajan et al. (2021); Chen et al. (2021e) use MoCo as a baseline framework for self-supervised pre-training. MedAug (Vu et al., 2021) uses a unique approach of using patient metadata to pair scans to construct positive pairs in contrastive learning. Tiu et al. (2022) uses a multimodal contrastive framework to learn representations from chest radiograph images, to predict pathology in the downstream task. A similar approach is also adopted in Liao et al. (2021b). Hu et al. (2021) also uses MoCov2 (Chen et al., 2020e) framework as the baseline for representation learning from panoramic radiographs of the jaw, for subsequent classification and segmentation of tumours or cysts in downstream tasks.

Liu et al. (2021b) uses JCL as the baseline framework for pre-training the mean teacher for semi-supervised classification of chest X-rays. Sun et al. (2021a) uses both patch or node based contrastive learning and graph level contrastive learning to learn both global and local representations from chest radiographs.

DiRA (Haghighi et al., 2022) and Haghighi et al. (2024) use a combination of image restoration, adversarial, and instance discrimination framework for learning representation from chest radiographs.

ConVirt (Zhang et al., 2022g) uses paired chest radiographs and text reports for text-guided cross-modal contrastive learning of visual representations. Zhang et al. (2023d) uses a disease classifier by distilling knowledge from a network trained using cross-modal contrastive loss using paired image and text information.

### 3.4.3   Other frameworks

Li et al. (2023b) uses a novel SSL framework based on SimSiam with an additional cross-view MSE loss for gastritis detection from x-ray images. Park et al. (2022) uses DINO as the baseline framework for learning representations from a teacher network pre-trained on a small dataset.

## 3.5   Retinal Images

### 3.5.1   Application of Context-based prediction task

Holmberg et al. (2020) used the macular thickness obtained from the automatic segmentation of the optical coherence tomography volume as pseudo-labels for the pretext task of predicting macular thickness from IR fundus images. The pre-trained network is then used for the classification of diabetic retinopathy in colour fundus images.

Hervella et al. (2018) uses multimodal reconstruction as a self-supervised pretext task. This work is used in Álvaro S. Hervella et al. (2020) to deal with label scarcity. In Álvaro S. Hervella et al. (2021), the pretext task of multimodal reconstruction of fluorescence angiography from retinography is approached using aligned retinography-angiography pairs as pretraining data. In Hervella et al. (2020), the same pretext is used for joint optical disc and cup segmentation in images of the eye fundus.

Uni4Eye (Cai et al., 2022) proposes a masked image modelling approach with a novel unified patch embedding module to learn unified representations from 2D colour fundus images or Fundus Fluorescein Angiography (FFA)) and 3D optical coherence tomography (OCT) and optical coherence tomography (OCTA) images.

Yang et al. (2023c) uses multi-modal masked relational modelling, to enrich the semantic relationship among diseases. Relational matching is proposed to capture an abundant disease-related relationship by aligning the sample-wise feature relation between intact and masked features at both the self- and cross-modality levels.

### 3.5.2   Application of Instance Discrimination framework

Mojab et al. (2020) uses data from multiple devices/domains and applies SimCLR (Chen et al., 2020b) as the baseline framework to learn representations and shows that a multidomain self-supervised contrastive learning approach performs better than supervised transfer learning. Gupta et al. (2023b) also uses instance discrimination for representation learning from fundus images.

Li et al. (2020c) uses two stages for self-supervised representation learning. First, it trains a CyCleGAN (Zhu et al., 2017) to synthesize Fundus Fluorescense Angiography (FFA) images from colour fundus images and uses this network to synthesize FFA images from fundus images in the target dataset. The different modalities are then used to learn modality-invariant representations using a patient discrimination (contrastive learning) framework. Li et al. (2021c) also uses two different pretext tasks, but in a collaborative learning or multitask setting. This work uses rotation prediction and patient/instance discrimination together for pre-training.

**Other Applications**   Srinivasan et al. (2022) study the effect of self-supervised pretraining and imagenet-pre-trained weights on data sets for diabetic retinopathy.

## 3.6   Histopathology

### 3.6.1   Application of Context-based Frameworks

Štepec & Skočaj (2020) uses generative image synthesis as a pretext task for anomaly detection in the downstream task. StarDist (Prakash et al., 2020) uses a denoising framework for learning representation from

biomedical microscopy images for downstream segmentation tasks. Stacke et al. (2020) uses the framework of CPC (van den Oord et al., 2018) on histopathological images for representation learning. The study found that only low-level CPC features are relevant for tumour classification.

Sahasrabudhe et al. (2020) uses a scale prediction network along with enforcing the equivariance of representations under transformations and smoothness regularization for representation learning.

Xie et al. (2020a) uses a count ranking and a scale discrimination loss based on triplet loss to learn representations for nuclei segmentation. The scale discrimination loss is used to learn nuclei shape aware information, and the count ranking loss is simply used to learn context-aware representations by training the network to learn the number of nuclei-shaped objects in the input.

### 3.6.2 Application of Instance Discrimination Frameworks

Ciga et al. (2022) uses SimCLR (Chen et al., 2020b) as the baseline framework for applying contrastive learning on histopathological images. DSMIL (Li et al., 2020a) uses a pre-trained SimCLR (Chen et al., 2020b) backbone for weakly supervised multi-instance learning on whole slide images. Saillard et al. (2021) uses a pre-trained U-Net to extract background subtracted whole slide images, and then divide them into multiple patches. These patches are then used to learn representations in a self-supervised way using a multiple-instance learning framework. Srinidhi et al. (2022) uses self-supervised learning only for learning representations from histopathology images. RNAPath (Cisternino et al., 2023) uses a multi-instance learning framework too, to learn representations from 1.7M wide slide images across 23 healthy tissues in 838 donors using a ViT encoder for downstream tasks like tissue substructure segmentation. The model estimates the gene expression at the patch level by independent gene-wise linear regressions, to obtain patch-level scores, which are averaged to obtain a sample-level prediction.

CELLULOSE (Wolf et al., 2023) uses an object-centric contrastive approach by maximizing the distance between the embeddings of patches from different objects and minimizing the distance between the embeddings of patches from the same object, to allow the segmentation of individual cells in microscopy images.

### 3.6.3 Application of Few-Shot segmentation based approaches

Pseudo-label based few shot segmentation approaches similar to Ouyang et al. (2020) was also adopted in Dawoud et al. (2022a) for cell segmentation in microsopy images. An edge-based reconstruction branch is also used as self-supervision in a semi-supervised framework in Dawoud et al. (2022b).

**Miscellaneous Applications** Self-Path (Koohbanani et al., 2021) uses a host of self-supervised tasks as auxiliary tasks along with the primary task from pathological images. In a recent work, Kang et al. (2023a), presents an in-depth analysis of four different SSL frameworks, MoCoV2, SwAV, Barlow Twins, and DINO, when applied to large scale pathology data. This large-scale study yields several useful contributions such as it shows the benefit of using SSL pre-training over using ImageNet pre-trained weights for histopathological images, and demonstrating that SSL is label-efficient in pathology where gathering annotation is expensive.

### 3.7 Echo Cardiogram

Echo-SyncNet (Dezaki et al., 2021) uses multiview echocardiogram videos to learn spatiotemporal information by optimizing consecutive frame similarity, correspondence matching, and temporal order of frames. EP (Chen et al., 2021b) synthesizes ECG panorama which allows real-time querying of any ECG views, from one input view using a reconstruction-based framework. Mehari & Strodthoff (2022) uses BYOL (Grill et al., 2020) to learn the representations of the ECG data.

Yang et al. (2022b) uses a combined image reconstruction and colourization-based self-supervised learning framework to learn representations from colour Doppler echocardiography images. Lee et al. (2023) uses ConvNextv2-based (Woo et al., 2023) masked autoencoder to learn representations to diagnose myocardial diseases such as left ventricular hypertrophy and hypertensive heart disease. SimLVSeg (Maani et al., 2024) uses a masked video modeling framework to learn representations from echocardiogram videos for downstream left ventricle segmentation tasks.

In a recent work Damasceno et al. (2024), the authors use ViewFormer (Kulhanek et al., 2022) pre-trained with DINOv2 (Oquab et al., 2023) framework to learn representations from Transthoracic Echocardiography to detect Pulmonary hypertension.

### 3.8 Skin Images

#### 3.8.1 Application of Context-based pretext task

JIANet (Zhang et al., 2022b) uses a jigsaw shuffled skin lesion image as one sample in a positive pair in a jigsaw invariant instance discrimination task. This work also uses a VAE-based reconstruction branch as part of the proposed collaborative learning framework. The reconstruction branch serves as the means to preserve the important semantic features necessary for melanoma segmentation in the downstream task. Zhi et al. (2024) uses both a masked autoencoder and a self-distillation based framework to learn representations from skin images. The student network in the self-distillation framework shares parameters with the encoder in the masked autoencoder branch. To enhance generalization, Zhi et al. (2024) applies exterior conversion augmentation and dynamic feature generation to the inputs to the teacher network.

#### 3.8.2 Application of Contrastive learning based pretext task

STCN (Wang et al., 2021a) uses a combination of transformation invariance, reconstruction, and pseudo-label based classification tasks for learning representations. The pseudo-labels are obtained by clustering the embeddings obtained from the encoder using a modularity-inspired deep topology clustering algorithm.

Wang et al. (2023c) uses dermoscopy and clinical skin images for multi-modal contrastive learning. Yang et al. (2024) uses a combination of discriminative and generative SSL for skin lesion classification.

MHC-PO (Liang et al., 2023b) combines self-supervised contrastive learning with supervised classification tasks. However, to adjust the conflicting gradients between contrastive clustering and classification, the authors use a Pareto optimality phase the authors employ CAGrad, a multi-objective gradient manipulation method (Liu et al., 2021a).

#### 3.8.3 Application of Non-Contrastive learning based pretext task

In one of the most recent works, BOLT (Li et al., 2024c) combines ViT-based BYOL with a difficulty awareness loss. As a pre-processing step, BOLT perturbs the input tokens. An auxiliary task predicts which branch, student or teacher is processing the tokens with a larger level of perturbation, and is optimized by the difficulty awareness loss. Useini et al. (2024) uses a self-distillation framework DINO as the baseline, whereas Morita & Han (2023) uses a BYOL-based network with an adaptive augmentation module for efficient self-supervised representation learning.

### 3.9 Miscellaneous

**ImageNet pretraining boosts self-supervised pre-training:** Azizi et al. (2021) demonstrate that self-supervised learning on ImageNet, followed by additional self-supervised learning on unlabeled domain-specific medical images, significantly improves the accuracy of medical image classifiers. Similar findings were also reported in MoCo-CXR (Sowrirajan et al., 2021), and Manna et al. (2021).

**Medical Image Visualization:** Another innovative application of self-supervised learning is presented in 2D visualization of medical images. In Nwabufo et al. (2024), the authors adopt t-SimCNE (Böhm et al., 2023) to learn a projection of medical images into a two-dimensional space, by replacing the cosine similarity function by the Cauchy similarity used in t-SNE (van der Maaten & Hinton, 2008), to prevent mapping of the feature embeddings to an unit circle manifold.

**Fairness in SSL:** In Seth & Pai (2024), the authors investigate the fairness of different self-supervised learning frameworks on skin lesion image databases which include underrepresented demographic groups.

**Summary:** In the above section, we discussed the works implementing self-supervised learning frameworks for representation learning from medical imaging modalities like magnetic resonance, computed tomography,

ultrasound, retinal images, histopathology images, etc. We explored different anatomical regions even within the same imaging modalities to understand the spectrum of application.

We observe that the modalities with the higher number of applications of SSL are the magnetic resonance and computer tomography modalities, followed by ultrasound and histopathology images. We can speculate that one primary reason is the relatively abundant availability of magnetic resonance or computed tomography data compared to the other modalities. Several researchers have opted for private datasets as well which makes further research progress on such a domain difficult. This points to the significant role data plays in representation learning, whether labelled or unlabeled.

In the next section, we make an effort to summarise the datasets that have been used in all the works we have discussed in this manuscript. While we could not gain access to the private datasets, we have summarised all the public datasets in the respective modalities.

# 4 Datasets and Benchmarks

Researchers have also proposed some datasets specifically for the purpose of benchmarking the performance of self-supervised learning frameworks. We will discuss two such works which have proposed benchmarking in 11 different domains to test the versatility and adaptability of self-supervised learning frameworks.

## 4.1 Benchmarking datasets

DABSv1.0 (Tamkin et al., 2021) proposes a mixture of datasets for domain-agnostic benchmarking of self-supervised learning frameworks. It consists of datasets from several domains such as natural images, speech, monolingual and multilingual text, medical imaging datasets, multi-channel sensor data, and paired images and text data. In Table 1, we document the different datasets used for pre-training and downstream performance evaluation for each domain covered in DABS.

DABSv2.0 (Tamkin et al., 2022) included more domains in addition to the ones used in DABSv1.0 (Tamkin et al., 2021). The bacterial genomics sequence dataset, semiconductor wafers manufacturing database, particle physics tabular dataset, protein sequence dataset, and multispectral satellite images are the new domains added in DABSv2.0. In Table 2, we document the different datasets used for pre-training and downstream performance evaluation for each domain covered in DABS.

Table 1: Summary of datasets used in DABSv1 benchmarking for SSL

| Domain | Pre-training | Downstream |
|---|---|---|
| Natural Images | ImageNet (Deng et al., 2009) | FGVC-Aircraft dataset (Maji et al., 2013), the Caltech-UCSD Birds dataset (Wah et al., 2011), the German Traffic Sign Recognition Benchmark dataset (Houben et al., 2013), the Describable Textures Dataset (Cimpoi et al., 2014), the VGG Flower Dataset (Nilsback & Zisserman, 2008), and the CIFAR-10 dataset (Krizhevsky, 2009) |
| Speech | LibriSpeech (Panayotov et al., 2015) | VoxCeleb (Nagrani et al., 2020) and LibriSpeech (Panayotov et al., 2015) speaker recognition datasets, Fluent Speech Commands cls. (Lugosch et al., 2019), Google Speech Commands (Warden, 2018), and AudioMNIST (Becker et al., 2023) utterance classification tasks |
| Monolingual Text | WikiText-103 (Merity et al., 2017) | GLUE Benchmark (Wang et al., 2018) |
| Multilingual Text | mC4 dataset (Raffel et al., 2020) | PAWS-X tasks (Yang et al., 2019b) |
| Medical Imaging | CheXpert (Irvin et al., 2019) | Chest-Xray8 (Wang et al., 2017a) |
| Multi-Channel Sensor data | | PAMAP2 (Reiss & Stricker, 2012) |
| Paired Image-Text | MS-COCO (Lin et al., 2014) | MS-COCO and Visual-Question Answering (Antol et al., 2015) modelled as binary classification tasks |

## 4.2 Natural Image and Video Datasets

In this section, we also summarize the different datasets used in the works discussed in this survey. In Table 3 and 3, we can see the summary of the natural image and video datasets, respectively, or non-medical datasets.

Table 2: Summary of datasets added to DABSv1 ot form the DABSv2 benchmarking for SSL

| Domain | Pre-training | Downstream |
|---|---|---|
| Bacterial Genomics | Genomics OOD dataset (Ren et al., 2019) | |
| Semiconductor Wafer Manufacturing | WM-811K (Wu et al., 2015a) | |
| Particle Physics | HIGGS dataset (Baldi et al., 2014) | |
| Protein Sequence dataset | Pfam (El-Gebali et al., 2019) | TAPE benchmark (Rao et al., 2019) |
| Multispectral Satellite Imagery | EuroSAT (Helber et al., 2019) | |

We make two columns to identify the task (both pretext and downstream) in which the datasets have been used. We see that most small-scale datasets have been used primarily, for instance discrimination-based tasks, while context-based pretext tasks have used datasets that have images with large resolutions. This is mainly due to the nature of the tasks, as the context is less identifiable in images with lower resolution. Whereas, for paired embedding-based tasks, learning a global semantic feature aided the downstream objective of image classification. However, for downstream tasks like object detection, and semantic segmentation, image datasets with higher resolution are preferred.

Table 3: Summary of natural image datasets used in Self-supervised pre-training

| Dataset | Training Samples | Num. of Classes | Source | PT Task | DS Task |
|---|---|---|---|---|---|
| Image Datasets | | | | | |
| CIFAR10 | 50K | 10 | Krizhevsky (2009) | Paired Emb. tasks | Image Cls. |
| CIFAR100 | 50K | 100 | Krizhevsky (2009) | Paired Emb. tasks | Image Cls. |
| FC100 | | 100 | Oreshkin et al. (2018) | - | Few Shot Cls. |
| STL10 | 100K (unlabeled) + 5K (train) | 10 | Coates et al. (2011) | Paired Emb. tasks | Image Cls. |
| Tiny ImageNet | 100K | 200 | Le & Yang (2015) | Paired Emb. tasks | Image Cls. |
| Aircraft | 10200 | 102 | Maji et al. (2013) | - | Image Cls. |
| DTD | 5640 | 47 | Cimpoi et al. (2014) | - | Image Cls. |
| Oxford Pets | 7.5K | 37 | Parkhi et al. (2012) | - | Image Cls. |
| Oxford Flowers | 102 | >5K | Nilsback & Zisserman (2008) | - | Image Cls. |
| ImageNet100 | 130K | 100 | Tian et al. (2020a) | Inst. disc. | Image Cls. |
| ImageNet1K | 1300K | 1000 | Deng et al. (2009) | Paired Emb. tasks, MIM, etc. | Image Cls. |
| Places205 | 2.4M | 205 | Zhou et al. (2014) | | Scene Classification |
| PACS | 5156 | - | Li et al. (2017) | Jigsaw, Reconstruction | - |
| ADE20K | 20K | 150 | Zhou et al. (2017) | | Semantic seg. |
| PASCAL VOC | 3K | 20 | Everingham et al. (2010) | - | Obj. Det. & Seg. |
| MS COCO | 328K | 91 | Lin et al. (2014) | - | Obj. Det., Inst. / Semantic Seg. |
| NYU-depth v2 | 407,024 (unlab.) + 1449 (lab.) | 1000+ | Silberman et al. (2012) | - | Depth Estimation |
| CityScapes | 25K | 30 | Cordts et al. (2016) | - | Semantic Seg. |
| JFT-300M | 300M | 18291 | Sun et al. (2017) | Inst. disc. | Multilabel cls. |
| YFCC100M | 100M | - | Thomee et al. (2016) | Inst. disc. | - |
| SUN397 | 108754 | 397 | Xiao et al. (2014) | - | Scene classification |

Table 4: Summary of Video Datasets used in Self-supervised pre-training

| Dataset | Training Samples | Num. of Classes | Source | PT Task | DS Task |
|---|---|---|---|---|---|
| Video Datasets | | | | | |
| Moment in Time | 1M | 339 | Monfort et al. (2019) | Context / Inst. disc. | Action Cls. |
| Kinetics600 | 500K | 600 | Kay et al. (2017) | Context / Inst. disc. | Action Cls., Video ret. |
| Kinetics400 | 240K | 400 | Kay et al. (2017) | Paired Emb. tasks | Action Cls, Video ret. |
| UCF101 | 13K | 101 | Soomro et al. (2012) | Context | Action Cls., Video ret. |
| HMDB51 | 6766 | 51 | Kuehne et al. (2011) | Context | Action Cls., Video ret. |
| ActivityNet | 19995 | 200 | Heilbron et al. (2015) | | Action cls. |
| BreakFast | ≈2K | 48 | Kuehne et al. (2014) | - | Action cls. & seg. |
| FineGym | 32697 | 530 | Shao et al. (2020) | - | Action cls. & Seg. |
| 50Salads | 50 | 19 | Stein & McKenna (2013) | - | Action cls. & Seg. |
| R2V2 | 2,788,424 | - | Gordon et al. (2020) | Inst. disc. | - |
| OTB | 100 | - | Wu et al. (2015b) | - | Object Tracking |
| Something-Something | 100K | 174 | Goyal et al. (2017) | - | Action cls. |
| FCVID | 91223 | 239 | Jiang et al. (2018) | - | Video Categorization |
| VidSitu | 29.2K | - | Sadhu et al. (2021) | | Video / Movie Und. |
| AudioSet | 2M | 632 | Gemmeke et al. (2017) | Multi-modal Inst. disc. | Audio Event rec. |
| HowTo100M | 136M | 2K | Miech et al. (2019) | - | Action cls. |
| YouCook2 | 2K | 89 | Zhou et al. (2018) | - | Action rec. |
| MSR-VTT | 10000 | 20 | Xu et al. (2016) | - | Video Retrieval |
| DAVIS 2017 | 150 | - | Pont-Tuset et al. (2018) | - | Video Obj Seg. |
| Diving48 | 16067 | 48 | Li et al. (2018) | - | Diving action Cls. |
| SoundNet | 2M | - | Aytar et al. (2016) | Multi-modal inst. disc. | Visual and Sound Cls. |
| AVA | 427 | 80 | Gu et al. (2018) | Multi-modal inst. disc. | Action loc. |

## 4.3 Medical Datasets

In Table 5, 6, 7, 8, 9, 10, 11, 12, and 13, we present the different medical data sets used in self-supervised learning for magnetic resonance imaging (MRI), coherence tomography (CT), ultrasound (USG), radiographs, electrocardiogram (ECG), retinal fundus image, endoscopic videos, histopathological images, and skin image datasets, respectively. We have also tried to note the purpose for which the respective datasets were used. Note that not all datasets were used for self-supervised pretraining. The datasets which were only used for downstream evaluation have an empty "PT Task" cell. We have also provided the source for each of the datasets in each table and also the part of the human body from which the respective datasets are taken. However, apart from the documented datasets, there are some datasets that are not open to public access. Some works have used private datasets, or not provided proper citations to the datasets used, and hence could not be documented. We also eliminated duplicate entries from the datasets. Some abbreviations used in the tables, such as, 'cls.', 'seg.', 'det.', 'rec.' denote classification, segmentation, detection, and recognition, respectively. 'Inst. disc.' denotes instance discrimination.

Table 5: Summary of Magnetic Resonance Imaging (MRI) Datasets used in Self-supervised pre-training

| Dataset | Training Samples | Body Part | Source | PT Task | DS Task |
|---|---|---|---|---|---|
| MRI Datasets | | | | | |
| BraTS 2018 | ∼300 | Brain | Menze et al. (2015) | - | Tumor seg. |
| BraTS 2021 | 1251 | Brain | Menze et al. (2015) | - | Tumor Seg. |
| | | | | | Continued on next page |

**Table 5 – continued from previous page**

| Dataset | Training Samples | Body Part | Source | PT Task | DS Task |
|---|---|---|---|---|---|
| KneeMRI | 917 | Knee (ACL) | Štajduhar et al. (2017) | - | ACL Tear severity cls. |
| MRNet | 1370 | Knee | Bien et al. (2018) | Jigsaw | Knee injury diag. |
| Human Connectome Project | 1200 | Brain | Elam & Van Essen (2022) | Reconstruction | - |
| M&Ms | 375 | Heart | Campello et al. (2021) | Reconstuction | Cardiac seg. |
| ACDC | 150 | Heart | Bernard et al. (2018) | | Cardiac seg. & heart disease cls. |
| MSD (heart) | 30 | Heart | Antonelli et al. (2022) | - | Left-atrium seg. |
| ADNI | 819 | Brain | Petersen et al. (2010) | - | Alzheimer's pred. |
| PI-CAI | 1500 | Prostate | Saha et al. (2022) | Context, Inst. disc. | Prostate cancer diag. |
| Prostate158 | 158 | Prostate | Adams et al. (2022) | Context, Inst. disc. | Prostate cancer diag. |
| CRL Fetal | 81 | Fetal brain | Gholipour et al. (2017) | Reconstruction | Segmentation and analysis |
| OASIS | 434 | Brain | Marcus et al. (2007) | Reconstruction | Segmentation |
| CANDI | 103 | Brain | Kennedy et al. (2012) | Reconstruction | Segmentation |
| BigBrain | 7404 | Brain | Amunts et al. (2013) | Reconstruction | Modeling |
| UMCL Multi-rater Consensus | 30 | Brain | Lesjak et al. (2018) | - | White matter /sclerosis lesion seg. |
| MICCAI MSSeg 2016 Challenge | 53 | Brain | Commowick et al. (2016) | - | Sclerosis lesion seg. |
| Longitudinal MS Lesion Segmentation Challenge (ISBI 2015) | 82 | Brain | Carass et al. (2017) | - | Sclerosis lesion sg. |
| MRI-WHS | 60 | Heart | Gao et al. (2023) | - | Whole Heart seg. |
| MRBrainS18 | 7 | Brain | Kuijf et al. (2024) | | Brain structure seg. |
| Left Atrium (LA) dataset | 100 | Heart | Xiong et al. (2021b) | Inst. disc. | Left atrium seg. |
| OASIS3 | 2842 | Brain | LaMontagne et al. (2019) | Reconstruction | Alzheimer's det |

Table 6: Summary of Coherence Tomography (CT) Datasets used in Self-supervised pre-training

| Dataset | Training Samples | Body Part | Source | PT Task | DS Task |
|---|---|---|---|---|---|
| CT Datasets | | | | | |
| MICCAI 2015 Multi-Atlas Abdomen Labeling Challenge | 50 | Abdomen | Challenge (2015) | Inst. Disc., Few Shot seg. | Few shot seg., organ seg. |
| LUNA 2016 | 888 | Lung | Setio et al. (2017) | Context, Inst. disc. | Lung nodule seg. |
| NIH Pancreas-CT | 82 | Pancreas | Roth et al. (2016) | - | Pancreas seg. |
| LIDC-IDRI | 7371 | Lung | Armato et al. (2011) | Reconstruction, Inst. disc. | Lung nodule seg. |
| CAD-PE | 91 | Lung | Gonzalez Serrano (2019) | - | Pulmoray embolism det. & seg. |
| LiTS 2017 | 200 | Liver | Bilic et al. (2023) | Reconstruction | Liver tumor seg. |
| TCIA-COVID19 | 461 | Lung | Desai et al. (2020) | Reconstruction | Lung diseae diag. |
| C4KC-KiTS | 621 | Kidney | Heller et al. (2019) | Reconstruction | Kidney tumor seg. |
| NIH Lymph Nodes | 352 | Abdomen and Mediastinum | Roth et al. (2015) | Reconstruction | Lymphadenopathy |
| Sliver07 | 40 | Liver | Heimann et al. (2009) | - | Liver Seg. |
| | | | | | Continued on next page |

**Table 6 – continued from previous page**

| Dataset | Training Samples | Body Part | Source | PT Task | DS Task |
|---------|------------------|-----------|--------|---------|---------|
| CHAOS | 40 | Abdomen | Valindria et al. (2018) | Inst. disc., Few shot seg. | Few shot seg., Multi organ Seg. |
| 3Dircadb-01,02 | 20,2 | Liver | Soler et al. (2010) | | Hepatic tumor seg. |
| COPDGene | 947 | Lung | Regan et al. (2010) | Inst. disc. | COPD (Ephysema) det. |
| MosMed | 1110 | Lung | Morozov et al. (2020) | Inst. disc. | COVID19 severity cls. |
| DeepLesion | 10594 | multiple | Yan et al. (2018) | Instance disc. | Lesion det. |
| FLARE | 511 | Abdomen | Ma et al. (2022) | Inst. disc. | Abdominal organ seg. |
| AMOS | 500 | Abdomen | Ji et al. (2022) | - | Abdominal organ seg. |
| NSCLC Radiomics | 1265 | Lung | Aerts et al. (2019) | Inst. Disc. | Tumor seg. |
| NLST Lung Cancer | 203,099 | Lung | National Lung Screening Trial Research Team (2013) | - | Lung cancer det. |
| MIDRC-RICORD-1A | 120 | Chest | Tsai et al. (2020) | | Thoracic seg. |
| SDOCT | 154 | Eye | Tee et al. (2017) | Context based | Retinal disease diag. |
| GAMMA | 300 | Eye | Wu et al. (2023a) | - | Glaucoma grading |
| OCTA500 | 500 | Eye | Li et al. (2024b) | - | Retinal seg. |

Table 7: Summary of Ultrasound Datasets used in Self-supervised pre-training

| Dataset | Training Samples | Body Part | Source | PT Task | DS Task |
|---------|------------------|-----------|--------|---------|---------|
| Ultrasound Datasets | | | | | |
| Thyroid Nodule Segmentation | 466 | Neck | Pedraza et al. (2015) | Reconstruction | Thyroid lesion, cystic nodules, adenomas seg. |
| LEPSet | 11500 | Pancreas | Li et al. (2023c) | context base, inst. disc. | Pancreatic cancer cls. |
| BUSI | 780 | Breast | Al-Dhabyani et al. (2020) | context based, inst. disc. | Tumor seg. and cls. |
| CLUST | 86 | Liver | Luca et al. (2015) | inst. disc., context based | Liver landmark tracking |
| UDIAT | 163 | Breast | Yap et al. (2018) | context based | Tumor seg. |
| TN-SCUI 2020 | 3644 | Thyroid | Zhou et al. (2020) | Inst. disc. | segmentation |
| KiTS2019 | 210 | Kidney | Weight et al. (2019) | Inst. disc. | Tumor seg. |
| POCUS | 34 | Lung | Born et al. (2021) | Inst. disc. | Lung disease diagnosis |
| UltraSuite | - | Tongue | Eshky et al. (2018) | Contrastive | Speech diagnosis |
| PULSE | 1000 | Fetal | Drukker et al. (2021) | - | Pregnancy monitoring, diagnosis |

Table 8: Summary of Radiograph Datasets used in Self-supervised pre-training

| Dataset | Training Samples | Body Part | Source | PT Task | DS Task |
|---------|------------------|-----------|--------|---------|---------|
| Radiograph Datasets | | | | | |
| CheXpert | 224316 | Chest | Irvin et al. (2019) | Inst. disc., Context | Chest disease cls. |
| ChestX-ray8 | 108948 | Chest | Wang et al. (2017a) | Inst. disc., Context | Chest disease cls. |
| ChestX-Ray14 | 112120 | Chest | Wang et al. (2017a) | Inst. disc., Context | Chest disease cls. |
| SIIM-ACR-2019 | 15000 | Chest | Zawacki et al. (2019) | Inst. disc., Context | Pneumothorax Seg. |
| Montgomery | 138 | Chest | Jaeger et al. (2014) | - | Pneumothorax Seg. |
| MIMIC-CXR v2 | 371920 | Chest | Johnson et al. (2019) | Contrastive | Disease cls. |
| EdemaSeverity | 16108 | Chest | Liao et al. (2021a) | - | Edema Severity cls. |

Table 9: Summary of ECG Datasets used in Self-supervised pre-training

| Dataset | Training Samples | Body Part | Source | PT Task | DS Task |
|---|---|---|---|---|---|
| ECG Datasets | | | | | |
| MIT-BIH | 48 $\frac{1}{2}$-hour @ 360Hz | Heart | Moody & Mark (2001) | Reconstruction | Arythmia det. |
| PTB | 549 @ 1KHz | Heart | Bousseljot et al. (1995) | - | Cardiac disease det. |
| PTB-XL | 21837 | Heart | Wagner et al. (2022) | Contrastive | Cardiac anomaly det. |
| Tianchi ECG | 31779 @ 500 Hz | Heart | - | - | Cardiac anomaly det. |
| CinC2020 | 43,093 | Heart | Perez Alday et al. (2021) | Contrastive | Cardiac anomaly det. |
| Chapman | 10,646 | Heart | Zheng et al. (2020) | Contrastive | Cardiovascular condition det. |

Table 10: Summary of Retinal Fundus Image Datasets used in Self-supervised pre-training

| Dataset | Training Samples | Body Part | Source | PT Task | DS Task |
|---|---|---|---|---|---|
| Retinal Fundus Image Datasets | | | | | |
| EyePACS | 35126 | Eye | Dugas et al. (2015) | Inst. disc. | Diabetic Retinopathy det. |
| APTOS 2019 | 3662 | Eye | Karthik (2019) | Inst. disc. | Blindness det. |
| Messidor | 1200 | Eye | Decencière et al. (2014) | | Diabetic retinopathy & risk of macular edema |
| IchallengeAMD | 1200 | Eye | iChallenge (2018) | Inst. disc., Context | age-related macular degeneration |
| IchallengePM | 1200 | Eye | iChallenge (2018) | Inst. disc., Context | pathological myopia |
| Isfahan MISP | 59 | Eye | Hajeb Mohammad Alipour et al. (2012a) | reconstruction | retinography-angiography registration |
| DRIVE | 40 | Eye | Staal et al. (2004) | - | Blood vessel seg. and optic disc loc. |
| DRIONS | 110 | Eye | Carmona et al. (2008) | - | optic disc seg. |
| IDRiD | 516 | Eye | Porwal et al. (2018) | - | Diabetic retinopathy cls. and fovea loc. |
| REFUGE | 800 | Eye | Orlando et al. (2020) | - | glaucoma det. |
| ADAM | 400 | Eye | Fang et al. (2022) | Inst. disc. | age-related macular degeneration |
| DRISHTI-GS | 101 | Eye | Sivaswamy et al. (2014) | Inst. disc. | glaucoma det. |
| RFMiD | 3200 | Eye | Pachade et al. (2021) | Context | fundus disease cls. |
| PALM | 1200 | Eye | Fu et al. (2019) | Context | disc and atrophy segmentation |
| Fundus FFA | 70 | Eye | Hajeb Mohammad Alipour et al. (2012b) | Context | Diabetic retinopathy cls. |

Table 11: Summary of Endoscopic Datasets used in Self-supervised pre-training

| Dataset | Training Samples | Body Part | Source | PT Task | DS Task |
|---|---|---|---|---|---|
| Endoscopy Datasets | | | | | |
| CVC-ColonDB | 300 images | Colon | Bernal et al. (2012) | Context based | Polyp seg. |
| CVC-ClinicDB | 612 images | ” | Bernal et al. (2015) | Context based | Polyp seg. |
| | | | | | Continued on next page |

**Table 11 – continued from previous page**

| Dataset | Training Samples | Body Part | Source | PT Task | DS Task |
|---|---|---|---|---|---|
| ETIS Larib | 196 videos | " | Bernal et al. (2017) | Context based | Polyp seg. |
| CVC-VideoClinicDB | 40 videos | " | Bernal et al. (2018) | - | Polyp seg. and det. |
| Kvasir Seg | 1000 images | " | Jha et al. (2020) | Context based | Polyp seg. |
| HyperKvasir | 373 videos | " | Borgli et al. (2020) | - | Polyp seg. and det. |
| LD-PolypDB | 160 videos | " | Ma et al. (2021c) | Non-contrastive siamese | Polyp seg. and det. |

Table 12: Summary of Histopathological Image Datasets used in Self-supervised pre-training

| Dataset | Training Samples | Body Part | Source | PT Task | DS Task |
|---|---|---|---|---|---|
| Histopathology Datasets | | | | | |
| DSB 2018 | 4470 | Nucleus | Goodman et al. (2018) | denoising / context based | Nucleus seg. |
| BBBC 004 | 880 | Nucleus | Ljosa et al. (2012) | " | Nucleus seg. |
| BACH | 400 | Breast | Polónia et al. (2019) | | Breast cancer cls. and seg. |
| BreakHisv1 | 9109 | Breast | BreakHis (2018) | | breast cancer cls. |
| NCT-CRC-HE-100K | 100K | Colorectal | Kather et al. (2018) | Inst. disc. | colorectal cancer tissue cls |
| Gleason2019 | 333 | Prostate | Nir et al. (2018) | - | prostate cancer cls. |
| DigestPath2019 | 687 | Intestine, colon | Da et al. (2022) | - | early-stage colon tumors seg. |
| BreastPathQ | 3700 | Breast | Petrick et al. (2021) | - | cancer cellularity |
| Camelyon16 | 271 | Breast | Ehteshami Bejnordi et al. (2017) | - | Breast cancer metastasis det. |
| PAIP | 100 | Liver | Kim et al. (2021) | | Liver cancer seg. & Viable tumor burden estimation |
| TissueNet | - | Cervix | Greenwald et al. (2022) | | Cervical epithelial lesion cls. |
| Cell Tracking Challenge | - | - | Ulman et al. (2017) | - | Cell tracking and seg. |
| Electron Microscopy | 165 | Cell | Lucchi et al. (2013) | - | Mitochondria seg. |
| TNBC | 50 | Breast | Naylor et al. (2019) | - | Cell seg. |
| MoNuSeg | 30 | - | Kumar et al. (2017) | Context | Nuclei seg. |
| CoNSeP | 41 | Colon & rectum | Graham et al. (2019) | Context | Nuclei seg. |

Table 13: Summary of Skin Image Datasets used in Self-supervised pre-training

| Dataset | Training Samples | Body Part | Source | PT Task | DS Task |
|---|---|---|---|---|---|
| Skin Images | | | | | |
| ISIC 2020 | 33126 | Skin | Rotemberg et al. (2021) | Inst. disc., Reconstruction | Melanoma cls. |
| ISIC 2018 | 2594 | Skin | Codella et al. (2019) | Reconstruction | Melanoma cls. and segmentation |
| ISIC 2017 | 2750 | Skin | Codella et al. (2018) | " | Melanoma segmentation |
| HAM10000 | 10015 | Skin | Tschandl et al. (2018) | - | Skin lesion cls. and seg. |

**Summary:** In this section, we compiled a complete summary of the public datasets used in the studies explored in this review. The dataset summary will facilitate the readers to figure out the suitable datasets

according to the compute capacities at their disposal, diversify their work using different modalities of medical images and develop robust self-supervised frameworks.

In the following section, we compile the performance metrics of sveral foundational and notable SSL frameworks on several benchmark image and video datasets, enabling the readers to easily compare the several exisitng frameworks and determine a framework suitable for their application or research domain based on their performance.

# 5 Comparison of self-supervised frameworks on benchmark datasets

In this survey, we discuss a plethora of frameworks. However, to truly assess the effectiveness of the frameworks, we need to look into the performance of those frameworks on a few benchmark datasets. Although we have tried to provide comparisons on the same downstream or target datasets for all frameworks, it is to be noted that the pre-training conditions may differ in some.

## 5.1 Comparison of Image-based SSL frameworks

In this subsection, we compare the image-based SSL frameworks based on the performance on (1) the ImageNet1K classification task, (2) Classification, detection, and segmentation tasks on the PASCAL VOC dataset, and (3) Object detection and Instance segmentation tasks in MS COCO dataset. In some frameworks, the version of the PASCAL VOC dataset used for fine-tuning in the downstream task varies between VOC2007, VOC2012 (†), or VOC07+12 (∗), and has been discriminatively indicated in the table. The results in Table 14 are either obtained from the original papers or from papers that are mentioned in the table (Table 14) and have been compared to those works in their respective manuscripts, that is, the results are cross-verified.

Besides that, we often find that the nature of the backbone encoder, or the number of pre-training epochs do not match for all the frameworks. As the domain has evolved, so has the choice of hyperparameters to obtain better performance on the benchmark datasets. However, we have done our best to document every detail of notable work done right from the advent of SSL with works like Agrawal et al. (2015) or Pathak et al. (2016) to the recent works like SMoG (Pang et al., 2022b), I-JEPA (Assran et al., 2023a), ConvNextv2 (Woo et al., 2023), etc.

All the frameworks mentioned in Table 14, are pre-trained on ImageNet1K (Deng et al., 2009) dataset for varying number of epochs as per their respective needs. We have attempted to present a comparative analysis using both linear classification (using a frozen encoder) and fine-tuning Top-1 accuracy on the ImageNet-1K dataset. We also present our findings on the PASCAL VOC dataset for object classification (mAP), detection (AP50) and segmentation (mIOU) tasks. Unless otherwise mentioned, the default dataset for PASCAL VOC tasks is VOC2007. There are a few frameworks that opt for the VOC2012 and VOC07+12 versions of the dataset for finetuning. We also report the bounding box AP ($AP_{bb}$) and mask AP ($AP_{mk}$) in the MS COCO dataset.

Table 14: Comparison of performance of a few notable image-based self-supervised frameworks. † and ∗ indicate the use of PASCAL VOC2012 and PASCAL VOC07+12, respectively. ‡ and § indicate that the encoder was pre-trained for 400 and 200 epochs, respectively.

| Frameworks | Encoder | ImageNet | | PASCAL VOC07 | | | MS-COCO | |
|---|---|---|---|---|---|---|---|---|
| | | Lin. | FT | Cls. | Det. | Seg. | Det. | ISeg. |
| *Context based frameworks* | | | | | | | | |
| Agrawal et al. (2015) | | - | - | 54.2 | 43.9 | - | - | - |
| Context (Doersch et al., 2015) | | 31.7 | 45.6 | 65.3 | 51.1 | 38.4 | - | - |
| Pathak et al. (2016) | | 21.0 | - | 56.5 | 44.5 | 29.7 | - | - |
| Wang & Gupta (2015) | | - | 38.8 | 63.1 | 47.4 | 35.4 | - | - |
| Colorization (Zhang et al., 2016) | AlexNet | 31.5 | 40.7 | 65.6 | 46.9 | 35.6 | - | - |
| Counting (Noroozi et al., 2017) | | 34.3 | - | 67.7 | 51.4 | 36.6 | - | - |
| Continued on next page | | | | | | | | |

**Table 14 – continued from previous page**

| Frameworks | Architecture | ImageNet | | PASCAL VOC | | | MS-COCO | |
|---|---|---|---|---|---|---|---|---|
| | | Lin. | FT | Cls. | Det. | Seg. | Det. | ISeg. |
| ColorProxy (Larsson et al., 2017) | | - | - | 65.9 | | 38.4 | - | - |
| Jigsaw (Noroozi & Favaro, 2016) | | 34.0 | 45.3 | 67.6 | 53.2 | 37.6 | - | - |
| RotNet (Gidaris et al., 2018) | | 38.7 | 50.0 | 72.97 | 54.4 | 39.1 | - | - |
| *MIM based frameworks* | | | | | | | | |
| BEiT (Bao et al., 2021) | | 56.7 | 83.2 | - | - | - | 50.1 | 43.5 |
| mc-BEiT (Li et al., 2022e) | | | | | | | 50.1 | 43.1 |
| iBOT (Zhou et al., 2022a) | | 79.5 | 84.0 | - | - | - | 51.2 | 44.2 |
| MAE (He et al., 2022) | | - | 83.6 | - | - | - | 50.3 | 44.9 |
| LocalMIM (Wang et al., 2023b) | ViT-B | - | 84.0 | - | - | - | 50.7 | 44.9 |
| SimMIM (Xie et al., 2022) | | 56.7 | 83.8 | | | | 50.4 | 44.4 |
| CAE (Chen et al., 2023c) | | 71.4 | 83.9 | | | | 52.9 | 45.5 |
| BootMAE (Dong et al., 2022) | | 66.1 | 84.2 | | | | 48.5 | 43.4 |
| ConvNext v2 (Woo et al., 2023) | | - | 84.9 | | | | 52.9 | 46.6 |
| *Clustering-based frameworks* | | | | | | | | |
| DeepCluster (Caron et al., 2018) | AlexNet | 41.0 | - | 73.7 | 55.4 | 45.1 | - | - |
| UIC (Chen et al., 2020d) | AlexNet | 41.6 | - | 75.9 | 54.9 | 45.9 | - | - |
| ODC (Zhan et al., 2020) | AlexNet (ResNet50) | 41.4 (55.7) | - | (78.2) | - | - | - | - |
| LA (Zhuang et al., 2019a) | ResNet50 (AlexNet) | 60.2 (42.4) | | | 53.5 | | | |
| CLIM (Li et al., 2020b) | ResNet50 | 75.5 | - | 82.8 | - | - | 41.8 | 37.7 |
| CoKe (Qian et al., 2021a) | ResNet50 | 76.4 | - | 83.2 | | | 40.9 | 37.2 |
| *Instance Discrimination frameworks* | | | | | | | | |
| CPC (van den Oord et al., 2018) | ResNetv2 101 | 48.7 | | | | | | |
| PIRL (Misra & van der Maaten, 2019) | ResNet50 | 63.6 | - | 81.1 | 80.7* | | | |
| MoCo (He et al., 2020) | ResNet50 (RN50w4×) | 60.6 (68.6) | - | - | 81.5* | | 40.8 | 36.9 |
| SimCLR (Chen et al., 2020b) | ResNet50 (RN50w4×) | 69.3 (76.5) | 89.0 (93.2) | 86.6 | 79.4 | | 38.5 | 34.8 |
| MoCov2 (Chen et al., 2020e) | ResNet50 | 71.1 | | 82.5 | 82.4†§ | | 39.8 | 36.1 |
| CPCv2 (Hénaff et al., 2020) | ResNet50 (ResNet161*) | 63.8 (71.5) | 85.3 (90.1) | (76.6) | | | | |
| InfoMin (Tian et al., 2020b) | ResNet50 | 73.0 | 91.1 | | 82.7 | | 42.5 | 38.4 |
| SimCLRv2 (Chen et al., 2020c) | ResNet50 (RN152 w3×+SK) | 71.7 (79.8) | | | | | | |
| DenseCL (Wang et al., 2020) | ResNet50 | | | | 82.8* | 69.4* | 40.3 | 36.4 |
| MoCov3 (Chen et al., 2021d) | ViT-B | 76.7 | 83.2 | - | - | - | 47.9 | 42.7 |
| SSL-HSIC (Li et al., 2021d) | ResNet50 (RN200w2×) | 74.8 (79.6) | | 84.1 | 76.0† | | 41.3 | 36.8 |
| PCL (Li et al., 2021b) | ResNet50-MLP | 67.6 | | 85.4 | 71.7 (78.5*) | | | |
| MUGS (Zhou et al., 2022c) | ViT-B | 80.6 | 84.3 | | | | 49.8 | 43.0 |
| SeLa (Asano et al., 2020b) | ResNet50 (AlexNet) | 61.5 | | 77.2 | 59.2 | 45.7 | | |
| SwAV (Caron et al., 2020) | ResNet-50 | 75.3 | | 88.9 | 82.6* | | 42.1 | |
| SMoG (Pang et al., 2022b) | ResNet50 (RN50w4×) | 76.4 (79.0) | | 85.01† | 76.2† | | 40.1 | 36.9 |
| MDRA (Cheng et al., 2023) | ResNet50 | 71.9 | | | | | 40.2 | 36.0 |
| *Dimension-Contrastive frameworks* | | | | | | | | |
| BYOL (Grill et al., 2020) | ResNet50 (RN200w2×) | 74.3 (79.6) | | 85.4 | 77.5† | 76.3† | 38.4 | 34.9 |
| I-JEPA (Assran et al., 2023a) | ViT-B | 72.9 | | | | | | |
| Barlow Twins (Zbontar et al., 2021) | ResNet50 | 73.2 | | 86.2 | 82.6* | | 39.2 | 34.3 |
| VICReg (Bardes et al., 2022a) | ResNet50 | 73.2 | | 86.6 | 82.4* | | 39.4 | 36.4 |
| W-MSE (Ermolov et al., 2021) | ResNet50 | 72.56‡ | | | | | | |
| Zero-CL (Zhang et al., 2022d) | ResNet50 | 72.6‡ | | | | | | |
| *Non-Contrastive frameworks* | | | | | | | | |
| SimSiam (Chen & He, 2020) | ResNet50 | 71.3 | | | 48.5§ | | 39.2§ | 34.4§ |
| DINO (Caron et al., 2021) | ViT-B | 78.2 | 83.6 | - | - | - | 50.1 | 43.4 |
| ReSSL (Zheng et al., 2021b) | ResNet50 (+5crops) | 69.9§ (74.7) | | | | | | |
| OBoW (Gidaris et al., 2021) | ResNet50 | 73.8§ | | 89.3 | 82.9* | | | |
| MSN (Assran et al., 2022) | ViT-L (ViT-B) | 80.7 | 83.4 | | | | | |
| CrOC (Stegmüller et al., 2023) | ViT-S | | | | 70.6 | | | |

## 5.2 Comparison of different frameworks on Video benchmark

In this subsection, we present the comparison of several SSL frameworks on benchmark video datasets. Similar to the image-based frameworks, the base encoder architecture choice differs between different frameworks, as does the choice of pre-training dataset. For UCF101, HMDB51, and Kinetics400 datasets, we report the action recognition accuracy. For UCF101 and HMDB51, we report the average accuracy over the 3 splits. Unless otherwise mentioned, the values mentioned in Table 15, are obtained after finetuning on the respective datasets.

Table 15: Comparison of performance of a few notable video-based self-supervised frameworks. The results on UCF-101 and HMDB-51 were obtained after finetuning. The results on the Kinetics dataset are obtained by linear evaluation unless otherwise mentioned. $lin$ indicates Linear Evaluation. † indicates that the use of Kinetics600 instead of Kinsteics400 for finetuning and evaluation.

| Frameworks | Encoder | Pretrain Data | UCF-101 | HMDB51 | Kinetics 400 |
|---|---|---|---|---|---|
| *Context based frameworks* | | | | | |
| Shuffle & Learn (Misra et al., 2016) | CaffeNet | UCF-101 | 50.9 | 19.8 | |
| 3DRotNet (Jing et al., 2019) | 3D RN18 | Kinetics600 | 76.6 | 47.0 | |
| VCOP (Xu et al., 2019) | R(2+1)D-18 | Kinetics | 72.4 | 30.9 | |
| VidCloze (Luo et al., 2020) | C3D | | 68.5 | 32.5 | |
| OOO (Fernando et al., 2017) | AlexNet | UCF101, HMDB51 | 60.0 | 32.4 | |
| SkipClip (El-Nouby et al., 2019) | 3D RN18 | UCF-101 | 64.4 | | |
| (Jenni et al., 2020) | 3D RN18 [R(2+1)D-18] | Kinetics600 [UCF-101] | 79.3 [81.6] | 49.8 [46.4] | |
| CPNet (Liang et al., 2022) | R(2+1)D-18 | UCF-101 [Kinetics400] | 81.8 [83.8] | 51.2 [57.1] | |
| TransRank (Duan et al., 2022) | R(2+1)D-18 | Kinetics200 | 90.7 | 64.2 | |
| *MIM based frameworks* | | | | | |
| BEVT (Wang et al., 2022c) | Video-SWIN | ImageNet1K+ Kinetics400 | | | 81.1 |
| VideoMAE (Tong et al., 2022) | ViT-B | Kinetics400 | 96.1 | 73.3 | 81.5 |
| VideoMAEv2 (Wang et al., 2023e) | ViT-H | Kinetics400 | 99.6 | 88.1 | 88.6 |
| AdaMAE (Bandara et al., 2023) | ViT-B | Kinetics400 | | | 81.7 |
| OmniMAE (Girdhar et al., 2023) | ViT-B (ViT-H) | ImageNet1K+ SSv2 | | | 80.6 (85.4) |
| *Instance discrimination based frameworks* | | | | | |
| CoCLR (Han et al., 2020) | S3D | UCF-101 [Kinetics400] | 87.1 [90.6] | 58.7 [62.9] | |
| CVRL (Qian et al., 2021b) | R3D-152w2× | Kinetics600 | 93.9 | 69.9 | 72.9 |
| SCVRL (Dorkenwald et al., 2022) | MViT-B | Kinetics400 | 89.0 | 62.6 | |
| FAME (Ding et al., 2022) | R(2+1)D [I3D] | Kinetics400 | 84.8 [88.6] | 53.5 [61.1] | |
| HDC (Zhang & Crandall, 2022) | R(2+1)D-10 | Kinetics400 | 76.8 | 40.0 | |
| SeCo (Yao et al., 2020) | 3D RN18 | Kinetics400 | 88.26 | 55.55 | $50.81^{lin}$ |
| VINCE (Gordon et al., 2020) | ResNet50 | Kinetics400 | | | $49.1^{lin}$ |
| VCLR (Kuang et al., 2021) | R2D-50 | Kinetics400 | 85.6 | 54.1 | $64.1^{lin}$ |
| *Dimension Contrastive frameworks* | | | | | |
| V-JEPA (Bardes et al., 2024) | ViT-L | Kinetics400 [VideoMix2M] | | | 78.7 [79.1] |
| *Non-contrastive frameworks* | | | | | |
| BraVe (Recasens et al., 2021) | R3D50 | Kinetics600 | 95.1 | 74.3 | $68.1^{lin†}$ |

# 6 Challenges and Limitations

In this paper, we discuss several SSL methods and their derivatives on natural image data and medical imaging modalities. Although SSL has shown great potential, particularly in medical image analysis by reducing the dependency on labelled data and improving feature representations, it also exhibits several limitations as follows.

**Training collapse scenario:** SSL methods in medical imaging suffer from several issues like complete collapse, dimensional collapse of representation, shortcut learning in context-based tasks, etc. This often refers to the phenomenon when the network learns to detect low-level supervisory signals like edges, artefacts, blank regions, etc. to minimize the loss.

**Limited data:** As medical data is hard to collect and annotate by expert personnel, we often find limited publicly available medical data. From the discussion of works on medical imaging modalities, we can observe that a large number of those works are on MR and CT images, but applications of SSL to other medical imaging modalities are limited.

**Domain-specific generalizability:** Under the common terminology of medical data, we encompass diverse data types. The diversity in the medical image analysis domain ensures that any attempts at multimodal applications of SSL on medical images are hindered due to the domain gap and the difference in the target anatomy of each imaging modality.

**Limited freedom of data augmentation:** The data augmentations used for natural images and videos cannot always be used for all medical imaging modalities. Improper design of the data augmentation pipeline can have two effects: (a) affect the supervisory signal in the pretext task, consequently influencing downstream performance, and (b) lead to the collapse of representations in the pretext task of SSL. For instance, in skin lesion images, the colour of the lesion plays an important role in diagnosing the lesion. In abdominal MR or CT scans, the orientation of the organs like the left and right kidneys are crucial for downstream applications.

**Quality of supervisory signal:** In self-supervised learning, the supervisory signal in the pretext task, controls the quality of representations. However, designing a suitable pretext task which captures the underlying structure in the data and finally learns useful representations can be challenging in SSL. Often the representations learnt in the pretext task encode information which are of no semantic relevance. Consequently, the downstream task performance in SSL is affected.

Medical images can be quite complex, considering the diverse anatomy covered in some imaging modalities. Also, medical images are often affected by noise and artefacts from the imaging equipment, which can hinder the learning of representations. The quality of supervisory signals and consequent performance on the downstream tasks varies based on the target anatomy and also requires modality-wise hyper-parameter fine-tuning in the pretext task.

**Lack of benchmarking:** Medical data collected from different sources are often captured using different instruments. A proper benchmarking protocol to evaluate methods trained on different data following clinical norms is necessary, which is absent in the current scenario. Consequently, this leads to poor interpretability of the SSL frameworks.

## 7 Conclusion and Future Directions

In this survey, we take a different approach to reviewing the work done in the domain of self-supervised learning. Firstly, we divided the works into several categories according to the approaches taken. We then try to further categorise each into finer subcategories. This allows us to learn about the different avenues of research pursued in the past years and the research directions currently being explored. The finer discussions in each subcategory allow us to better understand the differences between the frameworks or approaches.

From this survey, we have noticed that the several pieces of work done in the last few years on masked image modelling show the potential of the approach in visual self-supervised learning. Contrastive learning approaches like SimCLR and MoCov2, and non-contrastive approaches like BYOL and SimSiam are the most popular and were also adopted for several medical applications.

Though we have noticed that a majority of self-supervised learning applications used MRI or CT scans, this is mainly due to the easy availability of benchmark datasets for the MR or CT modality, recently we

have observed that SSL has been applied to many other medical imaging modalities as well, indicating the acceptability and adaptability of SSL in the current image analysis domain.

In future, several measures can be attempted which can extend the scope of SSL based methods in medical image analysis. Firstly, to deal with the issue of data scarcity and imbalance, the incorporation of generative methods into the conventional SSL pipeline to learn the data distribution along with representation could be a way forward. In addition, future research on the incorporation of incremental/continual learning and federated learning principles in the current SSL frameworks can also help to mitigate the data scarcity issue. Secondly, further investigation to understand the behaviour of SSL frameworks in handling data imbalance could open a separate research avenue. Even with large batch sizes, assuming uniform sampling without prior knowledge of the underlying data distribution, the imbalance may persist. Hence, the representations learnt in the pretext task will be biased towards the majority classes and will affect downstream performance. Moreover, SSL frameworks, which can learn domain-generalised anatomy-aware features, can utilise data from multiple medical imaging modalities to learn better representations. For instance, an SSL framework pretrained on both MR and CT abdominal datasets can utilise a larger pool of data with similar anatomical content and learn robust and more generalized representations. SSL frameworks also often employ methods that learn semantic features from the data. However, not all features are suitable for the downstream task. Hence, it is important to learn task-specific representations to enhance performance on the downstream tasks. Consequently, this topic requires further research to shed light on this matter. Furthermore, medical data are often sensitive and its privacy concerns must be addressed while handling medical data in SSL frameworks.

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
