# OpenReview forum: "Self-Supervised Visual Representation Learning for Medical Image Analysis: A Comprehensive Survey"
_TMLR — Accepted by TMLR_

### Review · Reviewer_otY9 · 2024-06-03

**Summary Of Contributions:**

This paper presents a literature review of self-supervised learning methods for medical image analysis. Compared to other reviews, the authors argue that their work provides more detailed analysis of the included papers and a better categorization of studies. They provide a review of the methods as well as datasets and benchmarks used in existing studies.

**Audience:**

Yes

**Claims And Evidence:**

Yes

**Requested Changes:**

Please see list above. I believe the scope of the paper should either be expanded to SSL in general, or focus on medical imaging applications only. Otherwise, the content doesn't really match what the title portrays.

**Strengths And Weaknesses:**

Strengths:
- The authors provide a good general framework for the categorization of the methods.
- For every included method, they provide a description of its methodology.
- They also summarize the datasets that were considered by existing studies and benchmarks.
- They also summarize the performance of different SSL frameworks on benchmark datasets.

Weaknesses:
- The review article is too long. The first 22 pages (almost) are dedicated to discussing the different SSL method. They only begin to discuss applications for medical imaging from page 23 onwards.
- Additionally, the transition from non-medical studies to medical oriented ones seems abrupt. They do so again when discussing the datasets and benchmarks, and then again when discussing the performance of the different models on the benchmark tasks. While I do appreciate the level of details and summaries provided for the benchmarks, they are not relevant to the scope of the paper (medical image analysis), and hence are a bit distracting. They would be more relevant for a general SSL literature review.
- The quality of the paper presentation requires improvement, specifically the language requires significant editing.
- There is an imbalance in terms of lengths of discussion, for example the ultrasound section is very brief.
- How did the authors select papers to be included in the review?

---

> ### Author Response · Authors · 2024-06-07
> **Reply to Reviewer otY9**
>
> Thank you very much for taking the time to review my paper and for your thoughtful and constructive feedback. We are particularly grateful for your kind comments on the strengths of our work. Your positive feedback is very encouraging and greatly appreciated.
>
> >Strengths:
> >- The authors provide a good general framework for the categorization of the methods.
> >- For every included method, they provide a description of its methodology.
> >- They also summarize the datasets that were considered by existing studies and benchmarks.
> >- They also summarize the performance of different SSL frameworks on benchmark datasets.
>
> We thank the reviewer for highlighting the strengths of our work.
> ***
> ***
> >Weaknesses:
> >- The review article is too long. The first 22 pages (almost) are dedicated to discussing the different SSL method. They only begin to discuss applications for medical imaging from page 23 onwards.
>
> We understand that the first 22 pages of the article focus extensively on discussing different SSL methods before transitioning to applications in medical imaging. We intended to provide a comprehensive overview of the SSL methods to ensure that readers have a solid understanding of these techniques before going through their specific applications in medical imaging so that this review serves for both the foundational works and their derivatives, as well as, for the applications in medical image analysis. If the reviewer suggests, we can delete Sec. 1.2, and merge sections 2.1.2 & 2.1.4, and 2.1.6 & 2.1.7. That will reduce the number of pages dedicated to discussing the different SSL methods.
>
> >- Additionally, the transition from non-medical studies to medical oriented ones seems abrupt. They do so again when discussing the datasets and benchmarks, and then again when discussing the performance of the different models on the benchmark tasks. While I do appreciate the level of details and summaries provided for the benchmarks, they are not relevant to the scope of the paper (medical image analysis), and hence are a bit distracting. They would be more relevant for a general SSL literature review.
>
> Thank you for the constructive feedback. We also realise that the transitions from non-medical studies to medical-oriented ones appear abrupt. We have revised the transitions between non-medical and medical studies to ensure a more seamless flow. This involves adding bridging paragraphs and contextual explanations between consecutive sections. **We have marked the changes to the manuscript in blue**.
> We also understand that some sections, such as the discussions on datasets, benchmarks, and performance of different models, might seem less relevant to the primary focus on medical image analysis. As exhaustive reviews on SSL are not easily available, we thought detailed descriptions and a fine-grained stratification would be beneficial for the readers. If the reviewer suggests, we will definitely remove these sections.
> Alternatively, we can also think about changing the title of the review to a more suitable one, to widen the scope of this review, as per the reviewer's suggestion.
>
> >- The quality of the paper presentation requires improvement, specifically the language requires significant editing.
>
> We appreciate your insights and have thoroughly reviewed and edited the language of the manuscript to improve clarity and readability.
>
> >- There is an imbalance in terms of lengths of discussion, for example the ultrasound section is very brief.
>
> We acknowledge that the ultrasound section is relatively brief. We have expanded this section to provide a more comprehensive analysis so that it aligns better with the other sections. We have also expanded the Echocardiography and Skin images sections. We have also added newly found datasets to the dataset table for ultrasound. **The changes to the manuscript are marked in blue**.
>
> >- How did the authors select papers to be included in the review?
>
> Thank you for your question regarding the selection of papers for inclusion in the review. We intended to write a review consisting of two major parts, one covering the foundation of SSL, and another covering the applications of SSL in medical image analysis. We accordingly divided the broad spectra of SSL algorithms into sub-categories based on different algorithmic principles that are coming up. We chronologically selected papers based on a comprehensive literature search using databases such as PubMed, IEEE Xplore, Google Scholar, etc. We have prioritised recent studies and research works which are highly cited in this domain. We used the same strategy for the other part of our review which presents applications of SSL in the field of medical image analysis.
>
> We have added a separate "Methodology" section in Sec. 1.4 in the revised manuscript.

---

> ### Comment · Action_Editor_yxMC · 2024-06-27
> **Please check the authors response and start the discussion**
>
> Dear reviewer otY9,
>
> Thank you for reviewing the submission!
>
> Now the authors have submitted their responses to your review comments, please take a look at the responses and see if they have addressed your concerns. Please do not hesitate to start the discussion if you have any questions/concerns.
>
> Best,
>
> AE

---

> ### Author Response · Authors · 2024-07-07
> **Reply to Reviewer otY9**
>
> Dear reviewer otY9,
>
> Thank you for your feedback on our work!
>
> We have responded to your comments and made some changes as per your suggestions. We are waiting for your reply and further suggestions / questions.
>
> Best,
>
> Authors

---

> > ### Comment · Reviewer_otY9 · 2024-07-13
> > **Response to authors**
> >
> > Thank you for your responses. I am happy with the updated manuscript.

---

### Review · Reviewer_udZB · 2024-06-10

**Summary Of Contributions:**

In their paper "Self-supervised visual representation learning for medical image analysis: a comprehensive survey", the authors aim to give a thorough review of self-supervised learning (SSL) approaches in computer vision, and applications for medical imaging.

**Audience:**

Yes

**Broader Impact Concerns:**

No concerns.

**Claims And Evidence:**

Yes

**Requested Changes:**

MAJOR COMMENTS

* The paper does not cite the SSL "cookbook" by the META team (https://arxiv.org/abs/2304.12210) which is another large survey.

* The papers includes a huge amount of references, but does not say anything about what it aimed to achieve or about how they selected the references. Was the aim to include ALL papers using SSL in medical image analysis? If yes, how did the authors search the literature? Medical meta-analysis papers usually explain how the searched the literature (which queries were used in which search engines etc.), and I think it would be good to add a Methods section explaining that. If that was not the aim, then what was the aim? Why did the authors select these specific 700 papers?

* section 1.3: this section caught my attention as likely having been either written or edited by ChatGPT. Given the dangers of using ChatGPT for writing reviews (as it can confabulate references etc.), I would like to ask the authors to insert some declaration on LLM usage. Did they use LLMs at all? If yes, how can they be sure that nothing was confabulated? E.g. if they only used it for minor editing, then it's okay. Perhaps this can be added into the Methods.

* Section 2.3.2: self-distillation methods like DINO are listed as "non-contrastive" whereas distillation methods like SimSiam are listed as "dimension-contrastive". Why is that? I did not find the rationale for this classification, or perhaps overlooked it. Since this affects the whole taxonomy (Figures 1/2 etc), I would suggest that the authors motivate their taxonomic choices better and provide rationale for their choices.

* It would be nice to have some explicit discussion / overview on the _uses_ of SSL in medical image analysis. What are the downstream tasks for which SSL makes sense? Is it mostly classification? Or something else too? Some recent papers used SSL for 2D visualisation -- e.g. https://arxiv.org/abs/2402.14566, based on https://openreview.net/forum?id=nI2HmVA0hvt. This could be added as another use-case of SSL approaches.


MINOR COMMENTS

* page 2: "We also discuss the work DeSa (1993) which can be ..." -- this sentence is unclear as it gives two references and it's unclear which one refers to which.

* sometimes \citep and \citet are not used appropriately, e.g. the first line of section 1.1 uses \citet but should use \citep (or alternatively remove the words "the authors").

* page 3: DINO (Caron et al 2021) is listed in the paragraph about transfer learning. Why is that? Isn't DINO a self-distillation framework that can be called "self-supervised only"?

* Section 3.6.2: this is the only subsection in section 3 where I know the literature a little bit, and I found some clear omissions, e.g. https://www.biorxiv.org/content/10.1101/2023.08.22.554251v1 or https://arxiv.org/abs/2212.04690.

**Strengths And Weaknesses:**

Strengths: The paper is really comprehensive. The main text takes 40+ pages and the list of references takes additional 60+ pages (!). The references are not numbered but assuming ~12 references per bibliography page, the paper probably has around 700 citations.

Weaknesses: I found it hard to learn much from this paper. For the most part, the paper looks like a briefly annotated list of references. As such, it can certainly be useful and can help navigate the literature. But having page after page listing dozens (if not hundreds) of papers that used SSL for e.g. MRI analysis with only several words per paper can hardly provide any insight.

Beyond an impressive attempt to give a comprehensive taxonomy of SSL methods (Figures 1/2), the paper does not contain any figures and, in my opinion, can't really be used to learn about SSL.

Overall, despite these clear inherent limitations, it is an impressive survey, which I think can be published in TMLR almost as is.

---

> ### Author Response · Authors · 2024-06-14
> **Reply to Reviewer udZB - Part 1**
>
> Thank you for reviewing our paper and for your thoughtful and constructive feedback. We are grateful for the kind comments on the strengths of our work. Your positive feedback is very encouraging and greatly appreciated.
>
> >Strengths And Weaknesses:
> >
> >Strengths:
> >The paper is really comprehensive. The main text takes 40+ pages and the list of references takes additional 60+ pages (!). The references are not numbered but assuming ~12 references per bibliography page, the paper probably has around 700 citations.
>
> We thank the reviewer for recognising the strength of our work.
>
> >Weaknesses: I found it hard to learn much from this paper. For the most part, the paper looks like a briefly annotated list of references. As such, it can certainly be useful and can help navigate the literature. But having page after page listing dozens (if not hundreds) of papers that used SSL for e.g. MRI analysis with only several words per paper can hardly provide any insight.
>
> Thank you for your valuable comment. We intended to provide a comprehensive account of the foundational works and their derivatives in the field of SSL, which will provide the readers with a broad overview of the works done. We then proceed to provide a detailed overview of the applications of SSL frameworks on different modalities in the domain of medical image analysis.
>
> The comprehensive review of the works done in medical image analysis, say, on MRI data, also provides a clear account of the methods and datasets frequently chosen by researchers. This review aims to give researchers an organized set of information to begin with. We intended to provide a limited discussion on each work primarily to limit the length of the manuscript. However, we have tried our best to summarise the approaches undertaken in those few lines to give a basic idea about the same, thereby, providing researchers an idea about the trends in the current SSL research landscape.
>
> >Beyond an impressive attempt to give a comprehensive taxonomy of SSL methods (Figures 1/2), the paper does not contain any figures and, in my opinion, can't really be used to learn about SSL.
>
> Thank you for your comment. We agree that except for the figures depicting the comprehensive taxonomy of SSL, the manuscript did not contain any figures. We believe that the huge spectra of derivatives of the foundational models are too dense to be represented using a single diagram, hence we have added diagrammatic representations of a few foundational frameworks to provide an abstract idea about the information flow in Fig. 3, 4, and 5, in the revised manuscript.
>
> >Overall, despite these clear inherent limitations, it is an impressive survey, which I think can be published in TMLR almost as is.
>
> Thank you for your positive feedback and for acknowledging the value of our survey. We sincerely thank you for recommending the work for publication in TMLR.
>
> >Requested Changes:
>
> >MAJOR COMMENTS
>
> >The paper does not cite the SSL "cookbook" by the META team (https://arxiv.org/abs/2304.12210) which is another large survey.
>
> Thank you for the constructive suggestion. We have now added a discussion about the SSL Cookbook in the Introduction section. We have highlighted the changes in yellow.
>
> >The papers includes a huge amount of references, but does not say anything about what it aimed to achieve or about how they selected the references. Was the aim to include ALL papers using SSL in medical image analysis? If yes, how did the authors search the literature? Medical meta-analysis papers usually explain how the searched the literature (which queries were used in which search engines etc.), and I think it would be good to add a Methods section explaining that. If that was not the aim, then what was the aim? Why did the authors select these specific 700 papers?
>
> Thank you for your valuable feedback. The intention behind this review was to provide a comprehensive and fine-grained review of the SSL landscape and also study the works which have applied SSL principles to the domain of medical image analysis. We intended to focus on works which are significant and expand the spectra of the foundational works and recent studies as well, rather than including every single paper on the topic. More specifically, we intended to write a review consisting of two major parts, one covering the foundation of SSL, and another covering the applications of SSL in medical image analysis. We accordingly divided the broad spectra of SSL algorithms into sub-categories based on different algorithmic principles and chronologically selected papers based on a comprehensive literature search using databases such as PubMed, IEEE Xplore, Google Scholar, etc.
>
> To address your concerns, we have now added a Methods section (Sec. 1.4) to the manuscript discussing the above. This clarification will ensure that our selection process is transparent and the purpose of our survey is clear to the readers. The changes are highlighted in the latest revised manuscript.

---

> ### Author Response · Authors · 2024-06-14
> **Reply to Reviewer udZB - Part 2**
>
> >section 1.3: this section caught my attention as likely having been either written or edited by ChatGPT. Given the dangers of using ChatGPT for writing reviews (as it can confabulate references etc.), I would like to ask the authors to insert some declaration on LLM usage. Did they use LLMs at all? If yes, how can they be sure that nothing was confabulated? E.g. if they only used it for minor editing, then it's okay. Perhaps this can be added into the Methods.
>
> We understand the concern of the reviewer regarding Section 1.3 being edited or written using ChatGPT. We suppose the concern arises primarily due to the presence of the word “delve”, as recent reports suggest that this phrase is being found more frequently in texts written by ChatGPT (Nguyen, J. (2024)). However, we would like to assure the reviewer that no part of this manuscript was written or edited using ChatGPT, and all the references in this manuscript do exist. We have included a declaration statement in the Methods section (Sec. 1.4) as well.
>
> >Section 2.3.2: self-distillation methods like DINO are listed as "non-contrastive" whereas distillation methods like SimSiam are listed as "dimension-contrastive". Why is that? I did not find the rationale for this classification, or perhaps overlooked it. Since this affects the whole taxonomy (Figures 1/2 etc), I would suggest that the authors motivate their taxonomic choices better and provide rationale for their choices.
>
> Thank you very much for this comment. The rationale is based on the difference in principles between SimSiam and DINO. DINO follows a knowledge distillation framework, whereas SimSiam does not use any momentum encoder. As we have categorized, SimSiam uses a Siamese principle. The objectives of both are also different.
> DINO uses a self-distillation framework which uses a cross-entropy loss to match the probability distributions of the student and the momentum-updated teacher network.  However, in SimSiam, although the basic underlying framework is asymmetric, the objective employs cosine similarity to match the representations of two different views. Hence, these two frameworks are categorized separately.
> We thank the reviewer for pointing this out in our taxonomy. As the framework, SimSiam and other similar frameworks should rightly be categorized under “Non-contrastive” frameworks. Instead of “Dimension Contrastive”, we believe “Implicit Variance Regularization” should be a more apt name for the category. This is supported by the findings of Tian et al. (2021), where the authors say that the predictor acts as a whitening transformation. Furthermore, Halvagal et al. (2023), show that both BYOL and SimSiam perform implicit variance regularization through eigenspace analysis of the predictor. We have updated the taxonomy figures and included these newfound references in the revised manuscript (Sec. 2.3.2) as supporting evidence.
>
> >It would be nice to have some explicit discussion/overview on the uses of SSL in medical image analysis. What are the downstream tasks for which SSL makes sense? Is it mostly classification? Or something else too? Some recent papers used SSL for 2D visualisation -- e.g. https://arxiv.org/abs/2402.14566, based on https://openreview.net/forum?id=nI2HmVA0hvt. This could be added as another use-case of SSL approaches.
>
> Thank you for your valuable comment. We understand your suggestion and agree that including the works referred by the reviewer could add another dimension to our work. We have added an explicit overview of the uses of SSL in the medical image analysis domain in Sec. 3. We also discussed the tasks which are common in the medical image analysis domain. Furthermore, we have added a subsection discussing the works on medical image visualization in Sec. 3.9. We have highlighted the recommended changes in the revised manuscript.

---

> ### Author Response · Authors · 2024-06-14
> **Reply to Reviewer udZB - Part 3**
>
> >MINOR COMMENTS
> >
> >page 2: "We also discuss the work DeSa (1993) which can be ..." -- this sentence is unclear as it gives two references and it's unclear which one refers to which.
>
> Thank you for pointing out this confusing statement. We have rectified the sentence and highlighted the change in the latest revised version of the manuscript.
>
> >sometimes \citep and \citet are not used appropriately, e.g. the first line of section 1.1 uses \citet but should use \citep (or alternatively remove the words "the authors").
>
> Thank you for your comment. We have gone through the manuscript and made the necessary changes. Changes are highlighted in the revised manuscript.
>
> >page 3: DINO (Caron et al 2021) is listed in the paragraph about transfer learning. Why is that? Isn't DINO a self-distillation framework that can be called "self-supervised only"?
>
> Thank you for this comment. We would like to point out that the indicated paragraph only discusses a characteristic of some approaches which use a pre-trained network for the self-supervised pre-training stage. As for the given example, we simply state that the framework UP-DETR uses a network pre-trained using DINO as the encoder.
>
> >Section 3.6.2: this is the only subsection in section 3 where I know the literature a little bit, and I found some clear omissions, e.g. https://www.biorxiv.org/content/10.1101/2023.08.22.554251v1 or https://arxiv.org/abs/2212.04690.
>
> Thank you for your feedback and for highlighting the omissions in Section 3.6.2. We appreciate your expertise and the specific references you provided. We have reviewed the suggested papers and incorporated the relevant information into the revised manuscript to ensure a more comprehensive coverage of the literature. The added sentences are highlighted in yellow in the revised manuscript (Sec. 3.6.2).
>
> *References*:
>
> Jeremy Nguyen, (2024, March 30). Are medical studies being written with ChatGPT?  X. Retrieved June 12, 2024, from https://x.com/JeremyNguyenPhD/status/1774021645709295840
>
> Yuandong Tian, Xinlei Chen, and Surya Ganguli. Understanding self-supervised learning dynamics without contrastive pairs. In Proceedings of the 38th International Conference on Machine Learning, ICML 2021, 18-24 July 2021, Virtual Event, Volume 139 of
> Proceedings of Machine Learning Research, pp. 10268–10278. PMLR, 2021.
>
> Manu Srinath Halvagal, Axel Laborieux, and Friedemann Zenke. Implicit variance regularization In non-contrastive SSL. In Advances in Neural Information Processing Systems 36: Annual Conference on Neural Information Processing Systems 2023, NeurIPS 2023, New Orleans, LA, USA, December 10 - 16, 2023, 2023.

---

> ### Comment · Action_Editor_yxMC · 2024-06-27
> **Please check the authors response and start the discussion**
>
> Dear reviewer udZB,
>
> Thank you for reviewing the submission!
>
> Now the authors have submitted their responses to your review comments, please take a look at the responses and see if they have addressed your concerns. Please do not hesitate to start the discussion if you have any questions/concerns.
>
> Best,
>
> AE

---

> > ### Comment · Reviewer_udZB · 2024-06-27
> >
> > I don't see any responses from the authors. Neither to my review, nor to other reviews.

---

> ### Author Response · Authors · 2024-06-27
> **Reply to Reviewer udZB**
>
> Apologies from our side. There was a mistake in selecting the "Readers" setting. We have changed it. We hope you can see the responses now.

---

> ### Comment · Reviewer_udZB · 2024-06-27
>
> Thanks. Now it works.

---

> > ### Comment · Reviewer_udZB · 2024-06-28
> > **Thank you**
> >
> > I read the other reviews, the authors' responses, and looked at the revised manuscript. In my opinion, the paper is ready for acceptance.

---

### Review · Reviewer_uALV · 2024-06-23

**Summary Of Contributions:**

This paper provides an extensive survey of self-supervised learning (SSL) methodologies applied to medical image analysis. The authors discuss the evolution of SSL, a subcategory of unsupervised learning, which is especially useful where labeled data is scarce or costly to obtain, as in medical imaging. They detail various SSL approaches used in different modalities of medical imaging, like MRI and X-rays, and discuss their applications. The survey also compares recent SSL methods, categorizes them based on the SSL strategy and modality, and offers a compilation of datasets used and performance metrics.

**Audience:**

Yes

**Broader Impact Concerns:**

none noted.

**Claims And Evidence:**

Yes

**Requested Changes:**

please include discussions that are more neural,

 - such as limitations of the current SSL applied to medical images
 - and based on the limitations, suggestions to the future work.

**Strengths And Weaknesses:**

strengths:
  - The survey thoroughly reviews the state of SSL in medical image analysis, presenting a wide array of methods and categorizations that offer readers a clear understanding of the field.
  - It includes detailed discussions and comparisons of different SSL methodologies, making it a valuable resource for researchers to understand nuances and advancements.
  - By providing information on datasets and performance metrics, the paper serves as a practical guide for practitioners to choose appropriate methods for their specific applications

weakness:
  - while the paper has a comprehensive list of papers, it will serve the community better if the authors include certain discussions such as limitations and suggestions of the future directions.

---

> ### Author Response · Authors · 2024-06-27
> **Reply to Reviewer uALV**
>
> Thank you for reviewing our paper and for your thoughtful and constructive feedback. Your feedback is instrumental in improving the quality of this work.
>
> > *Summary Of Contributions:*
>
> > This paper provides an extensive survey of self-supervised learning (SSL) methodologies applied to medical image analysis. The authors discuss the evolution of SSL, a subcategory of unsupervised learning, which is especially useful where labeled data is scarce or costly to obtain, as in medical imaging. They detail various SSL approaches used in different modalities of medical imaging, like MRI and X-rays, and discuss their applications. The survey also compares recent SSL methods, categorizes them based on the SSL strategy and modality, and offers a compilation of datasets used and performance metrics.
>
> > *Strengths And Weaknesses:*
>
> > strengths:
> > - The survey thoroughly reviews the state of SSL in medical image analysis, presenting a wide array of methods and categorizations that offer readers a clear understanding of the field.
> > - It includes detailed discussions and comparisons of different SSL methodologies, making it a valuable resource for researchers to understand nuances and advancements.
> > - By providing information on datasets and performance metrics, the paper serves as a practical guide for practitioners to choose appropriate methods for their specific applications
>
> We thank the reviewer for recognising the strengths of our work.
>
> > *weakness:*
> > - while the paper has a comprehensive list of papers, it will serve the community better if the authors include certain discussions such as limitations and suggestions of the future directions.
>
> Thank you for your constructive comments. We agree that including the limitations and future directions of research on SSL in the medical image analysis domain will certainly improve this review and be helpful to the community. We have considered it in the revised version, accordingly.
>
> > *Requested Changes:*
>
> >please include discussions that are more neural,
> > - such as limitations of the current SSL applied to medical images
> > - and based on the limitations, suggestions to the future work.
>
>  Thank you for the suggestions. As discussed above, we have considered it in the revised version. We have added a section named "Challenges and Limitations" in Sec. 6 and discussed the future directions in Sec. 7 (Conclusion and Future Directions). The changes are marked in violet.

---

> > ### Comment · Reviewer_uALV · 2024-07-04
> > **reply to the rebuttal**
> >
> > Thanks for the responses, I believe the response has addressed my concerns.

---

> ### Comment · Action_Editor_yxMC · 2024-07-04
> **Please check the authors response and start the discussion**
>
> Dear reviewer uALV,
>
> Thank you for reviewing the submission!
>
> Now the authors have submitted their responses to your review comments, please take a look at the responses and see if they have addressed your concerns. Please do not hesitate to start the discussion if you have any questions/concerns.
>
> Best,
>
> AE

---

### Author Response · Authors · 2024-07-19
**Reply to all Reviewers and Action Editor**

Dear Action Editor and Reviewers,

We express our sincere gratitude for the time and effort you invested in reviewing our manuscript. Your insightful comments and constructive feedback have been invaluable in refining the manuscript, and the clarity and quality of the paper have been enhanced. We appreciate your contributions and the swiftness and thoroughness of the review process.

Thank you once again for your valuable feedback and your contribution to the improvement of the manuscript.

Thanks and regards,

Paper 2448 Authors

---

### Decision · Action_Editor_yxMC · 2024-07-22

**Recommendation:** Accept as is

**Comment:**

In this paper, the authors presented a survey of self-supervised representation learning with a particular focus on medical image analysis. A review of various existing self-supervised learning methods with detailed analysis and discussions is presented. A discussion and review on benchmarking datasets (for both image and video data) along with the performance of different methods are included as well. Challenges and limitations are discussed briefly in the end.

Overall, the paper provided a detailed and extensive survey of the self-supervised representation learning for medical image analysis, and could be of interest to a group of audience in TMLR.

The paper was reviewed by three domain experts. Strengths and weaknesses were identified by the reviewers. Through back-and-forth discussion and revision, the raised concerns from the reviewers were addressed by the authors and reflected in the updated revised version of the paper. In the end, all the three reviewers were happy with the updated paper and recommended positive scores (2 Accept and 1 Leaning Accept).

As a result, the AE is pleased to inform the authors that this paper is accepted to be published in TMLR!

Best,

AE

**Audience:**

Yes, the AE believes there would be a (large) group of individuals in TMLR's audience be interested in knowing the findings of this paper.
This paper is a survey of self-supervised representation learning particularly in medical image analysis. It presented a good overview of existing related works and summarised the corresponding algorithms, datasets, and performance, which could be a good reference and guidance for related researchers.

**Claims And Evidence:**

Yes, the claims made in the submission were supported by accurate, convincing and clear evidence.